



# MIPAS IMK/IAA version 8 retrieval of nitric oxide and lower thermospheric temperature

Bernd Funke[1], Maya García-Comas[1], Norbert Glatthor[2], Udo Grabowski[2], Sylvia Kellmann[2], Michael Kiefer[2], Andrea Linden[2], Manuel López-Puertas[1], Gabriele P. Stiller[2], and Thomas von Clarmann[2]

[1]Instituto de Astrofísica de Andalucía, CSIC, Spain
[2]Karlsruhe Institute of Technology, Institute of Meteorology and Climate Research, Karlsruhe, Germany

**Correspondence:** Bernd Funke (bernd@iaa.es)

**Abstract.** New global nitric oxide (NO) volume mixing ratio and lower thermospheric temperature data products, retrieved from Michelson Interferometer for Passive Atmospheric Sounding (MIPAS) spectra with the IMK-IAA MIPAS data processor, have been released. The dataset covers the entire Envisat mission lifetime and includes retrieval results from all MIPAS observation modes. The data are based on ESA version 8 calibration and were processed using an improved retrieval approach compared to previous versions, specifically regarding the choice and construction of the a priori and atmospheric parameter profiles, the treatment of horizontal inhomogeneities, the treatment of the radiance offset correction, and the selection of optimized numerical settings. NO retrieval errors of individual observations are dominated by measurement noise and range from 5% to 50% in the stratosphere and thermosphere, and reach 40% to 90% in the mesosphere. Systematic errors are typically within 10–30%. Lower thermospheric temperature errors are 5 K to 50 K with a systematic component of around 10 K, the latter being dominated by non-LTE related uncertainties. NO data from different observation modes are consistent within 5–10%. MIPAS version 8 temperatures have a better representation of the diurnal tide in the lower thermosphere compared to previous data versions. The new MIPAS temperatures are systematically warmer than results from the empirical NLRMSIS2.0 model by 30 K to 80 K in the 100–120 km region, and are colder above.

## 1 Introduction

Nitric oxide (NO) is a key agent of atmospheric chemistry over a wide altitude range. It acts as a pollutant near the surface, interferes with stratospheric ozone chemistry by catalytic reactions, plays a key role in transferring space weather impacts from the upper atmosphere down to lower altitudes, and drives the lower and middle thermospheric heat balance by infrared cooling. NO has been measured from ground and from space using different techniques (Barth, 1964; Eparvier and Barth, 1992; Rusch and Barth, 1975; Zachor and Sharma, 1985; Barth et al., 1988; Russell III et al., 1988, 1993; Taylor et al., 1993; Barth et al., 2003; Bernath et al., 2005; Funke et al., 2005; Bermejo-Pantaleón et al., 2011; Bender et al., 2013; Pérot et al., 2014; Bailey et al., 2014).

One of the instruments providing NO observations was the Michelson Interferometer for Passive Atmospheric Sounding (MIPAS) on board the Envisat satellite, a limb-viewing mid-infrared Fourier transform spectrometer designed to sound tem-



perature and trace gas abundances from atmospheric emissions (Fischer et al., 2008). During the mission lifetime from 2002
25  to 2012, high-resolution spectra at 4.15–14.6 $\mu$m were measured globally from a polar sun-synchronous orbit with equator
crossing times at 10 a.m. and 10 p.m. in its descending and ascending nodes, respectively, hereinafter referred to as "am" and
"pm" measurements. MIPAS operated at an apodized spectral resolution of 0.05 cm$^{-1}$ during the "full spectral resolution"
(FR) period until March 2004. Due to a technical defect, the spectral resolution was reduced afterwards to 0.125 cm$^{-1}$ while
the vertical sampling was increased. We refer to this second operation phase with degraded spectral resolution as "reduced
30  spectral resolution" (RR) period. Most of the time, MIPAS operated in the nominal measurement mode (NOM) with 17 and
tangent altitudes for FR and RR, respectively, covering the vertical range of approximately 6–70 km. The NOM vertical
sampling increases with altitude from 3 km (below 50 km) altitude to 8 km for FR and from 1.5 km (below 50km) to 4.5 km for
RR. Further modes, taken less frequently (about one out of ten days) during the RR period, target the middle and upper atmo-
sphere (MA and UA modes, respectively). In addition, noctilucent cloud measurements (NLC mode) were taken during about
8 days per year during solstice periods in the RR period. The spectra taken in the MA, UA, and NLC modes were recorded
at 29, 35, and 25 tangent heights and with an altitude coverage of 18–102 km, 42-172 km, and 39–102 km, respectively. The
vertical sampling of the MA and UA modes is 3 km below 102 km altitude and 5 km above. NLC mode measurements have a
denser sampling of 1.5 km in the 78–87 km range.

Vertical abundance profiles of NO are retrieved from MIPAS measurements with the scientific level-2 processor developed
and operated by the Institute of Meteorology and Climate Research (IMK) at the Karlsruhe Institute of Technology, and the
Instituto de Astrofísica de Andalucía (IAA, CSIC). The retrieval uses spectral lines of the NO fundamental band at 5.3 $\mu$m and
requires the consideration of non-thermodynamic equilibrium (non-LTE) for vibrational, rotational and spin states emitting in
this band. The first MIPAS NO observations obtained from FR NOM measurements were described nearly two decades ago
by Funke et al. (2005). Since then, the level-1b processing and the level-2 retrieval algorithm have been updated and applied
to the entire FR and RR dataset (Funke et al., 2014), as well as to MA, UA and NLC observations. For UA measurements
above approximately 100 km, the retrieval had to be modified to include the kinetic temperature as a joint fit parameter due to
the unavailability of reliable temperature information from $CO_2$ 15 $\mu$m emissions at these altitudes (Bermejo-Pantaleón et al.,
2011). The resulting 5.3 $\mu$m temperature product represents one of the very few available global kinetic temperature datasets
in the lower and middle thermosphere. MIPAS NO and thermospheric temperature data have been used widely in a variety of
scientific studies (Funke et al., 2010, 2011; von Clarmann et al., 2013; Oberheide et al., 2013; Xu et al., 2013; Funke et al.,
2014; García-Comas et al., 2016; Klimenko et al., 2019; Pettit et al., 2019; Emmert et al., 2021; Sinnhuber et al., 2022, among
others).

Recently, the European Space Agency (ESA) has distributed the new level-1b data version 8.03 which improves upon
previous versions in several aspects, in particular, by use of a time-dependent model of detector non-linearity (see Kiefer
et al., 2021, for more details). In this paper, we describe the reprocessed set of MIPAS 5.3 $\mu$m data products that build on this
new level-1b data version 8.03. In addition, it includes several updates of the level-2 algorithm that aim at addressing issues
encountered in validation activities and science studies. This new dataset covers NOM, MA, UA, and NLC observations taken
during both FR and RR periods. Specifically, it contains NO level-2 data versions V8H_NO_61 (NOM) for the FR period, and





V8R_NO_261 (NOM), V8R_NO_561 (MA), and V8R_NO_761 (NLC) for the RR period. The NO and temperature products from UA observations are labeled V8R_NOwT_662 and V8R_TwNO_662, respectively.

The retrieval setup and the improvements with respect to previous level-2 versions are discussed in Section 2. Averaging kernels and vertical resolution of the retrieved profiles are discussed in Section 3. The uncertainty budget is presented in Section 4. We discuss the characteristics of the new dataset, differences with respect to previous versions, and the consistency of data from different observation modes in Section 5 for NO, and in Section 6 for the lower thermospheric temperature.

## 2    Retrieval

The IMK-IAA level-2 processor relies on multi-parameter non-linear least squares fitting of measured and modeled spectra (von Clarmann et al., 2003). Its extension to retrievals involving non-LTE emissions is described in Funke et al. (2001). The underlying mathematical framework for V8 retrievals is described in detail in Kiefer et al. (2021). The forward model incorporated in the level-2 processor is the Karlsruhe Optimized Radiative transfer Algorithm (KOPRA, Stiller et al., 2002), which, in its current version, is internally interfaced with the Generic Radiative transfer and non-LTE population algorithm (GRANADA, Funke et al., 2012). In the following, we discuss all settings relevant for the retrieval of NO volume mixing ratio (vmr) from NOM and MA mode measurements, as well as for the joint NO and lower thermospheric temperature retrieval from UA mode measurements.

Within the sequential processing chain of the IMK-IAA level-2 processor, these retrievals are performed after the determination of a frequency shift, the retrieval of tangent altitudes and temperature from 15 $\mu$m $CO_2$ emissions, ozone ($O_3$), water vapor ($H_2O$), methane ($CH_4$), nitrous oxide ($N_2O$), and nitrogen dioxide ($NO_2$). The 15 $\mu$m temperature retrieval provides the temperature profile up to a maximum altitude of approximately 115 km. The $O_3$ retrieval provides information on atomic oxygen (O) concentrations below ∼95 km, required for the NO non-LTE modeling (see Sec. 2.5), and constraints interfering $O_3$ emission contributions in the 5.3 $\mu$m region. Similarly, information on $H_2O$, $N_2O$ and $CH_4$ are required to account for spectral interferences in the upper troposphere and stratosphere. $NO_2$ is required for the NO non-LTE modeling (see Sec. 2.5) and a priori generation (see Sec 2.3). It is clear that improvements in the version 8 retrievals of these parameters, compared to older versions, will also improve the quality of the 5.3 $\mu$m data products. The version 8 temperature retrieval is documented in Kiefer et al. (2021) for NOM mode measurements and in García-Comas et al. (2022) for MA, UA, and NLC measurements. The new ozone data product is described in Kiefer et al. (2022) for NOM measurements and in López-Puertas et al. (2022) for MA, UA, and NLC measurements. The documentation of other version 8 data products is underway.

### 2.1    The unknowns of the retrieval

The target quantity of the retrieval is the profile of NO vmr which, in the case of UA retrievals is accompanied by that of the kinetic temperature, in the vertical range from the ground up to 200 km. Since version 4, and in contrast to the original FR setup described in Funke et al. (2005), NO is retrieved as the natural logarithm of the vmr to implicitly adjust the strength of the retrieval constraint (see Section 2.2) to the large dynamical range of atmospheric NO abundances.



NO and temperature profiles are represented on a discrete retrieval grid with grid widths of 1 km up to 56 km, 2 km at 56–70 km, 2.5 km at 70–115 km, and 5 km at 115–150 km in the NOM, MA, and NLC modes. Grid points covering higher altitudes are 160, 170, 180, and 200 km. The UA retrieval grid is identical up to 115 km, but uses a finer discretization above with grid widths of 2.5 km up to 130 km and 5 km at 130–200 km. Version 8 NOM, MA, and NLC retrieval grids are more resolved in the region of the lower thermospheric NO density maximum at 105–115 km compared to previous retrieval versions (2.5 km versus 5 km, respectively). Similarly, version 8 UA retrievals have a denser grid in the 105–200 km region, whereas previous UA versions (Bermejo-Pantaleón et al., 2011) were based on the same grid as used in NOM, MA, and NLC retrievals. Our new UA retrievals differ from previous versions also in the sense that the retrieval is performed using spectra from the entire scan range 42-172 km while in earlier versions the retrieval was split into two parts with a NO-only retrieval performed in the 42-102 km range and a joint NO and temperature retrieval in the 90-172 km range. For that reason, three UA data products were previously provided, e.g., V5r_NO_622 using the 42-102 km range, and V5r_NOwT_622 and V5r_TwNO_622 from the joint NO and temperature retrieval in the 90-172 km range.

In addition to the target quantities, NO horizontal gradients, a background continuum, and a radiance offset are retrieved as well. The retrieved horizontal gradients provide a first-order correction to the assumed spatial variations of NO, the latter being provided as three-dimensional a priori fields of relative variations with respect to the NO profile at the center of scan location (see Sec. 2.3). Horizontal gradients are implemented in terms of two profiles accounting for relative linear variations in latitudinal and longitudinal directions (in units of $km^{-1}$), respectively, both discretized in the same way as the target quantity NO.

Joint-fitting of background continuum profiles (in terms of optical depth with units of $km^{-1}$) is a standard feature of all MIPAS retrievals (e.g., von Clarmann et al., 2003). It was introduced to account for radiance contributions which are not considered in the line-by-line calculation of absorption cross-sections, or which are emitted by non-gaseous components of the atmosphere like clouds, aerosols, volcanic ash or meteoric dust. Since such contributions are more important in the lower atmosphere, previous NOM and MA retrievals (up to version 5 for NOM, and up to version 4 for MA) included the background continuum profiles up to 33 km altitude for each spectral window considered in the retrieval (hereinafter referred to as microwindows, see Sec 2.6). It turned out, however, that consideration of the background continuum up to higher altitudes improved the robustness of the retrievals and removed known biases in retrieved state variables, as it allows to account for possible meteoric dust contributions (Neely et al., 2011) and residual ozone non-LTE emissions at 5.3 $\mu$m from very high-energetic bands that are not included in the radiative transfer modeling. For that reason, the maximum altitude of the continuum profiles was increased to 60 km in version 5 MA and NLC retrievals. In version 8, we have further extended the vertical range to 68 km for NOM and to 72.5 km for MA, UA, and NLC retrievals.

Besides the background continuum, we also retrieve for each microwindow a radiance offset profile which is meant to correct the zero level radiance calibration. In previous versions, a scalar offset correction has been used. While the continuum is additive to the absorption coefficient, the offset correction adds directly to the radiances. However, in the case of linear radiative transfer, the altitude-dependent offset correction and the background continuum cannot be distinguished and the simultaneous retrieval of both leads to a null space of solutions. This problem is solved by strongly constraining the vertical





**Table 1.** Altitude dependence of the smoothing regularization term $\gamma_S$ for ln(NO) in NOM, MA, and NLC retrievals, as well as ln(NO) and temperature in UA retrievals.

| Altitude (km) | NO NOM/MA/NLC (none) | NO UA (none) | temperature UA (K$^{-2}$) |
|---:|---:|---:|---:|
| 0 | 100.0 | 100.0 | 4.00 |
| 10 | 10.0 | 10.0 | 4.00 |
| 15 | 5.0 | 5.0 | 4.00 |
| 20 | 3.0 | 3.0 | 4.00 |
| 25 | 2.3 | 2.3 | 4.00 |
| 35 | 2.0 | 2.0 | 4.00 |
| 40 | 1.8 | 1.8 | 4.00 |
| 50 | 1.8 | 1.8 | 4.00 |
| 60 | 2.7 | 2.7 | 4.00 |
| 70 | 3.6 | 3.6 | 4.00 |
| 80 | 4.6 | 4.6 | 4.00 |
| 90 | 6.0 | 6.0 | 4.00 |
| 100 | 7.0 | 5.0 | 4.00 |
| 105 | 7.0 | 5.0 | 0.15 |
| 110 | 7.0 | 5.0 | 0.11 |
| 120 | 13.0 | 5.0 | 0.04 |
| 130 | 20.4 | 5.0 | 0.03 |
| 150 | 22.8 | 8.0 | 0.05 |
| 180 | 28.5 | 14.9 | 0.14 |
| 200 | 60.0 | 18.0 | 0.36 |

offset profile towards an empirically determined offset correction profile (Kleinert et al., 2018), which is used as a priori for the fit of the zero level radiance correction.

## 2.2  Regularization

Version 8 5.3 $\mu$m retrievals are regularized using a smoothing term based on a squared first order difference cost function
(see, e.g., Tikhonov, 1963). In addition, a diagonal term which pushes the result towards the a priori profile values, similar as in optimal estimation or maximum a posteriori retrievals (Rodgers, 2000), can be applied locally, if required, in order to stabilize the retrieval. More details on the mathematical framework and implementation are provided in Kiefer et al. (2021). Our approach differs from the old approach of Steck and von Clarmann (2001) used in previous retrieval versions.

The altitude-dependent regularization acts on the logarithm of vmr, not on vmr. The choice of ln(vmr) for the representation
of the target variable has the effect that smoothness of the resulting profile is obtained in terms of vmr ratios between adjacent





altitudes rather than the vmr gradient. With this, a self-adaptive effect of regularization is achieved, where small mixing ratios are constrained stronger than large ones. The altitude dependence of both the smoothing and the diagonal regularization terms is controlled by so-called $\gamma$-vectors (c.f. Eq. 2 of Kiefer et al., 2021). Table 1 summarizes the chosen $\gamma_S$ values at given altitudes for the smoothing term used in the NO retrievals from NOM, MA, and NLC mode observations, as well as those used

in the joint NO and temperature retrieval from UA observations. At altitudes above 100 km, the smoothing term used here for NOM, MA, and NLC retrievals is about 20-50% weaker than in previous versions. In UA retrievals, the new NO smoothing constraint is a factor of 2 to 4 weaker than in version 5 in the vertical range 100–150 km, while the temperature smoothing constraint has been weakened by a factor of 4 to 8 and only in the 105–115 km range.

The diagonal term is employed at the two lowermost NO profile gridpoints ($\gamma_D$ values of 100 and 36 at altitudes of 0 km

and 4 km, respectively). NOM, MA, and NLC retrievals, which do not include measurements at thermospheric tangent heights, use further a weak diagonal constraint ($\gamma_D = 0.4$) for the NO profile above 150 km in order to stabilize the retrievals at high altitudes. Otherwise, the diagonal constraint is set to zero. For the temperature profile in the UA retrievals, we use a strong diagonal constraint ($\gamma_D = 100$ K$^{-2}$) below 100 km in order to fix the profile to the a priori profile taken from the 15 $\mu$m temperature retrieval. Above, a weak diagonal constraint ($\gamma_D = 10^{-5}$ K$^{-2}$) is used in order to avoid unphysical temperature

values.

The NO horizontal gradients are regularized towards zero with an altitude-dependent diagonal term ($\gamma_D$ values of $10^5$–$10^7$ km$^2$). In version 8, we have added a smoothing term with a constant $\gamma_S$ value of $10^3$ km$^2$ in order to stabilize the retrieval.

Only a smoothing constraint is applied to the continuum profile ($\gamma_D = 5 \times 10^3$ km$^2$ below 60 km, increasing to $10^5$ km$^2$ at higher altitudes). Above 68 km (72.5 km for UA), the continuum is forced to zero by a strong diagonal term. We also apply a

smoothing constraint in the frequency domain in order to avoid unrealistic jumps of the background continuum between adjacent microwindows. The radiance offset profile per microwindow is regularized using both a diagonal and a strong smoothing constraint. The diagonal term corresponds to a variance in the order of the offset uncertainty obtained by Kleinert et al. (2018). No regularization of the offset in the frequency domain has been applied.

## 2.3 A priori temperature and trace gas distributions

The selection of adequate a priori profiles is of high importance for the retrieval of atmospheric parameters from 5.3 $\mu$m measurements, in particular for those from NOM, MA, and NLC measurements which have to deal with significant NO emission contributions from thermospheric altitudes that are not covered by the scan range. Further, the joint retrieval of NO vmr and temperature from UA measurements, which exploits the complementary information provided by the intensity and the rotational envelope of the NO fundamental band, is affected by smoothing error cross-talk (see Sec. 4.1.4). The magnitude of the

introduced errors depends strongly on the quality of the a priori profile. In addition, tropospheric and stratospheric daytime NO abundances are in photochemical equilibrium with NO$_2$, a species that can be observed by MIPAS with high precision and good vertical resolution. This is exploited for the retrieval of NO by using an a priori profile that is derived from the retrieved NO$_2$ by means of a photochemical model. Since the smoothing constraint used in the NO retrieval penalizes deviations from





the a priori shape, the photochemically constrained a priori adds information on the vertical structure of the NO profile which
cannot be entirely resolved from 5.3 $\mu$m measurements.

### 2.3.1   NO a priori information

The NO a priori profiles are constructed differently in five vertical regions:

- Above 120 km (*region 1*), it is computed from an empirical model based on MIPAS version 5 data. This model uses similar regression terms as the Odin/SMR-based SAMONA model (Kiviranta et al., 2018), but adds a semi-annual term
and two cross terms (F10.7$\times$inclination and A$_p\times$inclination) accounting for seasonal modulations of the NO response to solar-geomagnetic forcing. Further, the MIPAS-based model computes am and pm NO concentrations separately.

- At 93–120 km (*region 2*), we use a climatology obtained from Whole Atmosphere Community Climate Model (WACCM) Version 4 (Marsh et al., 2013) fields of a specified dynamics run (Garcia et al., 2017), which provided output specifically for the MIPAS measurement geolocations and times. In order to avoid discontinuities at the upper edge of
this region, the profiles from the WACCM climatology are scaled to fit the NO concentration from the MIPAS-based empirical model at 120 km.

- Below 65 km at nighttime and 45 km at daytime (*region 3*), the NO a priori is computed with a photochemical box model that incorporates the results from the preceding NO$_2$ and O$_3$ retrievals. The box model is an updated version of the one described in Funke et al. (2005, see their Table 2) for computation of the partitioning of odd nitrogen and odd
oxygen species. The updates include (i) the consideration of the additional reaction pathways OH + OH $\rightarrow$O + H$_2$O, OH + O$\rightarrow$O$_2$ + H, and N($^4$S,$^2$D) + O$_2$ $\rightarrow$NO + O; (ii) the use of kinetic rate constants from the JPL evaluation no. 18 (Burkholder et al., 2015); (iii) the use of version 5.3 of the TUV model (Madronich and Flocke, 1998) for photolysis rate computation; and (iv) the consideration of transient solar irradiance variations in the latter model, based on Matthes et al. (2017). Besides NO$_2$, temperature, and O$_3$, which are taken from preceding retrievals, information on the abundances of
OH, H, ClO, N($^4$S) and N($^2$D) is required for the box model calculations. We take ClO abundances from the IG2 MIPAS database (Remedios et al., 2007), while OH and H are available from the WACCM climatology described above. Data sources for N($^4$S) and N($^2$D) are discussed in Sec. 2.3.3.

- Above, and up to 85 km (*region 4*), we use the same box model to compute the ratio NO/NO$_x$ for the actual conditions, which is then multiplied to NO$_x$ from the WACCM climatology. The resulting NO profile is then scaled to match the
NO abundance computed by the box model at the upper boundary of region 3.

- Finally, between 85 km and 93 km (*region 5*), the profiles of region 2 and region 4 are merged by linear tapering.

In previous retrieval versions, region 1 extended from 100 km to 200 km, and the NOEM empirical model (Marsh et al., 2004) was used instead of the MIPAS-based empirical model. Since the NOEM model is based on daytime measurements from the SNOE instrument taken during 1998–2003, nighttime NO concentrations are not well described. For this reason, an



empirical nighttime correction was introduced in the NO a priori generation for version 5. Further, since the NOEM model was trained with measurements taken at solar maximum conditions, it is not well suited for the extended periods of low solar activity covered by MIPAS. In fact, the version 5 a priori tends to overestimate the observed NO concentrations in the lower thermosphere, particularly during 2007–2010.

The NO a priori in region 2 and 4 was based in previous versions on simulations with the 2D model of Garcia (1983) and was
not scaled to adjust to the NO concentration of region 1 at the upper boundary. This caused artificial jumps of the NO a priori profile at 100 km, introducing systematic features in the retrieved NO. Further, the 2D model simulations largely underestimate the NO amount around the mesospheric minimum, which led to additional problems in the retrievals of the logarithm of the vmr. If the a priori is very low, the retrieval sensitivity is largely reduced since the Jacobian matrix scales with the inverse of the vmr and thus, the retrieval solution can get trapped by the a priori. As a result, a large fraction of NO data around the
mesospheric minimum was not useful in earlier retrieval versions.

### 2.3.2 Temperature a priori information

In version 8 UA retrievals, the temperature a priori profile below 110 km is taken from the preceding 15 $\mu$m temperature retrievals. Above 120 km, the a priori profile is based on the NRLMSIS 2.0 empirical model (Emmert et al., 2021). We apply, however, a seasonal correction (dependent on month, latitude, and altitude) to the NRLMSIS temperature profiles in order to
account for biases encountered when comparing to the MIPAS climatology based on temperature data version V5r_TwNO_622. Between 110 km and 120 km, the two profiles are merged by linear tapering. In previous UA retrieval versions, we used the NRLMSISE-00 empirical model (Picone et al., 2002), without applying any seasonal correction. Further, the transition between the 15 $\mu$m temperature profile and that from NRLMSISE-00 was performed at lower altitudes, between 97 km and 110 km. Since the temperature sensitivity in 5.3 $\mu$m retrievals is weak below 115 km, the obtained temperatures were strongly influenced
by the NRLMSISE-00 a priori in this region, instead of being constrained by the observed 15 $\mu$m temperatures.

The pressure profile is recalculated in the UA retrievals by numerical integration of the hydrostatic equation in each retrieval iteration. The altitude-dependent mean molecular weight is computed using the relative abundances of main constituents from the NRLMSIS 2.0 model, except for atomic oxygen below ∼130 km (see Sec. 2.3.3). In the hydrostatic adjustment, the pressure–altitude relation is kept fixed at at an altitude close to the lowermost tangent height, where it is derived from ECMWF
ERA-Interim reanalysis fields (Dee et al., 2011).

### 2.3.3 Atmospheric profile parameters for radiative transfer calculations

Several atmospheric parameter profiles, which do not form part of the unknowns of the retrieval, have to be provided for the radiative transfer calculations. The thermospheric temperature profile used in NOM, MA, and NLC retrievals, where temperature is not fitted simultaneously, is constructed in the same way as the temperature a priori profile in the UA retrievals. Here,
the hydrostatic adjustment is performed prior to the retrievals.

Several molecular species contribute to the radiance spectra measured at 5.3 $\mu$m and have to be considered. Besides $O_3$, $H_2O$, $N_2O$ and $CH_4$, available from preceding retrievals, these include $CO_2$, whose abundances are taken from a SD-WACCM4-based





monthly zonal mean dataset. Minor interferences at lower altitudes are produced by OCS, $COF_2$, acetone, and peroxyacyl nitrate. These species, which are retrieved from MIPAS measurements at a later step in the retrieval sequence, are taken into account with the profiles from MIPAS version 5 retrievals. Earlier $5.3\,\mu m$ retrieval versions used climatological data instead.

Abundances of several atmospheric compounds are also required as input for the non-LTE model calculations (see Sec 2.5). Besides $NO_2$, these are O, $O_2$, $N_2$, $N(^4S)$, and $N(^2D)$. Below 95 km, O abundances are computed with the photochemical box model described in Sec. 2.3.1. At 95–130 km, they are taken from the SD-WACCM4 output which is provided at MIPAS geolocations and times. Above, they are taken from the NRLMSIS 2.0 model, which also provides the abundances of $O_2$, $N_2$ and $N(^4S)$. For the computation of $N(^2D)$ abundances, we follow the approach of Vitt et al. (2000). Atmospheric ionization rates due to energetic particle precipitation, required for these calculations, are taken from the AIMOS model version 1.6 (Wissing and Kallenrode, 2009).

In previous retrieval versions, we extracted O ($>130\,km$), $O_2$, $N_2$, and $N(^4S)$ from the older NRLMSISE-00 model. $N(^2D)$ abundances were estimated from $N(^4S)$ by means of a simple parameterization of the $N(^2D)/N(^4S)$ ratio based on early model calculations (Fesen et al., 1989). The new approach, which accounts for observed particle fluxes by means of the AIMOS model, allows for a more realistic representation of $N(^2D)$ production in the auroral regions, thus improving the representation of NO non-LTE excitations by the recombination of $N(^2D)$ with molecular oxygen (Funke and López-Puertas, 2000) during geomagnetically active periods.

## 2.4 Horizontal inhomogeneity of NO and temperature distributions

The need to consider horizontal inhomogeneities of stratospheric NO distributions along the line of sight (LOS) during twilight conditions, caused by photochemically induced gradients, was already identified by Funke et al. (2005). In version 8, we follow the same approach as described in that work, that is, we constrain the NO variations along the LOS by means of the photochemical box model described in Sec. 2.3.1. In practice, relative NO variations with respect to the center-of-scan profile are computed under the assumption of a horizontally constant $NO_x$ abundance and are provided as 3D fields with a latitudinal $\times$ longitudinal discretization of $0.5° \times 1°$.

In addition, spatial variations of thermospheric NO abundances in the polar regions, caused by auroral productions, have an important impact on $5.3\,\mu m$ retrievals. Since thermospheric NO emissions contribute substantially to the radiances observed at lower tangent heights, and because the horizontal portion of the thermosphere "seen" at lower tangent heights varies with the tangent height itself, the neglect of spatial variations in the thermosphere causes significant errors in the NO retrieval, particularly in the upper stratosphere and mesosphere (Funke et al., 2005). An attempt to consider these spatial variations in the $5.3\,\mu m$ retrieval was first made in version 5 MA, UA, and NLC retrievals. There, the NO distributions above 100 km were constrained with NOEM model fields. NOEM parametrizes the spatial structure of auroral NO by means of one out of a set of empirical orthogonal eigenfunctions multiplied by a time varying coefficient which is proportional to the geomagnetic $k_p$ index (Marsh et al., 2004). The NOEM model fields were provided in terms of relative anomalies with respect to the profile at the center-of-scan position.



It is evident that an empirical model like NOEM cannot reliably constrain the actual thermospheric distributions at a given measurement time and location. In this sense, the prescription of NOEM fields in version 5 retrievals helped to reduce biases in averaged NO data that could have been introduced by the recurrent shape of the auroral oval, while not allowing to efficiently mitigate errors in individual profiles. For this reason we adopted a different approach in version 8 and prescribe NO anomaly

fields that have been computed from daily gridded MIPAS NO data from version 5 instead of using empirical model data. These gridded maps have been computed individually for am and pm measurements in order not to mix up observations taken with a 12-hour lag.

Horizontal temperature inhomogeneities are also considered in the 5.3 $\mu$m retrievals using the approach described in Kiefer et al. (2021). In brief, ERA Interim reanalysis fields are used to prescribe the horizontal temperature anomalies along the

LOS. Above 60 km, these anomalies are calculated from NRLMSISE-00 model fields. In addition, first-order corrections in terms of latitudinal and longitudinal gradients are fitted jointly with temperature in the 15 $\mu$m retrievals in order to correct for model errors in the prescribed fields. Both, prescribed anomaly fields and retrieved gradients from the 15 $\mu$m retrievals are considered in the 5.3 $\mu$m retrievals. Horizontal temperature inhomogeneities were not considered in earlier retrieval versions with the exception of version 5 thermospheric UA retrievals, which used prescribed anomalies from NRLMSISE-00.

The inclusion of temperature variations along the LOS in non-LTE retrievals requires the consideration of non-LTE population variations which, in turn, are driven by the temperature variations. This is done by means of a non-LTE parameterization as described in Kiefer et al. (2021, Sec. 3.11) for version 8 temperature retrievals from MIPAS NOM measurements. Here, however, the required parameters are re-computed in each step of the retrieval with the incorporated GRANADA non-LTE model rather than being read from climatology.

## 2.5   Non-local thermodynamic equilibrium

The GRANADA non-LTE model setup for the calculation of NO vibrational, rotational, and spin populations is described in detail in Funke et al. (2012) and has already been used in this configuration in version 5 MA, UA, and NLC retrievals. Earlier retrieval versions for these measurement modes (i.e., Bermejo-Pantaleón et al., 2011), as well as previous NOM retrievals used a slightly simpler setup which accounted for NO vibrational states only up to $\nu = 3$ (instead of $\nu = 4$) and rotational states with

$J \leq 35.5$ (instead of $J \leq 55.5$).

Concerning the rate constants used in the modeling of collisional and chemical processes, we have incorporated several updates with respect to those listed in Funke et al. (2012). First, for the collisional relaxation of NO vibrational states with O, we have adopted the quasi-classical trajectory results by Caridade et al. (2008) for the $\nu > 1$ vibrational levels, however, scaling them to the measured value (Hwang et al., 2003) for the $\nu = 1$ state. This change affects mainly thermospheric NO

populations, however, the impact on the 5.3 $\mu$m retrieval is very small.

Second, for the reaction $N(^4S) + O_2 \rightarrow NO + O$ we now use rate coefficients from the JPL Evaluation no. 18 (Burkholder et al., 2015). The rate constant for the reaction $N(^2D) + O_2 \rightarrow NO + O$ is taken from Vitt et al. (2000). The nascent distributions of vibrationally excited NO from both reactions have been adopted from the more recent theoretical values of Sultanov and





**Table 2.** Microwindows used in 5.3 $\mu$m retrievals from FR (first column) and RR (second column) measurements.

| Wavenumber range (FR) (cm$^{-1}$) | Wavenumber range (RR) (cm$^{-1}$) | Tangent Height Range (km) |
|---|---|---|
| 1831.7000–1832.0500 | 1831.6875–1832.0625 | 9–63 |
| 1837.8250–1838.2500 | 1837.8125–1838.2500 | 15–172 |
| 1842.8250–1843.1750 | 1842.8125–1843.1875 | 12–172 |
| 1849.0750–1853.9250 | 1849.0625–1853.9375 | 9–172 |
| 1857.0000–1861.1250 | 1857.0000–1861.1250 | 6–172 |
| 1863.5000–1863.8750 | 1863.5000–1863.8750 | 9–172 |
| 1880.7500–1881.2500 | 1880.7500–1881.2500 | 9–172 |
| 1887.2500–1891.1250 | 1887.2500–1891.1250 | 9–172 |
| 1896.7500–1900.8750 | 1896.7500–1900.8750 | 6–172 |
| 1902.9500–1906.8750 | 1902.9375–1906.8750 | 6–172 |
| 1909.0000–1912.9250 | 1909.0000–1912.9375 | 6–172 |
| 1914.8250–1915.1250 | 1914.8125–1915.1250 | 6–172 |
| 1923.3750–1927.4250 | 1923.3750–1927.4375 | 18–172 |
| 1928.8750–1931.8750 | 1928.8750–1931.8750 | 6–172 |
| 1935.3250–1935.6750 | 1935.3125–1935.6875 | 6–172 |

Balakrishnan (2006) instead of those of Duff et al. (1994), resulting in 3–6% larger excitations. Again, the impact of this change

on the 5.3 $\mu$m retrieval is very small and restricted to the thermosphere.

Third, the rate coefficients for the reaction of $NO_2$ with O is now taken from the JPL Evaluation 18. The nascent distribution of vibrationally excited NO from this reaction has been taken from Smith et al. (1992), resulting in a 40% lower efficiency for the production of the $\nu \geq 1$ compared to the previous values taken from Kaye and Kumer (1987). This change induces a 2–5% increase of the retrieved NO abundance in the upper stratosphere around 40 km, which brings it into better agreement with the

values expected from photochemical equilibrium.

### 2.6 Microwindows and spectroscopic data

For reasons of computational efficiency, the retrieval does not use the entire spectra measured by MIPAS but a set of small microwindows with high sensitivity to the unknowns of the retrieval and small contaminations by interfering emissions. The microwindows used in the 5.3 $\mu$m retrievals cover a large fraction of the ro-vibrational lines of the NO fundamental band

located in MIPAS channel D. Although no changes have been introduced in the microwindow selection compared to previous versions, they are listed, together with the tangent height range where they are employed, in Table 2 for completeness.





**Table 3.** Number of retrieved profiles, convergence rate, median value of the reduced $\chi^2$, and number of rejected profiles for the version 8 MA, UA, NOM RR and NOM FR datasets. Number in brackets refer to the corresponding values for version 5 datasets.

| Mode | Total number of profiles | Convergence rate (%) | Reduced $\chi^2$ (median) | No. of rejected profiles |
|---|---|---|---|---|
| MA | $0.20 \times 10^6$ | 99.943 (99.854) | 1.025 (1.039) | 3 (-) |
| UA | $0.17 \times 10^6$ | 99.501 (99.175) | 1.175 (1.188) | 58 (-) |
| NOM (RR) | $0.51 \times 10^6$ | 99.921 (99.883) | 1.179 (1.222) | 48 (47) |
| NOM (FR) | $1.89 \times 10^6$ | 99.705 (99.026) | 1.057 (1.121) | 251 (186) |

As in other version 8 retrievals, we use the HITRAN 2016 spectroscopic database (Gordon et al., 2017) except for $O_3$, for which a dedicated MIPAS spectroscopic database (Flaud et al., 2003) was used.

## 2.7 Numerical settings

The most relevant numerical settings used in the retrievals, in particular those controlling the discretization of the radiative transfer calculations and those controlling the convergence behavior of the retrieval, are summarized below:

– The layering of the radiative transfer calculations is bound to the grid of the retrieved NO (and temperature) profile. This implies that, in the thermosphere, the layering used in version 8 UA retrievals is finer than in NOM, MA, and NLC retrievals. For all measurement modes, the layer width has decreased in the 105–115 km with respect to previous
versions.

– The spectral grid width for monochromatic radiance calculations is $3.90625 \times 10^{-4}\,\mathrm{cm}^{-1}$ for FR retrievals and $4.8828125 \times 10^{-4}\,\mathrm{cm}^{-1}$ for RR retrievals, slightly smaller than in previous versions ($5 \times 10^{-4}\,\mathrm{cm}^{-1}$).

– The numerical integration of the radiance over the field of view is performed with 5 pencil beams above 30 km tangent height and 7 pencil beams below.

– As in previous retrieval versions, failure of convergence caused by iterations flipping back and forth between two minima of the cost function are avoided by means of an "oscillation detection" approach (see Kiefer et al., 2021, for details).

– Convergence of the retrieval is reached when changes of the solution between successive iterations do not exceed 70% of the noise error at any profile point of the retrieval targets. In previous versions, a less stringent convergence threshold was used (100% of the noise error). The maximum number of retrieval iterations is 15.

## 2.8 Numerical performance and data screening

The entire version 8 data set retrieved from all measurement modes and periods contains about 2.77 million profiles of NO and about 0.17 million temperature profiles. The convergence rate of the retrievals and the median of the reduced $\chi^2$ are listed in Table 3. Compared to version 5, $\chi^2$ values are slightly reduced. Despite the more stringent convergence threshold used in the new version, the convergence rate improved, particularly for UA and NOM FR retrievals where the improvement is significant.





A careful quality screening was applied to the dataset in order to remove corrupted observations. In a first step, retrievals which do not reach convergence after the maximum number of allowed iterations were rejected. The examination of retrievals with bad fitting residuals further indicated that these resulted frequently in strongly oscillating or unphysical results, particularly below 30 km. Therefore, we decided to reject retrievals with a reduced $\chi^2$ value larger than five. In most cases, retrievals with large $\chi^2$ values occur in the presence of clouds below or close to the lowermost tangent height considered in the retrieval,

particularly during late Southern hemispheric winter, in the presence of polar stratospheric clouds. A more conservative cloud filtering might have reduced these problems, however, at the cost of information loss in those retrievals which performed well with the current cloud filtering. Still, some profiles with unreasonably large NO vmrs due to cloud contamination were not identified by the $\chi^2$ threshold. Therefore, we applied a third filter which rejects unphysical NO profiles with vmrs larger than 1.5 ppbv, 11.5 ppbv, and 105 ppbv at 11 km, 20 km and 30 km, respectively. The total number of rejected scans per measurement

mode and period is indicated in Table3. They represent a negligible fraction of the total amount of profiles.

A quality screening was also applied to the NOM dataset of version 5, however, only NO profiles with vmrs larger than 10 ppbv below 17 km were rejected and no $\chi^2$ filtering was applied. Despite the more stringent filtering in version 8, the number of rejected NOM profiles is comparable to that of version 5. No screening, except for convergence, was applied to version 5 MA, UA, and NLC retrievals.

## 350  3   Averaging kernels and spatial resolution

Example averaging kernel (AK) rows of the retrieved NO from NOM RR, MA, and UA measurements, as well as of the retrieved temperature from UA measurements are shown in Fig. 1. These examples correspond to daytime measurements taken in January 2012 at latitudes around 55°N. The AKs from NOM FR measurements and those of NLC measurements (not shown) behave very similar to those from NOM RR and MA measurements, respectively. Note that the AK describes here the retrieval

response in the ln(vmr) domain, that is, its columns $i$ represent the relative retrieval response to a percentage perturbation at altitude $i$ rather than the absolute response to an absolute perturbation of unity amount as in the case of a linear vmr retrieval. The AK rows indicate which altitudes contribute to the retrieval response at a given profile point. Within the vertical scan range of the measurements, the rows of the AKs peak generally at their corresponding altitudes, except for the UA temperature profile below 105 km, where it is fully constrained towards the temperature profile obtained from the 15 $\mu$m measurements. The rows

of the NO AKs corresponding to middle and upper mesospheric altitudes, however, exhibit low peak values and a broad shape with tails that extend to both lower mesospheric and thermospheric altitudes. Profile points with corresponding AK diagonal elements smaller than 0.03 do not contain any significant information from the measurements and should thus be discarded. However, when data is to be averaged (e.g., zonal mean data), we recommend applying this criterium after averaging in order to avoid statistical biases (Funke and von Clarmann, 2012).

Although the vertical scan range of NOM and MA measurements do not cover the thermosphere, there is a pronounced retrieval response to thermospheric NO. This indicates that these measurements contain vertically unresolved information,



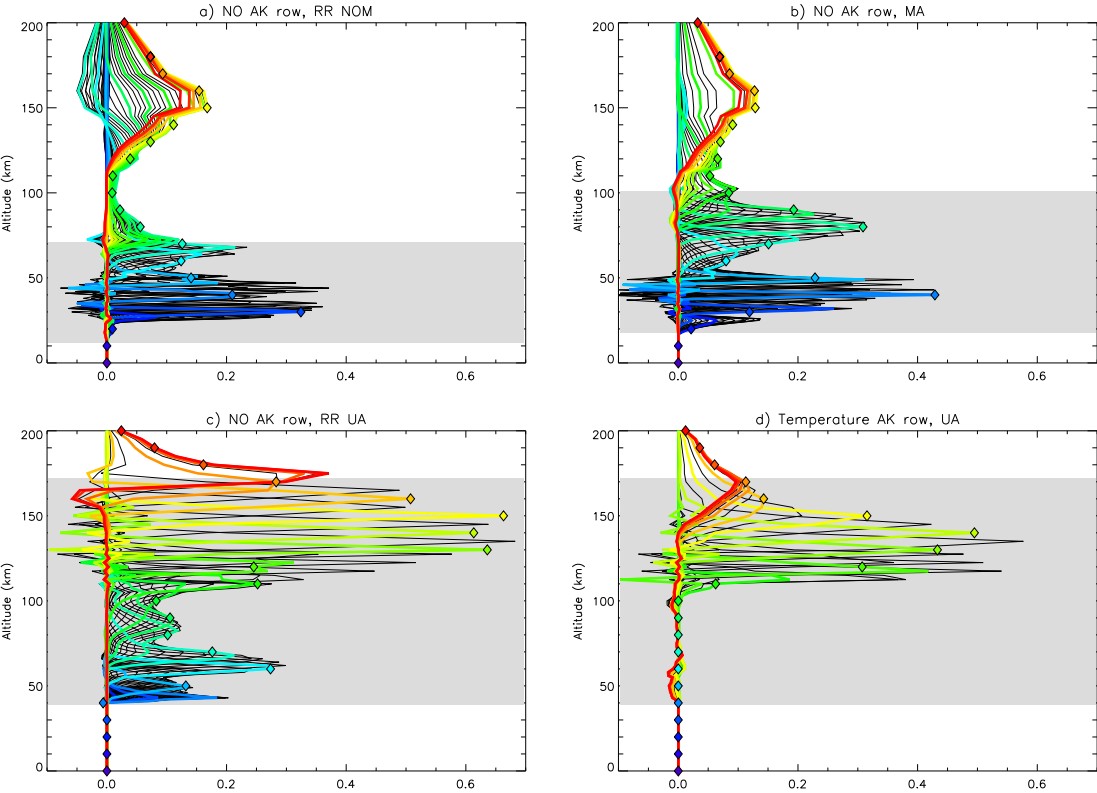

**Figure 1.** Example averaging kernel rows of the retrieved NO from (a) NOM RR measurements, (b) MA measurements, (c) UA measurements, and (d) of the retrieved temperature from UA measurements. Rows corresponding to profile altitudes of 0,10,20,...200 km are highlighted with colored lines. The corresponding averaging kernel diagonal elements are indicated by symbols. The vertical scan range of the respective measurements is indicated by grey-shading. All averaging kernels shown belong to daytime measurements taken in January 2012 at latitudes around 55°N.

e.g., on the thermospheric NO column. However, this information should be exploited with caution because the temperatures at these altitudes are not retrieved from these measurements but rely on the assumed a priori information.

The vertical resolution of the retrieved NO and temperature profiles is estimated as the full width at half maximum of the respective row of the AK matrix. The zonally averaged vertical resolution of NO during Northern winter seasons (DJF) from NOM RR, NOM FR, MA and UA measurements as a function of latitude and altitude is shown in Fig. 2. In these figures, the vertical resolution is displayed only for regions with useful NO information (AK diagonal elements $\geq 0.03$). No significant NO information can be obtained at nighttime conditions below approximately 55 km, as well as in the polar summer and tropical upper mesosphere. However, areas without reliable NO information are significantly smaller in version 8 compared to previous 375 versions. The vertical resolution of NO is 3–6 km in the sunlit stratosphere. In the polar winter mesosphere, vertical resolutions are 10–15 km in MA and UA retrievals and 15–20 km in NOM retrievals. Thermospheric NO from UA measurements





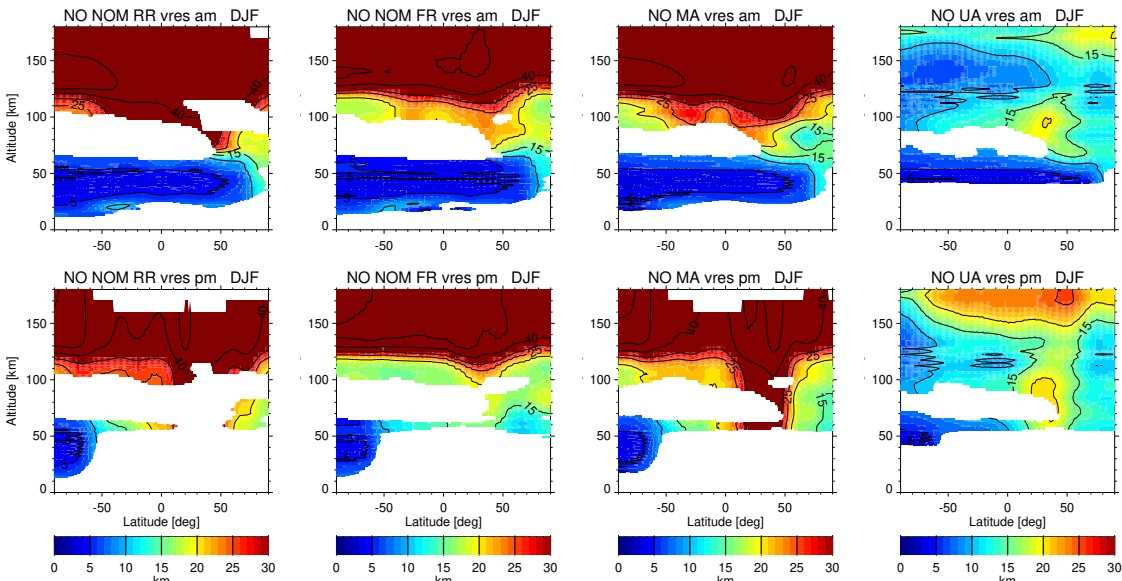

**Figure 2.** Zonal mean vertical resolution (in terms of full width at half maximum of the AK rows) of NO during the Northern winter season (DJF) from NOM RR measurements, NOM FR measurements, MA measurements and UA measurements as a function of latitude and altitude, separated for (top) am and (bottom) pm. White areas indicate data with insignificant information content (AK diagonal < 0.03). Note that NOM FR measurements are averaged over the 2002-2004 period, while measurements from other observation modes and periods are averaged over the 2006–2012 period.

has a vertical resolution of 8–15 km during daytime, being slightly worse above 140 km during nighttime. As expected, the vertical resolution of thermospheric NO in NOM and MA retrievals is poor (20–45 km). The better vertical resolution of FR NOM measurements compared to RR NOM measurements at 100–120 km can be explained by the prevailing solar maximum conditions, with larger NO concentrations in the lower thermosphere, during the FR period (2002–2004) compared to the RR period (2005–2012) which covered mostly solar minimum conditions.

Figure 3 shows the zonal mean vertical resolution of the retrieved temperature in the lower thermosphere from UA measurements for high (averaged over the years 2005, 2006, 2011, 2012) and low solar activity conditions (averaged over the years 2007-2010). The vertical resolution, as well as the profile range with meaningful temperature information, depends strongly on the measured NO radiances which are significantly smaller during solar minimum conditions. They are also smaller during nighttime compared to sunlit conditions. The useful height range of the retrieved temperatures extends from 105–110 km to about 180 km during daytime, but only up to 150–160 km in the nighttime tropical region. During high solar activity periods, vertical resolutions are 5–10 km below approximately 145 km and 10–30 km above. At low solar activity conditions, the vertical resolution is degraded to 10–15 km in the 130–145 km range.

Both, vertical resolution and AK diagonal element profiles are reported for each limb scan along with the retrieved NO and temperature profiles.





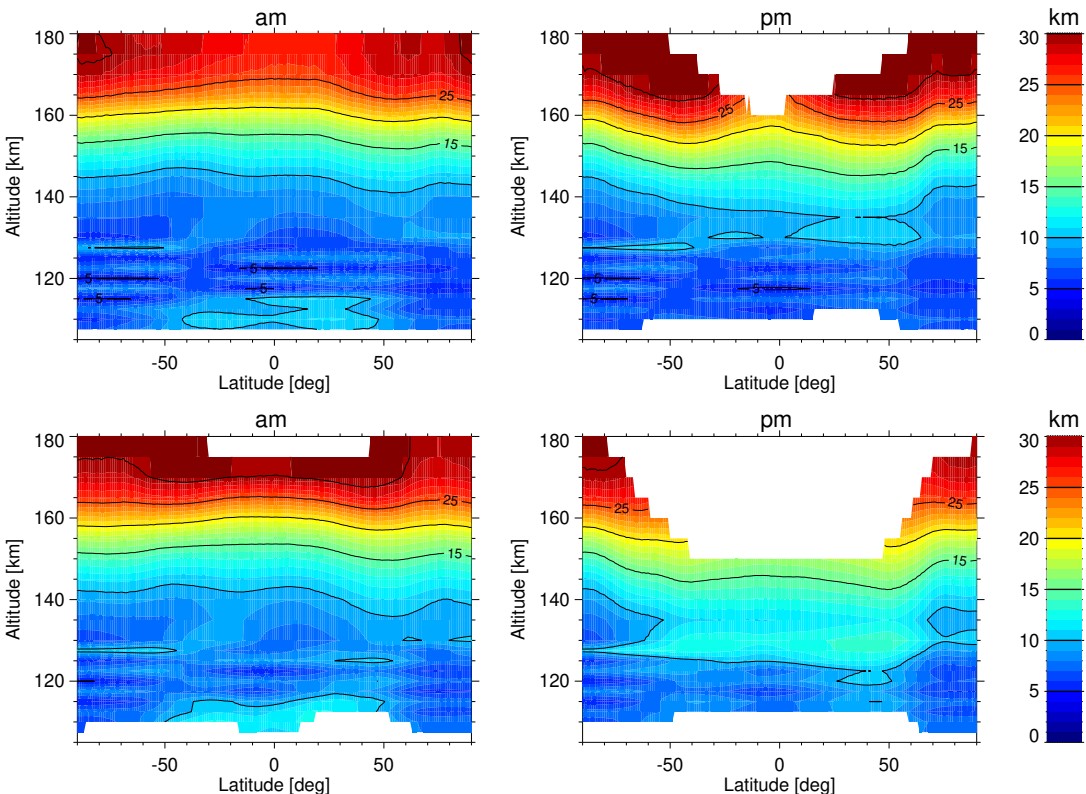

**Figure 3.** Zonal mean vertical resolution (in terms of full width at half maximum of the AK rows) of the retrieved temperature in the lower thermosphere from UA measurements as a function of latitude and altitude, separated for (left) am and (right) pm, as well as for high solar activity conditions (averaged over the years 2005, 2006, 2011, 2012, top) and low solar activity conditions (averaged over the years 2007-2010, bottom).

Horizontal information smearing and information displacement were analyzed using a two-dimensional averaging kernel, as described by von Clarmann et al. (2009). The horizontal smearing $r_{\mathrm{hor},z}$ at altitude $z$ is calculated as

$$r_{\mathrm{hor},z} = \sqrt{2\ln 2} \sum_l \frac{h_{z;l}(l - d_z)^2}{\sum_l h_{z;l}}, \tag{1}$$

where $d_z$ is the information displacement (see below) at altitude $z$, and where $h_{z;l}$ is the element of the horizontal information matrix of altitude $z$ that characterizes the horizontal grid point $l$. The latter is derived from the two-dimensional averaging kernel matrix by vertical integration of the absolute values of its entries. The information displacement is the difference between the sum of the horizontal information matrix-weighted distances from the centre-of-scan geolocation. Negative displacements indicate that most information comes from beyond the nominal geolocation, while positive displacements indicate a source

of information between the nominal geolocation and the satellite. Results are listed in Table 4 for the same example scans as shown in Fig. 1. For all measurement modes, the information smearing is generally larger than the horizontal sampling,





**Table 4.** Horizontal information distribution of NO and temperature at selected altitudes. All distances are given in km.

| Altitude | NO NOM | | NO MA | | NO UA | | Temperature UA | |
|---|---|---|---|---|---|---|---|---|
| | Smearing | Displacement | Smearing | Displacement | Smearing | Displacement | Smearing | Displacement |
| 200 | 2695 | 179 | 2320 | 24 | 1183 | −42 | 1413 | −184 |
| 180 | 2707 | 159 | 2308 | 23 | 935 | −56 | 1362 | −182 |
| 170 | 2707 | 132 | 2282 | 20 | 804 | −61 | 1323 | −180 |
| 160 | 2642 | 92 | 2217 | 12 | 675 | −55 | 1296 | −178 |
| 150 | 2562 | 54 | 2143 | 3 | 700 | −45 | 1164 | −150 |
| 140 | 2431 | 33 | 2018 | −13 | 776 | −36 | 908 | −97 |
| 130 | 2338 | 63 | 1872 | −29 | 741 | −28 | 715 | −62 |
| 120 | 2179 | 32 | 1631 | −42 | 722 | −21 | 669 | −27 |
| 110 | 1950 | −5 | 1398 | −41 | 820 | −3 | 680 | −23 |
| 100 | 1730 | −4 | 1141 | −36 | 961 | 20 | | |
| 90 | 1528 | −14 | 867 | −43 | 929 | 38 | | |
| 80 | 1540 | −15 | 630 | −39 | 849 | 54 | | |
| 70 | 1558 | 7 | 855 | 1 | 694 | 72 | | |
| 60 | 1139 | −29 | 893 | 24 | 715 | 108 | | |
| 50 | 548 | −52 | 461 | 35 | 674 | 145 | | |
| 40 | 473 | −49 | 473 | 51 | 619 | 146 | | |
| 30 | 528 | 0 | 539 | 86 | | | | |
| 20 | 759 | 79 | 705 | 86 | | | | |

defined by the horizontal distance between the center-of-scan geolocations of two subsequent limb scans. This indicates that the horizontal resolution of these measurements is limited by the horizontal smearing and not by the sampling.

## 4 Error budget

The determination of the error budget of the 5.3 $\mu$m retrieval products is based on the MIPAS version 8 error estimation scheme described in detail in von Clarmann et al. (2022). In contrast to earlier error estimations, this novel scheme allows to consider error correlations which may result in error compensation, as well as the error propagation of uncertainties through preceding retrievals.

Only measurement noise error estimates are provided for each profile. Other error components are reported within represen-
tative error budgets for 34 different atmospheric conditions defined in terms of latitude band, season, and illumination. These conditions cover most of the climatologically expected situations. For UA measurements, we further distinguish between high and low solar activity conditions. To each profile of the version 8 dataset, one of these representative error budgets is assigned.



We discuss below the relevant error sources and associated uncertainties which enter the error estimation for the MIPAS 5.3 $\mu$m retrievals. In order to comply with the TUNER (Towards Unified Error Reporting) recommendations (von Clarmann et al., 2020), we report uncertainties of chiefly random nature and systematic nature separately (Sections 4.2 and 4.3, respectively). All reported uncertainties are standard deviations (1$\sigma$).

## 4.1 Error sources

Following the terminology of von Clarmann et al. (2020) we distinguish measurement errors, parameter errors, and model errors. Measurement errors include measurement noise and all uncertainties related to less than perfect knowledge of the instrument state (see Sec 4.1.1). Parameter errors are uncertainties of atmospheric state parameters which are assumed to be sufficiently well known and thus are not treated as unknowns of the retrieval, or those which cannot be retrieved from the measurements (see Sec 4.1.2). Considered model errors include uncertainties in spectroscopic constants and non-LTE kinetic rate constants (see Sec 4.1.3). For the particular case of NO and temperature UA retrievals, we have to consider additionally smoothing error crosstalk (Sec. 4.1.4) as a relevant error source.

### 4.1.1 Measurement errors

The following measurement errors contribute to the overall error budget: measurement noise, gain calibration error, instrument line shape uncertainty, and frequency calibration uncertainties. The propagation of measurement noise was evaluated by means of Eq. 5 of von Clarmann et al. (2022), while the propagation of other measurement errors was estimated on the basis of sensitivity studies for the given atmospheric conditions.

Measurement noise, as estimated from the imaginary part of the spectra, is reported in the level-1b data. In the 5.3 $\mu$m spectral region (MIPAS channel D), the apodized noise is 2–4 nW/(cm$^2$ sr cm$^{-1}$) for RR measurements (a factor of 1.58 larger for FR).

Gain uncertainties were estimated from scaling ratios between overlapping channels deduced from dedicated measurements over the mission which are performed on a daily basis (Kleinert et al., 2018). The gain uncertainties are largely driven by the noise of the respective calibration measurements. In the 5.3 $\mu$m spectral region, they are estimated to be 0.7% during the FR period and 0.5% during the RR period. There is also a systematic component which includes inaccuracies of the calibration blackbody and in the correction of the detector nonlinearity.

For the instrument line shape errors we used the uncertainty estimates for modulation loss through self-apodization. We consider a residual frequency calibration error which accounts for the root mean squares error of the linear regression to the retrieved frequency corrections at different spectral regions. The resulting uncertainty at 5.3 $\mu$m is 0.00029 cm$^{-1}$.

Pointing uncertainties are not explicitly considered since tangent heights are derived together with temperature in the preceding 15 $\mu$m retrievals. Instead, error sources affecting these preceding retrievals are propagated into the 5.3 $\mu$m target space. Since the retrieved radiance offset correction accounts only for spectrally correlated calibration errors (as it is assumed to be constant within each microwindow), offset calibration noise is considered here as an error source.



### 4.1.2 Uncertainties of atmospheric parameters

Relevant atmospheric parameters considered in the radiative transfer and non-LTE calculations are discussed in Sec. 2.3.3. Temperature errors in NOM, MA, and NLC retrievals are implicitly taken into account by propagation of uncertainties affecting the preceding 15 $\mu$m temperature and tangent height retrieval into the 5.3 $\mu$m target space. Uncertainties of the spectrally interfering molecules $O_3$, $H_2O$, $N_2O$ and $CH_4$, as well as their vertical covariances, are estimated from the error covariance matrices of the preceding retrievals. For other interfering species (OCS, acetone, and peroxyacyl nitrate), error covariance matrices are available for NOM measurements from version 5 retrievals. Uncertainty estimates of these species for other measurement modes are based on climatological information obtained from version 5 data. Estimated 1-$\sigma$ uncertainties for $CO_2$, which is taken from WACCM4 simulations, are reported in Table 3 of Kiefer et al. (2021). $CO_2$ uncertainties contribute to the error budget not only because of spectral $CO_2$ interferences but also as entangled error via error propagation through the preceding 15 $\mu$m temperature retrieval (see von Clarmann et al., 2022).

Atmospheric abundance profiles required for the non-LTE model calculations are those of $NO_2$, O, $O_2$, $N_2$, $N(^4S)$, and $N(^2D)$. Uncertainties of $NO_2$ are provided by the total error estimates of the preceding $NO_2$ retrievals. Below 95 km, the uncertainty of daytime O is driven by the error of the $O_3$ retrieval and those introduced by the photochemical model used to derive O from $O_3$. The resulting uncertainty varies within 15–30%, depending on altitude and atmospheric conditions. During night, the O uncertainty is mainly ruled by that of atomic hydrogen, resulting in larger errors (around 100%) below 80 km. At 95–120 km, where O is taken from WACCM, we use the same uncertainties (5–30%) as reported in García-Comas et al. (2022). At higher altitudes, where O is taken from NRLMSIS2.0, we assume uncertainties that corresponds to those at 120 km (5–10%). Uncertainties in $O_2$ and $N_2$ are not considered because their abundances are well known in the altitude range where collisions with these molecules are relevant. No information on $N(^4S)$ and $N(^2D)$ uncertainties is available due to the lack of observations. As in Bermejo-Pantaleón et al. (2011), we assume here an uncertainty range of a factor of two for their abundances.

### 4.1.3 Uncertainties in spectroscopy and kinetic rates

Uncertainties of spectroscopic data for the NO lines included in the microwindows are taken from the error ranges provided with the HITRAN2016 edition. They are considered independently for line intensities, pressure broadening coefficients, and the temperature dependence of the latter.

Regarding kinetic rates needed in the non-LTE model calculations, we consider uncertainties of the five key processes specified below:

- Rates for the deactivation of vibrationally excited NO in collisions with $O_2$ are taken from Wysong (1994). They report an uncertainty of individual rates for different vibrational states in the 10–17% range. Here, we assume an overall uncertainty of 20% for this process.

- We also assume a 20% uncertainty for the multi-quantum relaxation of vibrationally excited NO in collisions with O. This uncertainty corresponds to the reported error of the laboratory measurements for the relaxation rate for the $\nu = 1$ state





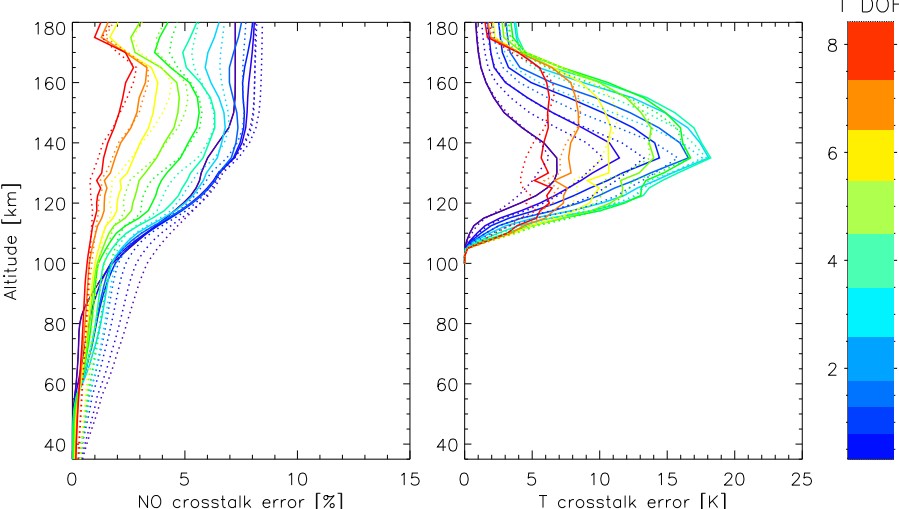

**Figure 4.** Crosstalk of the temperature smoothing error on the NO results (left) and vice versa (right) in 5.3 $\mu$m UA retrievals as a function of the obtained degree of freedom of the retrieved temperature profile. Resultant error contributions are shown separately for daytime (solid) and nighttime (dotted) conditions.

(Hwang et al., 2003). The rotational temperatures, used to describe the rotational nascent distribution of the collisionally excited NO, are 25% lower than the kinetic temperature (Sharma and Duff, 1997), with an assumed uncertainty of 10%.

For the nascent spin temperature, which is set to 200 K in our calculations (Lipson et al., 1994), we assume an uncertainty of 50 K.

– A relevant parameter for 5.3 $\mu$m retrievals is the spin-propensity factor $\beta$ which controls the spin conservation of NO in rotational-translational collisions with $N_2$, $O_2$, and O (see Bermejo-Pantaleón et al., 2011). The considered value of 0.9, however, is well constrained by atmospheric observations. We therefore assume a relatively small uncertainty of

5%. Larger uncertainties are expected for the rotational relaxation rates in NO-O collisions which have not been directly measured so far. Here, we assume an uncertainty of 50%.

– The assumed uncertainty of the production rate of vibrationally excited NO from the $NO_2$ + O reaction is 40%, which corresponds to the reported error of the experimental results from Smith et al. (1992).

– We assume an overall uncertainty of 10% for the production rate of vibrationally excited NO from $NO_2$ photolysis,

which encompasses uncertainties of $NO_2$ cross-sections, quantum yields and albedo effects.

We do not explicitly consider uncertainties related to the production rates of vibrationally excited NO in the reaction of $N(^4S)$ and $N(^2D)$ with molecular oxygen because the overall uncertainty of this chemical excitation process is dominated by the much larger uncertainties in the $N(^4S)$ and $N(^2D)$ abundances (see Sec. 4.1.2).



### 4.1.4  Smoothing error cross talk

The impact of smoothing error crosstalk between NO and temperature in UA version 4 retrievals was extensively investigated by Bermejo-Pantaleón et al. (2011). There, resulting errors were particularly pronounced due to the use of an inappropriate nighttime NO a priori (see Sec. 2.3), although these errors could be mitigated in the context of model comparisons by application of the entire averaging kernels and a priori vectors (covering the complete temperature and NO retrieval space). Since this solution is not always practical, we report here explicitly crosstalk errors for UA retrievals that correspond to the mapping

of NO a priori uncertainties on the retrieved temperature profile and vice versa. These errors are calculated as described in von Clarmann et al. (2022) on basis of estimated a priori covariance matrices for NO and temperature. For the altitude dependence of the covariances, we use a Gaussian dependence on $\Delta z_{i,j}$ with a full width at half maximum of $10\,\text{km}$, roughly representing expected correlation lengths at thermospheric altitudes. Assumed variances correspond to a 1-sigma uncertainty of 50% for NO. For temperature, we assume a linearly increasing uncertainty of $10\,\text{K}$ at $120\,\text{km}$ to $90\,\text{K}$ at $200\,\text{km}$. The magnitude of the

resulting errors in both NO and temperature strongly correlate with the degrees of freedom (DOF) of the retrieved temperature profile which, in turn, exhibit a pronounced dependence on the prevailing solar-geomagnetic conditions and the climatological situation. Temperature DOFs vary from about one to nine, being largest during polar winter and high solar activity, and lowest in the tropics during solar minimum conditions. Figure 4 shows the NO and temperature smoothing crosstalk errors for day and nighttime conditions in dependence of the temperature DOFs.

## 4.2  Random errors

The following error sources are considered to contribute to random errors: measurement noise, residual frequency calibration errors, gain calibration uncertainties, offset calibration noise, smoothing error cross-talk, $NO_2$ uncertainties, and those of the abundances of interfering species. In addition, random variations of retrieval responses to systematic uncertainties (so-called "headache errors", see von Clarmann et al., 2022) also contribute to the random error.

Regarding the uncertainties of interfering species, we also consider uncertainties in $CO_2$ vmrs as a random error source because the impact of spectral interferences in the $5.3\,\mu\text{m}$ microwindows is limited to altitudes below approximately $60\,\text{km}$, where the uncertainty of $CO_2$ from the WACCM model simulations is dominated by small mixing ratio fluctuations related to natural variability. This is not the case for higher altitudes, where systematic model biases could play a major role.

Classifying smoothing error crosstalk in UA retrievals as random is admittedly a simplification, as systematic deficiencies

with respect to the a priori profile shapes will likely remain despite the improvements incorporated in version 8. Nevertheless, the largest smoothing crosstalk error contributions are expected to be caused by random variations of the true profile shape related to wave activity and natural variability.

Figure 5 shows the error budget, including the random error and individual contributors, for three selected atmospheric conditions, namely northern midlatitude summer day, tropical day, and southern polar winter night, for NO retrievals from

the FR and RR NOM mode. The dominating contributor to the random error in MIPAS $5.3\,\mu\text{m}$ retrievals is measurement noise. Zonal mean distributions of the relative measurement noise error in NO retrievals from MA and UA measurements are





**Figure 5.** NO error budget for FR (a, c, e) and RR (b, d, f) NOM data. Additive and multiplicative errors are shown as relative errors of the respective NO vmr profiles. All error estimates are 1-$\sigma$ uncertainties. Error contributions are marked "T+LOS" for the propagated error from the T+LOS retrieval, "noise" for error due to measurement noise, "spectro" for spectroscopic error, "gain" for gain calibration error (see text), "shift" for spectral shift error (see text), "ILS" for instrument line shape error (see text), "offset" for error due to spectral offset (see text), "interf" for the error due to interfering gases, and "NLTE" for non-LTE related errors. (a, b) northern midlatitude summer day, (c, d) tropical day, and (e, f) southern polar winter night.



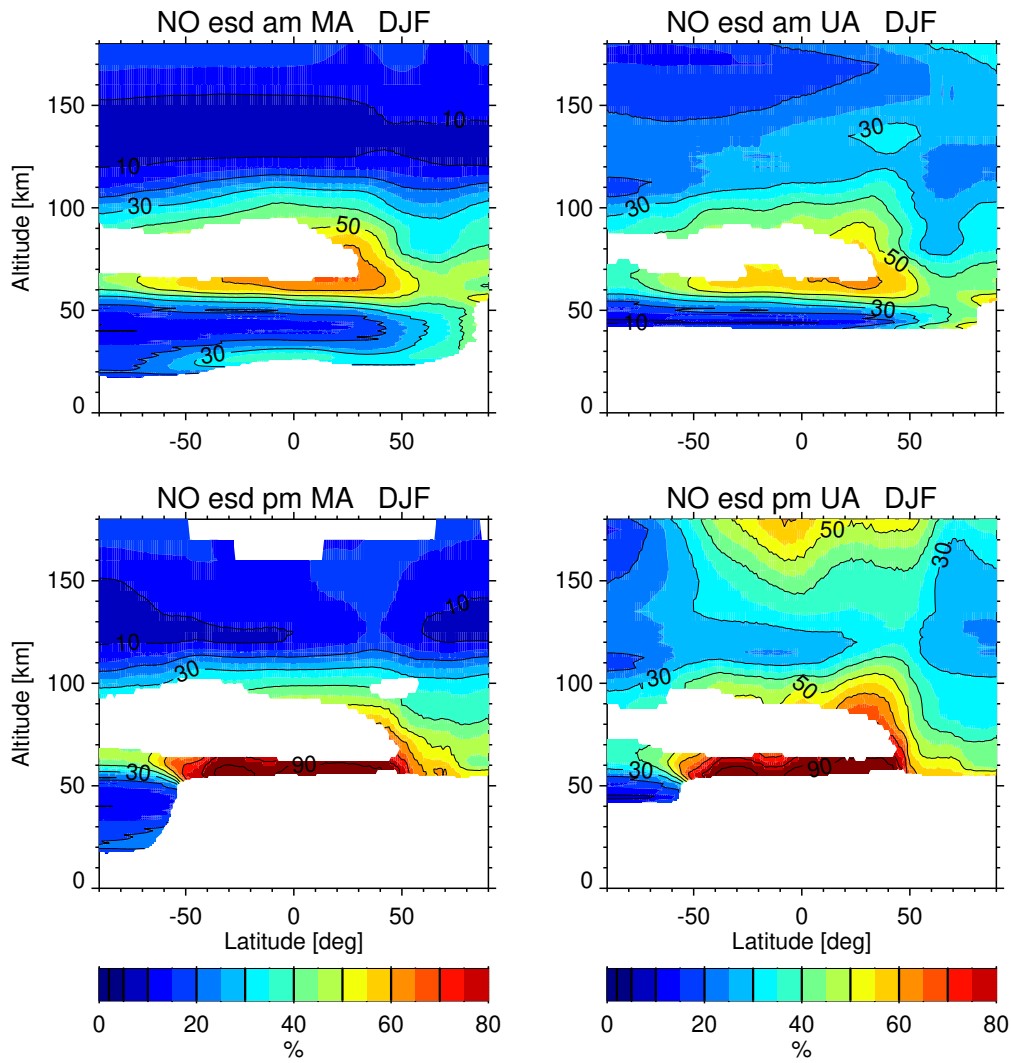

**Figure 6.** The measurement noise component of the NO retrieval error during the Northern winter season (DJF) for (left) MA and (right) UA measurements as function of latitude and altitude, separated for (top) am and (bottom) pm. White areas indicate data with insignificant information content (AK diagonal < 0.03).





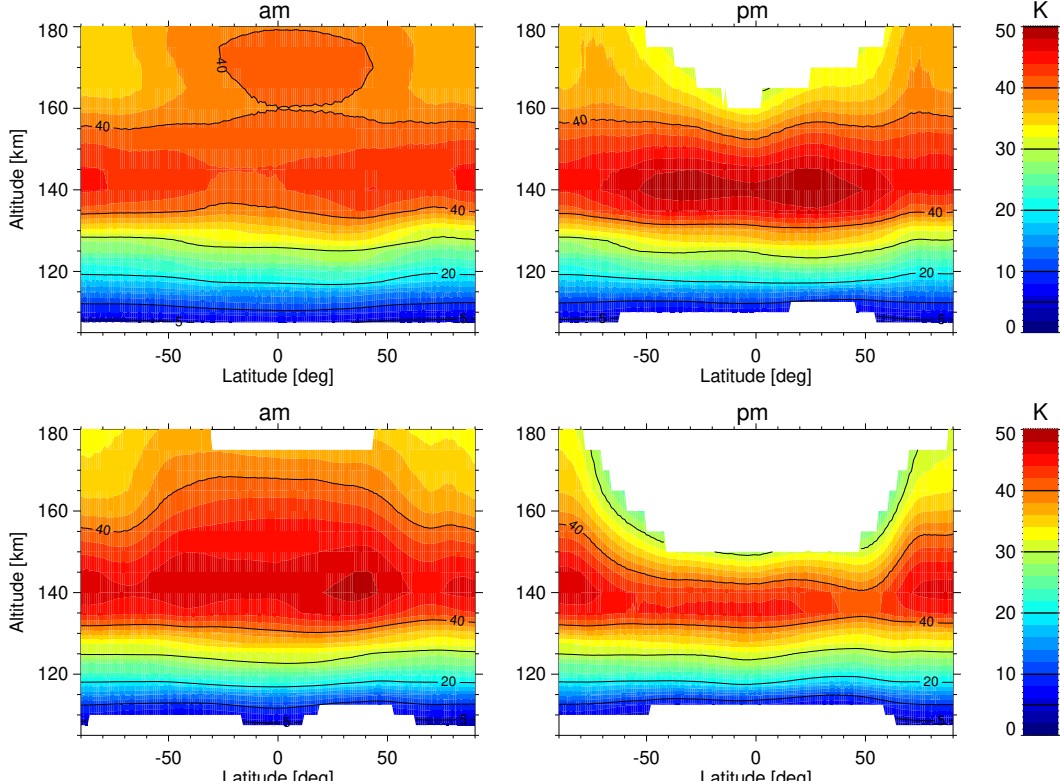

**Figure 7.** The noise component of the UA temperature retrieval error in the lower thermosphere as function of latitude and altitude, separated for (left) am and (right) pm, as well as for high solar activity conditions (averaged over the years 2005, 2006, 2011, 2012, top) and low solar activity conditions (averaged over the years 2007-2010, bottom). White areas indicate data with insignificant information content (AK diagonal < 0.03).

shown in Fig. 6 for Northern winter (DJF) conditions. Figure 7 shows the zonal mean measurement noise error of the retrieved temperature in the lower thermosphere from UA measurements for high solar activity conditions (averaged over the years 2005, 2006, 2011, 2012) and low solar activity conditions (averaged over the years 2007-2010).

Less relevant contributors, with contributions mostly below 10%, are offset and gain calibration uncertainties, and the propagation of the temperature and LOS random errors. Other random error components are typically very small (i.e., less than1%).

For most atmospheric conditions, the total NO random error at stratospheric altitudes ranges from 5% to 40%, being largest in the lower stratosphere and in polar winter. Mesospheric random errors range from 35% to 60% at daytime and can exceed 90% at night around 60 km. Thermospheric random errors from UA retrievals are within 20–50%. The thermospheric temperature

random error ranges from 5 K to 50 K with the largest values around 140 km. Temperature random errors are the smallest during daytime and for high solar activity conditions.



## 4.3  Systematic errors

Sources of systematic errors in MIPAS 5.3 $\mu$m retrieval are uncertainties in spectroscopic data, instrument line shape uncertainties, the persistent component of gain calibration uncertainties, and non-LTE related uncertainties. The latter includes both uncertainties of kinetic rate constants and uncertainties of atmospheric abundances required for the non-LTE modeling, except those of $NO_2$ which are dominated by the random errors of the preceding $NO_2$ retrievals. For other atmospheric abundances relevant for non-LTE, we expect that the systematic uncertainty component, either caused by the impact of uncertain kinetic constants in the photochemical modeling or due to biases of the climatologies used, are likely to be more relevant than the variability component.

Systematic error components of NO NOM retrievals are shown in Fig. 5. For most atmospheric conditions, the systematic NO errors at stratospheric and mesospheric altitudes are around 10%, with the exception of polar winter FR NOM retrieval where they can reach 50%. In UA retrievals, the NO systematic error is slightly larger with 10–30%. The systematic component of the thermospheric temperature error is typically around 10 K. The dominating contributor the the systematic error is non-LTE related uncertainties, followed by spectroscopic uncertainties. Other contributions are typically lower than 1%. The non-LTE error is primarily driven by uncertainties in the multi-quantum relaxation of vibrationally excited NO in collisions with O.

## 5  Nitric oxide results

Seasonal composite zonal mean distributions of the retrieved NO vmr from 10:00 am measurements taken in the NOM RR, MA, and UA observation modes are shown in Fig. 8 for the 20–105 km vertical range. These distributions were obtained by averaging the observations taken in the corresponding seasons during the period 2006–2012. They reflect all expected characteristics of middle atmospheric NO, specifically, the stratospheric peak in the tropics around 40 km, the mesospheric increase towards the lower thermosphere, and seasonal changes in the mesosphere driven by the meridional circulation. NO vmrs are only displayed in areas where data are meaningful (average AK diagonal elements $\geq 0.03$). In all datasets, meaningful mesospheric NO distributions are obtained from all observation modes in the winter hemisphere at least up to 80 km and at latitudes $> 50°$, where NO is enhanced due to descent. Otherwise, meaningful NO can only be obtained up to $\sim$65 km in the lower mesosphere. MA and UA observations have an upper mesospheric detection limit of about 50 ppbv which allows for meaningful data above about 85 km. Figure 9 shows the corresponding 10:00 pm distributions. There, the behavior is similar to the am distributions above 55 km. Below, NO can only be detected in the sunlit region. During nighttime, stratospheric NO abundances are orders of magnitude smaller than the detection limit (about 0.2 ppbv) due to rapid conversion to $NO_2$ by reaction with ozone.

Figure 10 shows the differences of the retrieved NOM vmr distributions from am measurements with respect to the previous retrieval version 5. Differences are only displayed in areas where data of both retrieval versions are meaningful. These areas are considerably reduced in the mesosphere compared to those shown in Fig. 8, demonstrating a significant gain of information in version 8 compared to previous versions at these altitudes. Differences are mostly consistent between FR and RR periods. Overall, the new data version tends to have 5–15% smaller NO abundances at 50–60 km. In the stratosphere, the differences are less





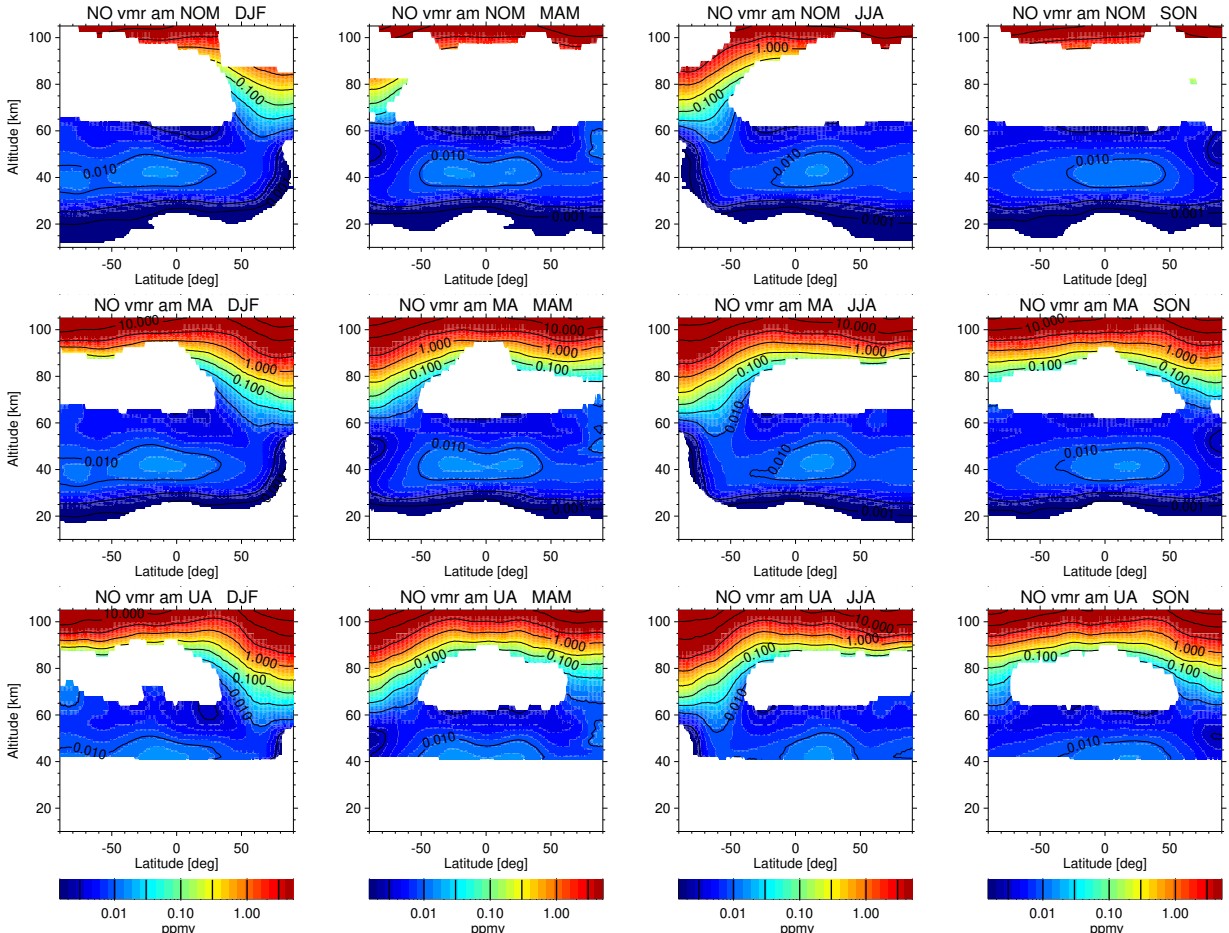

**Figure 8.** Seasonal composite zonal mean distributions of retrieved NO vmrs from (top to bottom) NOM RR, MA, and UA measurements at as function of latitude and altitude for am observations. White areas indicate data with insignificant information content (AK diagonal < 0.03).





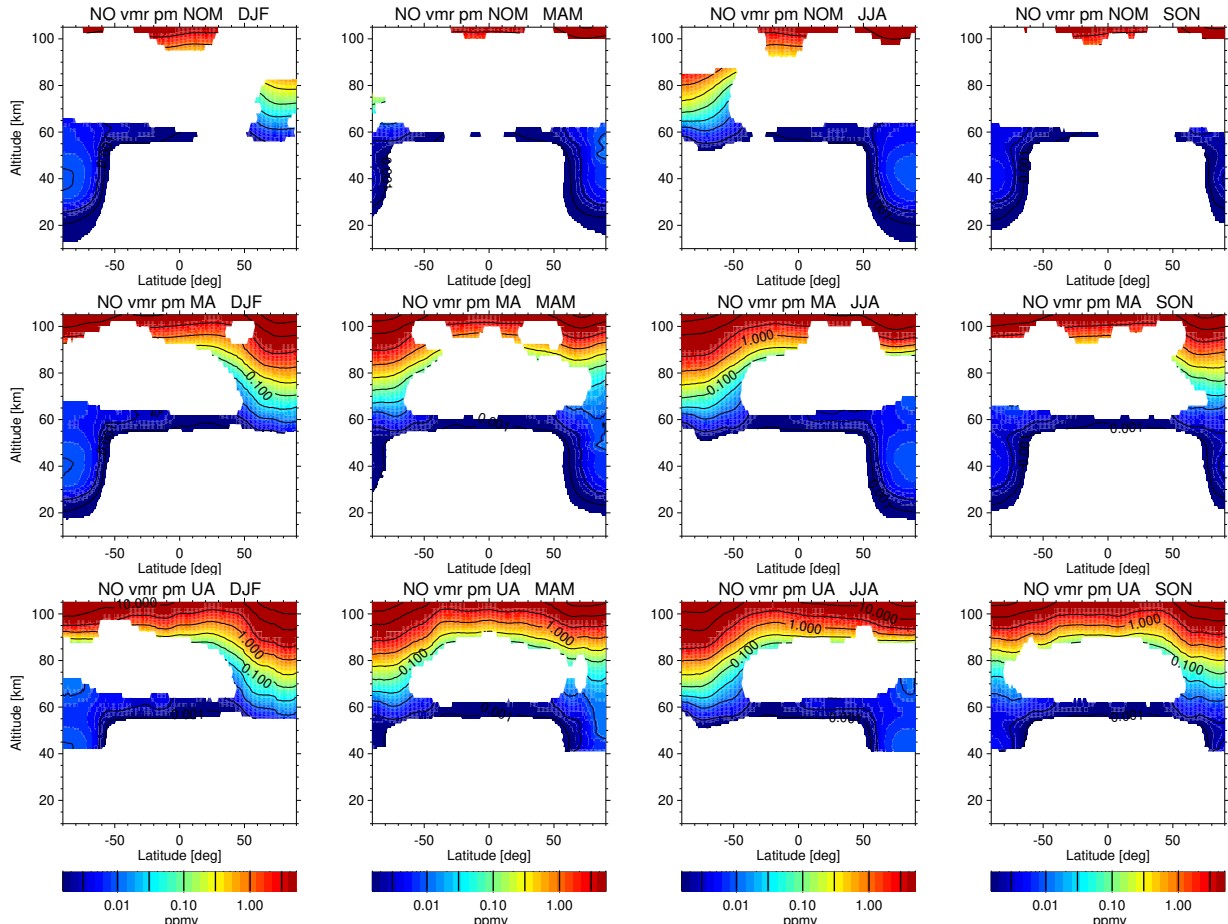

**Figure 9.** As Figure 8, but for pm observations.

systematic, exhibiting positive and negative deviations mostly within 15%. During polar winter, differences are larger, reaching up to 50% in the mesosphere. Except for Northern hemispheric winters in the FR period, larger polar winter abundances are obtained with the new data version.

Figure 11 shows the differences of the retrieved MA and UA vmr distributions from am measurements with respect to the previous retrieval version 5. There, the new UA dataset is compared to version V5r_NO_622 retrievals which are based on

measurements in the 40–105 km height range and do not include temperature as retrieval quantity. Differences of both MA and UA datasets to their predecessor versions are very consistent. Below 65 km the differences are also of similar magnitude (±15%) to those encountered in the NOM comparisons. At 65–100 km, the new datasets exhibit systematically larger NO abundances of up to 50–100%. Above, these differences tend to disappear or even change sign. Bender et al. (2015) compared NO observations from the SCIAMACHY/Envisat, SMR/Odin, and ACE-FTS/SciSat instruments with MIPAS UA version





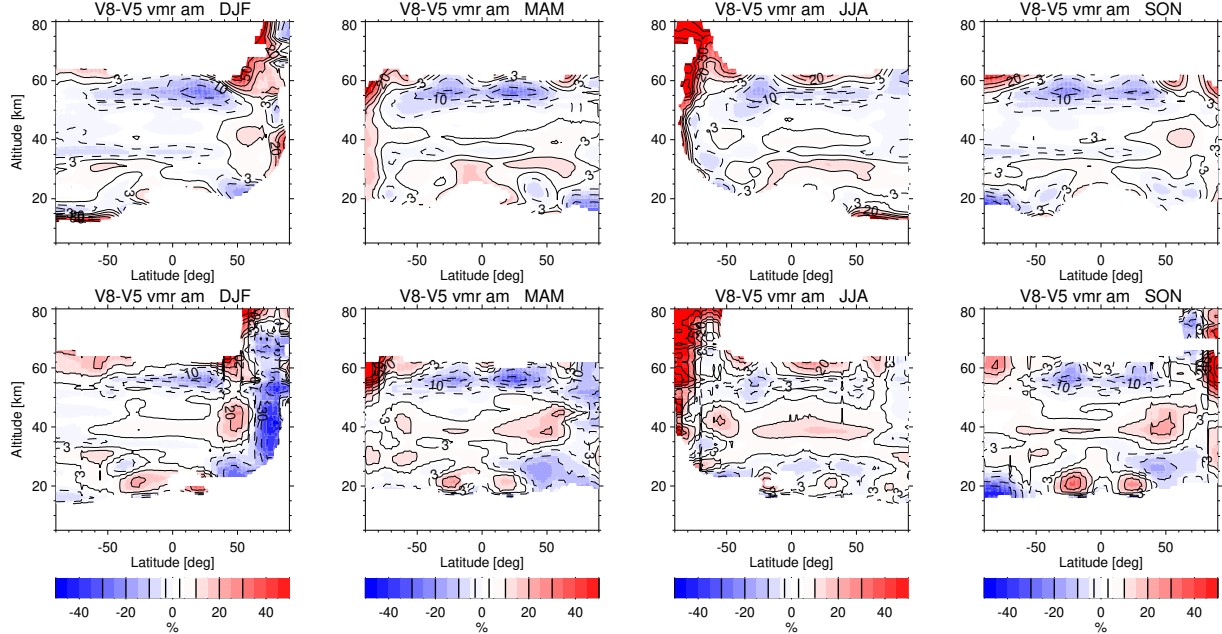

**Figure 10.** Seasonal composite zonal mean distributions of the relative NO vmr differences between version 8 and version 5 as a function of latitude and altitude for (top) NOM RR and (bottom) NOM FR am observations. Specifically, version 5 data used here are V5r_NO_221/220 (RR) and V5h_NO_20 (FR).

5. They found 30–50% lower MIPAS NO concentrations compared to the other instruments at 80–100 km in the NH polar regions. This finding is consistent with the results of Hervig et al. (2019) who compared NO from MIPAS UA version 5 with SOFIE/AIM and ACE-FTS data and found a MIPAS low bias of up to 50% compared to SOFIE and ACE-FTS in the same altitude range. These biases of the version 5 NO data in comparison with correlative measurements, found at 65–100 km, seem to have been considerably reduced or even removed with version 8.

Seasonal composite zonal mean distributions of the NO density from am and pm measurements taken in the UA observation mode are shown in Fig. 12 for the 35–180 km vertical range. We show density instead of vmr in order to better visualize the lower thermospheric NO distribution which is characterized by a density peak around 100 km, being more pronounced in the polar regions due to auroral NO production. The magnitude of this peak is larger, and its vertical position is slightly lower during polar winter. Above 120 km, pm NO densities are significantly smaller than those from am measurements.

Figure 13 shows the relative differences of the retrieved lower thermospheric density distributions from am measurements with respect to those obtained from the previous retrieval version 5, as well as to those from the original version 4 discussed in Bermejo-Pantaleón et al. (2011). Specifically, the new UA dataset is compared to version V5r_NOwT_622 and V4r_NOwT_611 retrievals which are based on measurements in the 90–172 km height range and include temperature as retrieval quantity. At 105–120 km, the NO densities of the new version are 20–50% smaller than those of the previous versions.





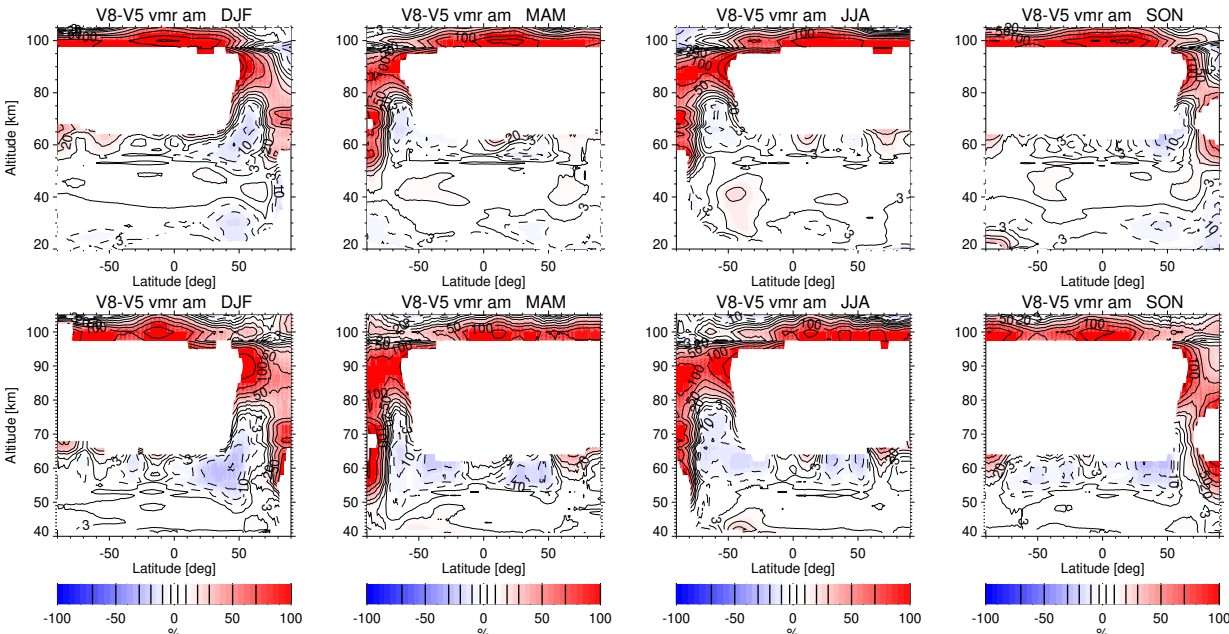

**Figure 11.** Seasonal composite zonal mean distributions of the relative NO vmr differences between am observations of version 8 and version 5 as a function of latitude and altitude for (top) MA and (bottom) UA. Specifically, version 5 data used here are V5r_NO_521 (MA) and V5r_NO_622 (UA).

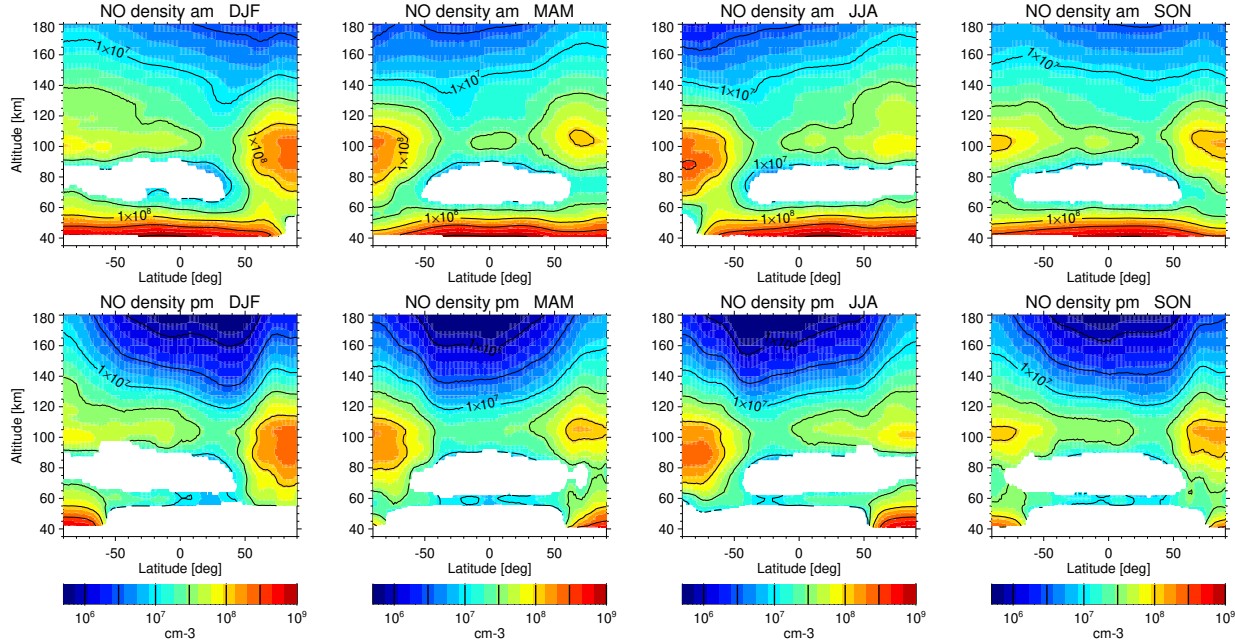

**Figure 12.** Seasonal composite zonal mean distributions of retrieved NO densities from UA measurements as a function of latitude and altitude, separated for (top) am and (bottom) pm. White areas indicate data with insignificant information content (AK diagonal < 0.03).





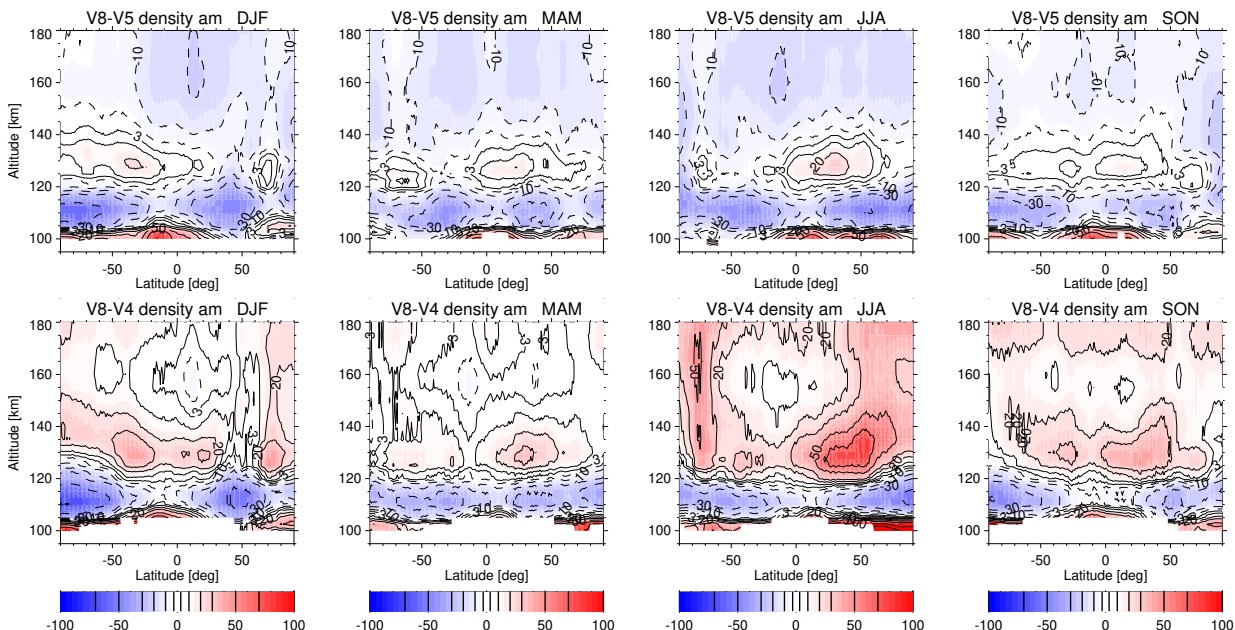

**Figure 13.** Seasonal composite zonal mean distributions of the relative NO density differences between am measurements of different UA data versions as a function of latitude and altitude: (top) V8R_NOwT_662 versus V5r_NOwT_622 and (bottom) V8r_NOwT_662 versus V4R_NOwT_611.

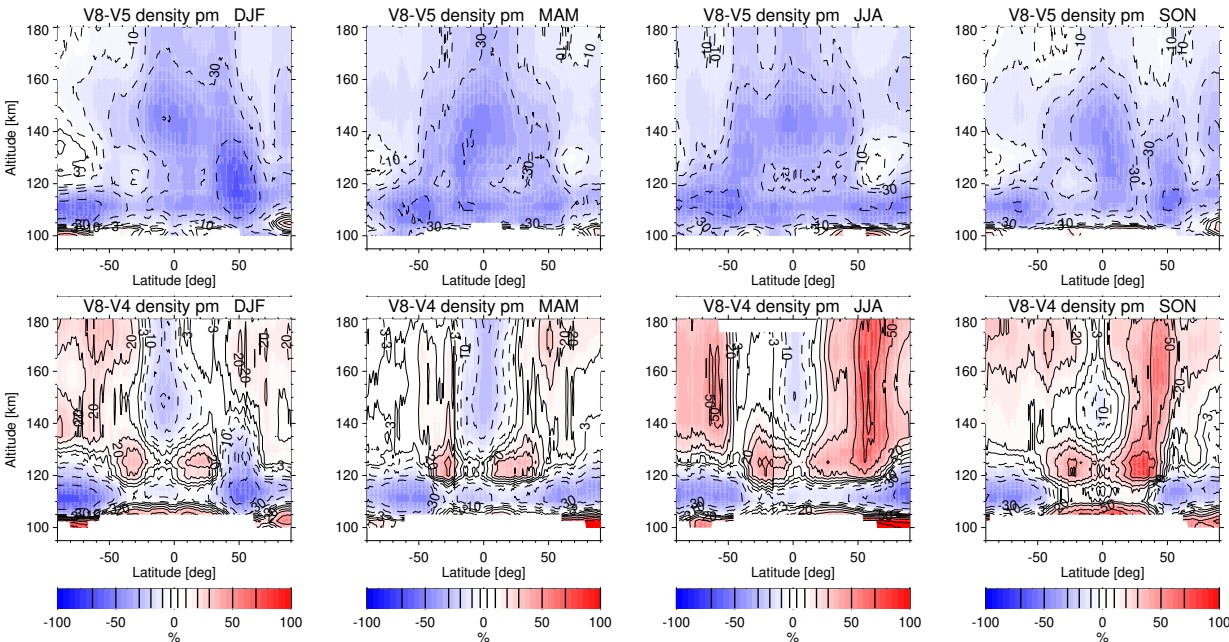

**Figure 14.** As Fig. 13, but for pm observations.





At these altitudes, the comparisons of MIPAS NO densities from version 5 retrievals with correlative measurements from ACE-FTS, SOFIE, and SCIAMACHY, conducted in the validation studies of Hervig et al. (2019) and Bender et al. (2015), indicated a high bias of MIPAS NO. Thus we conclude that the new NO data is likely in better agreement with NO observations from other satellite instruments both in the upper mesosphere, where the MIPAS NO from version 5 was low-biased, and in the lower thermosphere below 120 km, where NO from the previous version was biased high. At altitudes between 120 km and 140 km,

the new version agrees with version 5 within -10% and +20%, while 20–50% larger NO densities are obtained compared to version 4. At higher altitudes, the NO densities of the new version are about 10–15% smaller than version 5 and up to 30% larger than version 4.

    Figure 14 shows the lower thermospheric density differences between the new version and previous versions obtained from pm measurements. These differences are similar to those obtained from the am measurements below 120 km. At higher alti-

tudes, NO densities of the new version are systematically lower than in version 5 by 10–30%. Compared to version 4, these differences are less systematic and are in the tendency of positive sign.

    The consistency of NO data obtained from measurements taken in different observation modes is relevant in the context of data merging in order to fill up temporal gaps caused by the switching between the different modes. Figure 15 shows time series of daily zonal mean NO vmrs retrieved from NOM, MA and UA measurements during 2008–2012 at various altitudes.

The zonal mean data corresponds to the 70–80°S latitude band, which is an important region for the study of NO polar winter descent into the stratosphere. The good consistency between NOM, MA and UA data is particularly evident below 50 km where the day-to-day variability is small. At higher altitudes, the data points are more dispersed, largely due to the impact of dynamical and geomagnetic variability. However, no obvious biases between the data of different observation modes can be identified. NOM observations are mostly consistent with MA and UA observations even at upper mesospheric and lower

thermospheric altitudes, which are well above the scan range of the NOM measurements. Globally, NOM, MA and UA data in the stratosphere agree within 5–10%, whereby NOM observations are on average slightly lower than MA and UA observations.

## 6   Lower thermospheric temperature results

Seasonal composite zonal mean distributions of the retrieved lower thermospheric temperatures from am and pm observations of the UA mode are shown in Fig. 16. These distributions were obtained by averaging the observations taken in the corre-

sponding seasons during the period 2006–2012. The retrieved temperatures increase with altitude, from 200–300 K at 110 km to values of 700–800 K around 170 km, with largest temperatures in the sub-polar summer region. As for the NO results, temperatures are only displayed in areas where data are meaningful (average AK diagonal elements $\geq 0.03$). This is the case above 105–110 km up to altitudes around 175 km for am observations and up to around 160 km for pm observations.

    Figure 17 shows the Northern winter (DJF) differences of the retrieved lower thermospheric temperature distributions

with respect to those obtained from the previous retrieval version 5, as well as to those from the original version 4 discussed in Bermejo-Pantaleón et al. (2011). Specifically, the new UA dataset is compared to version V5r_NOwT_622 and V4r_NOwT_611 retrievals. Below 120 km, temperatures of the new version are significantly warmer than those of the pre-







**Figure 15.** Time series of daily zonal mean NO vmrs at various altitudes (as indicated in the plot titles) from NOM RR (grey: am, black: pm), MA (orange: am, light blue: pm), and UA (red: am, dark blue: pm) measurements during 2008–2012 at $70°$S–$80°$S latitude.

vious versions by 20–60 K. This difference is caused by the changes introduced to the a priori profile which is taken from the preceding 15 $\mu$m temperature retrieval up to about 115 km, instead of using NRLMSISE-00 which is significantly colder

in that region. Above 120 km, the new temperatures are generally colder by 5–30 K than those of version 5, except for the tropics above 140 km where the new version is warmer by 10–30 K. Compared to the original version 4 retrievals, we obtain significantly warmer temperatures by up to 70 K in the entire lower thermosphere, except for pm observations around 125 km and latitudes $< 50°$, which are colder in the new version by 20–30 K.

The temperature differences between am and pm observations, taken at fixed local times with 12 hours difference, allow

to assess the self-consistency of the new dataset. These differences are largely driven by the migrating diurnal tide (DW1) which exhibits a characteristic pattern in the zonal temperature distribution. Below approximately 120 km, DW1 is dominated by upwards propagation of tidal waves that are generated by radiative heating in the lower atmosphere. In this region, the pattern of the DW1 tide is characterized by an amplitude maximum in the tropics, a vertical wavelength of about 20 km, and





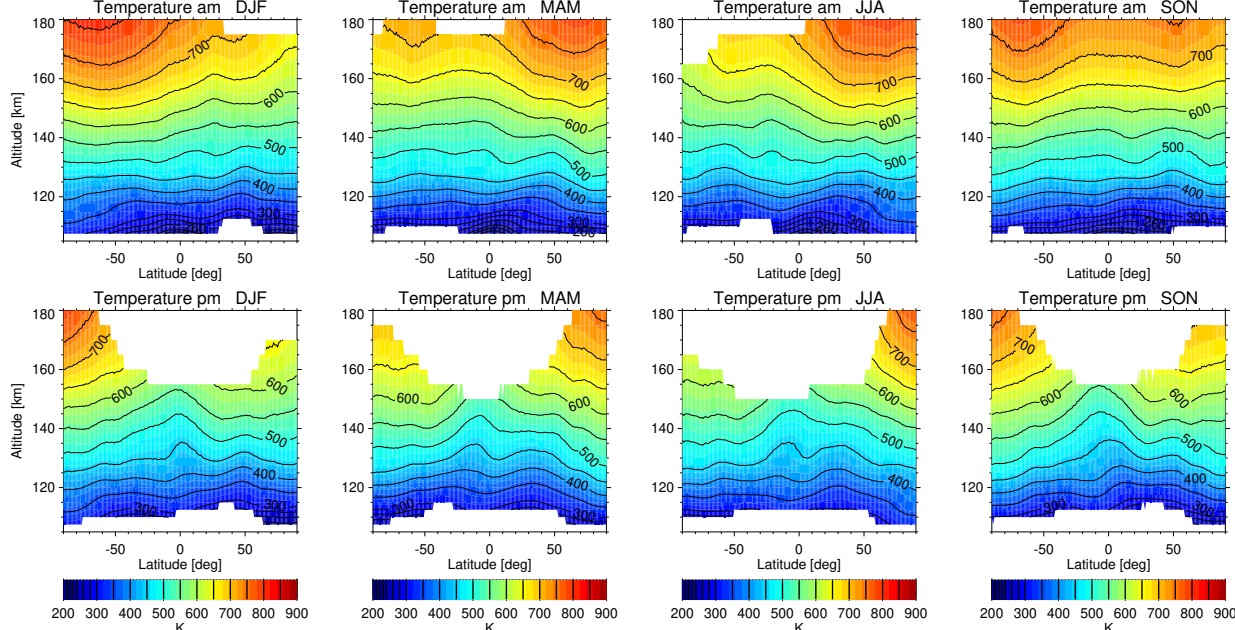

**Figure 16.** Seasonal composite zonal mean distributions of retrieved temperatures from UA observations as function of latitude and altitude, separated for am (top) and pm (bottom). White areas indicate data with insignificant information content (AK diagonal < 0.03).

a phase change of 180° between the tropics and the extra-tropics. The tidal amplitude is small at latitudes polewards of 50°.

At altitudes above 120 km, the dominating pattern is caused by the in-situ tide which is generated by EUV heating on the dayside. The thermospheric migrating tide is characterized by vertically increasing amplitudes which maximize close to the subsolar point in the meridional direction. Figure 18 shows the seasonal composites of zonal mean temperature differences between am and pm observations of the new version 8 data and those obtained from the previous version 5. No filtering with an AK diagonal threshold was applied here, that is, differences are also displayed at altitudes below 107 km where

the new temperatures are entirely constrained by the version 8 15 $\mu$m results. The latter have been compared to correlative measurements from SABER/TIMED and show a good agreement in the entire MLT region with differences typically smaller than 5–10 K (García-Comas et al., 2022). Version 5 retrievals depend also strongly on the a priori information below 105 km, however, in this case, it is taken from the 15 $\mu$m results only up to about 100 km while above it comes from NRLMSISE-00 (see Sec. 2.3.2).

The am–pm temperature differences of the new version show a clear tidal DW1 pattern, with alternating am and pm temperature enhancements in both vertical and latitudinal directions, up to about 120 km. Above, these differences are mainly positive, with a vertically increasing amplitude and a latitudinal variation consistent with the in situ generated diurnal tide. The vertical structure of the am–pm differences does not show a discontinuity in the 105–115 km region where the transition between 15 $\mu$m and 5.3 $\mu$m temperature information takes place. This suggests that the temperature results of both retrievals

are largely consistent despite of using spectral information from different emission sources. The am–pm differences of the





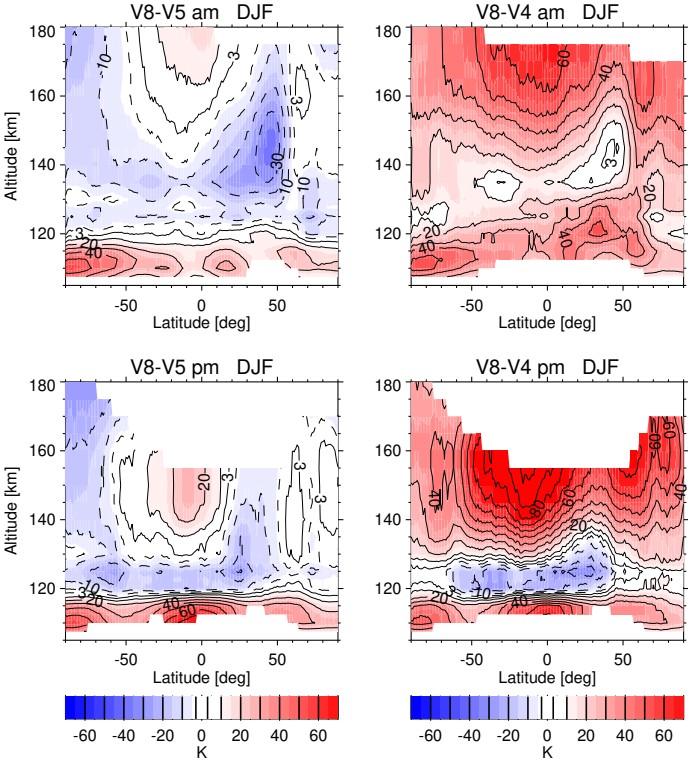

**Figure 17.** Northern winter (DJF) composite zonal mean distributions of the temperature differences between UA data version 8 with respect to (left) version 5 (V5r_TwNO_622) and (right) version 4 (V4r_TwNO_611) as function of latitude and altitude, separated for am (top) and pm (bottom).

version 5 temperatures are similar to those of the new retrieval version below 100 km and above 125 km. In the 100–125 km region, however, they show an entirely different pattern with positive differences around 110 km and negative differences around 120 km. They further do not exhibit the expected latitudinal structure of the DW1 tide, there. It is evident that the version 8 am–pm differences encountered in that region are more consistent with the expected DW1 temperature structure from tidal theory.

### 6.1 Comparison to NRLMSIS2.0

Thermospheric temperature observations in the 110-170 km region are still sparse. A widely used reference for the thermal structure in that region is the NRLMSISE-00 empirical model (Picone et al., 2002) which has been recently updated to NRLMSIS2.0 (Emmert et al., 2021), however, without introducing significant changes to the temperature distribution above 100 km. Lower thermospheric temperatures of NRLMSIS rely largely on Millstone Hill Incoherent Scatter Radar observations taken during the 80s and 90s. Emmert et al. (2021) compared NRLMSIS2.0 temperatures to MIPAS version 5 data. They encountered a 30–50 K high bias of the MSIS temperatures compared to MIPAS nighttime data above 120 km, while the agreement between MSIS and MIPAS daytime temperatures was found to be within 10–20 K.



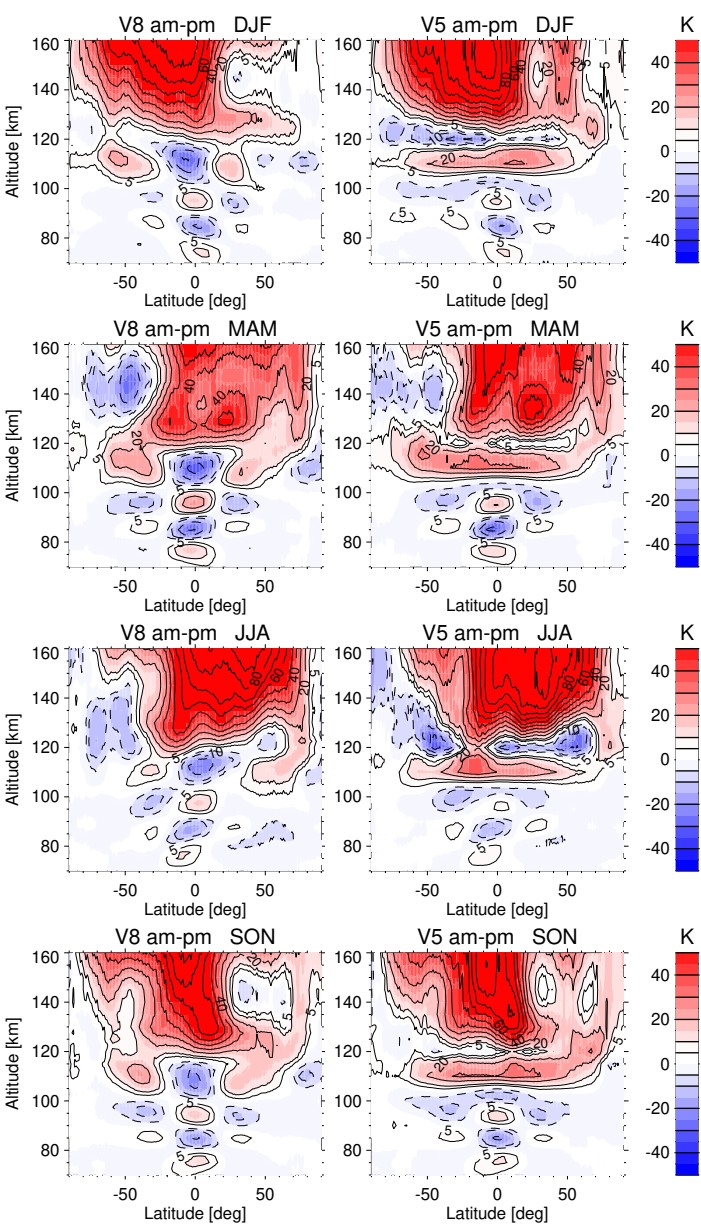

**Figure 18.** Seasonal composites of zonal mean temperature differences between am and pm observations of (left) version 8 and (right) version 5.





**Figure 19.** Seasonal mean temperature differences between MIPAS and NLRMSIS2.0 temperatures for various latitude bands. MIPAS version 8–NRLMSIS2.0 is shown for am (red) and pm (dark blue) observations. MIPAS version 5–NRLMSIS2.0 is shown for am (orange) and pm (light blue) for comparison. The grey-shaded areas indicate the vertical range where MIPAS temperatures are from the 15 μm retrievals.



Figure 19 shows the seasonal mean temperature differences between MIPAS version 8 and NLRMSIS2.0 temperatures for various latitude bands. The differences between MIPAS version 5 and NLRMSIS2.0 are also shown for comparison. Again, no filtering with an AK diagonal threshold was applied here to the MIPAS data in order to consider the combined temperature profile from the $15\,\mu$m and $5.3\,\mu$m retrievals. MSIS temperatures have been calculated for the locations and times of the MIPAS observations. NRLMSIS2.0 agrees very well with the MIPAS temperatures (mostly within $10\,$K) up to $100\,$km. This is the region where NRLMSIS2.0 has substantially improved with respect to the predecessor version by assimilation of contemporary satellite temperature observations. In the $100$–$130\,$km region, MIPAS is systematically warmer than NRLMSIS2.0 at all seasons and latitudes. Largest positive differences up to $80\,$K are found for pm observations in the tropics. There, the altitude of the maximum differences for am observations is located slightly higher (around $120\,$km) compared to those of the pm observations (around $115\,$km). Above $130\,$km, MIPAS is on average colder than NRLMSIS2.0, in particular for pm observations. The largest differences are found at high latitudes during the winter seasons. The differences of the new MIPAS temperature dataset with NRLMSIS2.0 are qualitatively similar to those found when comparing the version 5 temperature data. In the $105$–$120\,$km region, however, the version 5 – NRLMSIS differences often show a double peak structure which is caused by the impact of the MSIS a priori used there. The version 8–NRLMSIS differences, in contrast, show a consistent behavior between the vertical range dominated by the information from the $15\,\mu$m retrievals (below $\sim 115$ km) and the vertical range above, dominated by the information from the $5.3\,\mu$m retrievals.

## 7 Conclusions

MIPAS IMK/IAA nitric oxide and lower thermospheric temperature data presented in this work are based on the most recent version 8 level-1b spectra and were processed using a retrieval approach improved over previous versions with respect to the quality of the temperature data used in NOM, MA and NLC retrievals, the choice and construction of the a priori and atmospheric parameter profiles, the treatment of horizontal inhomogeneities, the treatment of the radiance offset correction, and the selection of optimized numerical settings.

A TUNER-compliant error assessment has been performed. Nitric oxide vmr retrieval errors at stratospheric altitudes range from 5% to 40%, being largest in the lower stratosphere and in polar winter. Mesospheric retrieval errors range from 40% to 70% at daytime and can exceed 90% at night around $60\,$km. Thermospheric errors from UA retrievals are within $20$–$50\%$. The thermospheric temperature error ranges from $5\,$K to $50\,$K with the largest values around $140\,$km. The total error of both NO vmr and temperature is dominated by random errors due to measurement noise. Systematic NO vmr errors are typically around 10%, with the exception of polar winter FR NOM retrieval where they can reach 50%. In UA retrievals, the NO systematic error is with $10$–$30\%$ slightly larger. The systematic component of the thermospheric temperature error is typically around $10\,$K. The dominating contributor to the systematic error is non-LTE related uncertainties.

There is a significant gain of information in version 8 NO retrievals compared to previous versions at mesospheric altitudes. This is attributed to the use of a more reliable a priori information with larger abundances at these altitudes. Overall, the new data version tends to have 5–15% smaller NO abundances at 50–60 km, while differences are less pronounced below. In the





mesosphere, biases of the version 5 NO data in comparison with correlative measurements, found at 65–100 km, seem to have been considerably reduced or even removed in the new version. The new NO data is likely also in better agreement with NO observations from other satellite instruments in the upper mesosphere, where the MIPAS NO from version 5 was low-biased, and in the lower thermosphere below 120 km, where a positive bias was found previously.

The consistency of NO data from different observation modes has been assessed. Globally, NOM, MA, and UA data in the stratosphere agree within 5–10%, whereby NOM observations are on average slightly lower than MA and UA observations.

Regarding thermospheric temperatures, version 8 is generally colder by 5–30 K than version 5, except for the tropics above 140 km where the new version is warmer by 10–30 K. Further, version 8 am–pm differences in the 100–120 km region are more consistent with the expected DW1 temperature structure from tidal theory, compared to previous versions. MIPAS version 8 temperatures are systematically warmer than results from the empirical NLRMSIS2.0 model by 30 K to 80 K in the 100–120 km region at all seasons and latitudes. Above 130 km, MIPAS is, on average, colder than MSIS, in particular for pm observations. The largest differences are found at high latitudes during the winter seasons.

*Data availability.* The MIPAS data can be obtained from the IMK/IAA MIPAS data server under https://www.imk-asf.kit.edu/english/308.php.

*Acknowledgements.* Spectra used for this work were provided by the European Space Agency. This work was partly funded by MCIU under project PID2019-110689RB-I00/AEI/10.13039/501100011033. The KIT team was supported by DLR under contract number 50EE1547 (SEREMISA). The computations were done in the frame of a Bundesprojekt (grant MIPAS_V7) on the Cray XC40 "Hazel Hen" of the High-Performance Computing Center Stuttgart (HLRS) of the University of Stuttgart. WACCM simulations used for a priori generation are based upon work supported by the National Center for Atmospheric Research (NCAR), which is a major facility sponsored by the National Science Foundation under Cooperative Agreement No. 1852977.

*Author contributions.* BF developed the retrieval setup, performed the data analysis, and wrote the manuscript. UG provided and maintained the retrieval software. SK and AL run the retrievals. MK, NG, and UG provided the error estimation software and error estimates. TvC cared about TUNER compliance of error estimates. All authors participated in the development of the retrieval setup, contributed to the discussions, and provided text and comments.

*Competing interests.* The authors declare that they have no conflict of interest.





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
