# Peer review of "MIPAS IMK/IAA version 8 retrieval of nitric oxide and lower thermospheric temperature"

_Atmospheric Measurement Techniques, 2022_

## Author Comment (AC2)

Manuscript prepared for Atmos. Meas. Tech.
with version 2015/04/24 7.83 Copernicus papers of the LaTeX class copernicus.cls.
Date: 19 January 2023

*Supplement to*

**MIPAS IMK/IAA version 8 retrieval of nitric oxide and lower thermospheric temperature**

**Bernd Funke et al.**

*Correspondence to:* Bernd Funke (bernd@iaa.es)

This document serves as reference for the definitions of the representative atmospheres used for the calculation of nitric oxide error budgets, as listed in Tab. S1, and as collection of the respective error budgets for V8R_NO_561 (MA) data, which are listed in tables S2–S35 and depicted in figures S1–S34.

The errors are presented as relative errors in percent, regardless of whether they are additive or multiplicative errors. They were calculated with respect to the average nitric oxide profile that was calculated from the single geolocations which contribute to the respective representative atmospheres.

**Table S1.** Labels and definitions of the representative atmospheric conditions which were used to calculate the error budget for FR and RR data.

| representative atmosphere label | month(s) used | latitude range | solar zenith angle range |
| --- | --- | --- | --- |
| Northern polar winter day | Jan, Feb | $65°N – 90°N$ | $< 90°$ |
| Northern polar winter night | Jan, Feb | $65°N – 90°N$ | $> 98°$ |
| Northern polar spring day | Apr | $65°N – 90°N$ | $< 90°$ |
| Northern polar spring night | Apr | $65°N – 90°N$ | $> 98°$ |
| Northern polar summer day | Jul, Aug | $65°N – 90°N$ | $< 90°$ |
| Northern polar summer night | Jul, Aug | $65°N – 90°N$ | $> 98°$ |
| Northern polar autumn day | Oct | $65°N – 90°N$ | $< 90°$ |
| Northern polar autumn night | Oct | $65°N – 90°N$ | $> 98°$ |
| Northern midlatitude winter day | Jan, Feb | $40°N – 60°N$ | $< 90°$ |
| Northern midlatitude winter night | Jan, Feb | $40°N – 60°N$ | $> 98°$ |
| Northern midlatitude spring day | Apr | $40°N – 60°N$ | $< 90°$ |
| Northern midlatitude spring night | Apr | $40°N – 60°N$ | $> 98°$ |
| Northern midlatitude summer day | Jul, Aug | $40°N – 60°N$ | $< 90°$ |
| Northern midlatitude summer night | Jul, Aug | $40°N – 60°N$ | $> 98°$ |
| Northern midlatitude autumn day | Oct | $40°N – 60°N$ | $< 90°$ |
| Northern midlatitude autumn night | Oct | $40°N – 60°N$ | $> 98°$ |
| Tropics day | Apr | $20°S – 20°N$ | $< 90°$ |
| Tropics night | Apr | $20°S – 20°N$ | $> 98°$ |
| Southern midlatitude winter day | Jul, Aug | $40°S – 60°S$ | $< 90°$ |
| Southern midlatitude winter night | Jul, Aug | $40°S – 60°S$ | $> 98°$ |
| Southern midlatitude spring day | Oct | $40°S – 60°S$ | $< 90°$ |
| Southern midlatitude spring night | Oct | $40°S – 60°S$ | $> 98°$ |
| Southern midlatitude summer day | Jan, Feb | $40°S – 60°S$ | $< 90°$ |
| Southern midlatitude summer night | Jan, Feb | $40°S – 60°S$ | $> 98°$ |
| Southern midlatitude autumn day | Apr | $40°S – 60°S$ | $< 90°$ |
| Southern midlatitude autumn night | Apr | $40°S – 60°S$ | $> 98°$ |
| Southern polar winter day | Jul, Aug | $65°S – 90°S$ | $< 90°$ |
| Southern polar winter night | Jul, Aug | $65°S – 90°S$ | $> 98°$ |
| Southern polar spring day | Oct | $65°S – 90°S$ | $< 90°$ |
| Southern polar spring night | Oct | $65°S – 90°S$ | $> 98°$ |
| Southern polar summer day | Jan, Feb | $65°S – 90°S$ | $< 90°$ |
| Southern polar summer night | Jan, Feb | $65°S – 90°S$ | $> 98°$ |
| Southern polar autumn day | Apr | $65°S – 90°S$ | $< 90°$ |
| Southern polar autumn night | Apr | $65°S – 90°S$ | $> 98°$ |

**Table S2.** Nitric oxide error budget for Northern polar winter day. All uncertainties are $1\sigma$.

| altitude (km) | mean target (ppmv) | NLTE (%) | interf (%) | ILS (%) | shift (%) | offset (%) | gain (%) | spectro (%) | T+LOS (%) | noise (%) | random (%) | syst (%) | total (%) |
|---|---|---|---|---|---|---|---|---|---|---|---|---|---|
| 30 | 0.00 | 9.85 | 0.52 | 0.65 | 0.37 | 1.37 | 2.16 | 12.39 | 2.15 | 40.62 | 41.14 | 14.83 | 43.73 |
| 40 | 0.01 | 13.46 | 0.18 | 2.16 | 0.31 | 1.43 | 2.68 | 12.96 | 2.83 | 44.07 | 44.84 | 17.40 | 48.10 |
| 50 | 0.01 | 14.72 | 0.22 | 2.09 | 0.19 | 0.99 | 1.77 | 12.56 | 4.72 | 56.08 | 57.24 | 16.58 | 59.59 |
| 60 | 0.02 | 16.88 | 0.30 | 2.14 | 0.24 | 1.52 | 1.24 | 12.67 | 9.44 | 54.86 | 56.90 | 17.74 | 59.60 |
| 70 | 0.11 | 15.42 | 0.20 | 2.19 | 0.17 | 1.31 | 1.04 | 11.19 | 11.58 | 36.75 | 39.91 | 16.20 | 43.08 |
| 80 | 0.55 | 15.01 | 0.20 | 1.77 | 0.12 | 1.06 | 0.91 | 9.69 | 12.40 | 30.84 | 34.85 | 14.65 | 37.81 |
| 90 | 3.31 | 16.85 | 0.20 | 1.36 | 0.21 | 1.85 | 0.69 | 7.91 | 13.14 | 32.28 | 35.96 | 16.55 | 39.58 |
| 100 | 21.54 | 20.06 | 0.16 | 1.00 | 0.20 | 2.50 | 0.55 | 6.69 | 19.17 | 24.75 | 32.48 | 19.49 | 37.88 |
| 110 | 120.59 | 15.12 | 0.27 | 1.19 | 0.05 | 1.66 | 0.62 | 6.49 | 39.83 | 16.55 | 43.77 | 14.85 | 46.22 |
| 120 | 186.00 | 16.79 | 0.19 | 1.67 | 0.05 | 1.28 | 0.58 | 7.86 | 32.04 | 10.18 | 34.07 | 17.84 | 38.46 |
| 130 | 203.89 | 21.15 | 0.07 | 1.81 | 0.07 | 1.16 | 0.63 | 8.85 | 21.22 | 8.06 | 23.25 | 22.47 | 32.34 |
| 140 | 236.83 | 23.02 | 0.20 | 1.80 | 0.09 | 1.07 | 0.71 | 9.06 | 15.86 | 9.13 | 18.97 | 24.33 | 30.85 |
| 150 | 264.19 | 24.40 | 0.28 | 1.72 | 0.10 | 1.19 | 0.77 | 8.85 | 11.98 | 11.15 | 17.37 | 25.40 | 30.77 |

[Figure]

**Figure S1.** V8R_NO_561 Northern polar winter day

**Table S3.** Nitric oxide error budget for Northern polar winter night. All uncertainties are $1\sigma$.

| altitude (km) | mean target (ppmv) | NLTE (%) | interf (%) | ILS (%) | shift (%) | offset (%) | gain (%) | spectro (%) | T+LOS (%) | noise (%) | random (%) | syst (%) | total (%) |
|---|---|---|---|---|---|---|---|---|---|---|---|---|---|
| 40 | 0.00 | 3.57 | 1.63 | 0.86 | 0.83 | 3.14 | 7.44 | 4.84 | 2.12 | 80.82 | 81.28 | 5.92 | 81.50 |
| 50 | 0.00 | 7.21 | 0.20 | 1.10 | 0.22 | 0.93 | 1.88 | 7.01 | 3.91 | 54.40 | 55.00 | 7.49 | 55.51 |
| 60 | 0.04 | 14.09 | 0.30 | 1.44 | 0.34 | 1.52 | 1.28 | 9.96 | 8.56 | 48.63 | 50.92 | 12.25 | 52.37 |
| 70 | 0.27 | 15.31 | 0.23 | 1.48 | 0.30 | 1.84 | 1.03 | 10.07 | 10.45 | 37.56 | 41.08 | 13.22 | 43.16 |
| 80 | 0.73 | 16.73 | 0.21 | 1.34 | 0.14 | 1.71 | 0.89 | 9.21 | 9.84 | 37.39 | 40.86 | 14.00 | 43.19 |
| 90 | 4.52 | 16.52 | 0.22 | 1.23 | 0.21 | 2.15 | 0.73 | 8.35 | 11.14 | 37.77 | 40.81 | 15.33 | 43.60 |
| 100 | 22.49 | 19.71 | 0.23 | 1.10 | 0.22 | 2.92 | 0.65 | 7.93 | 20.52 | 33.72 | 40.87 | 18.68 | 44.94 |
| 110 | 111.02 | 17.29 | 0.24 | 1.26 | 0.09 | 2.16 | 0.62 | 7.69 | 33.70 | 23.79 | 42.15 | 17.03 | 45.45 |
| 120 | 202.51 | 17.20 | 0.14 | 1.48 | 0.02 | 0.97 | 0.56 | 7.85 | 30.09 | 10.41 | 32.79 | 17.30 | 37.07 |
| 130 | 202.15 | 20.03 | 0.07 | 1.60 | 0.05 | 1.20 | 0.53 | 8.28 | 21.49 | 9.67 | 25.24 | 19.80 | 32.08 |
| 140 | 216.22 | 20.85 | 0.15 | 1.61 | 0.05 | 1.26 | 0.54 | 8.29 | 17.17 | 11.83 | 22.87 | 20.48 | 30.70 |
| 150 | 228.36 | 20.73 | 0.20 | 1.55 | 0.06 | 1.32 | 0.53 | 7.97 | 13.99 | 13.38 | 21.45 | 20.32 | 29.55 |

[Figure]

**Figure S2.** V8R_NO_561 Northern polar winter night

**Table S4.** Nitric oxide error budget for Northern polar spring day. All uncertainties are $1\sigma$.

| altitude (km) | mean target (ppmv) | NLTE (%) | interf (%) | ILS (%) | shift (%) | offset (%) | gain (%) | spectro (%) | T+LOS (%) | noise (%) | random (%) | syst (%) | total (%) |
|---|---|---|---|---|---|---|---|---|---|---|---|---|---|
| 20 | 0.00 | 5.13 | 0.37 | 0.49 | 0.08 | 0.52 | 2.02 | 5.77 | 1.50 | 21.95 | 22.48 | 6.58 | 23.42 |
| 30 | 0.01 | 9.68 | 0.23 | 1.58 | 0.49 | 1.31 | 2.79 | 9.96 | 1.83 | 34.48 | 35.16 | 12.71 | 37.38 |
| 40 | 0.01 | 5.79 | 0.16 | 1.44 | 0.32 | 0.70 | 1.81 | 11.09 | 1.31 | 23.98 | 25.19 | 10.22 | 27.19 |
| 50 | 0.01 | 3.76 | 0.06 | 2.16 | 0.20 | 0.21 | 2.04 | 10.83 | 1.58 | 18.63 | 19.76 | 9.97 | 22.13 |
| 60 | 0.01 | 15.60 | 0.18 | 1.59 | 0.21 | 1.41 | 0.97 | 9.16 | 4.05 | 49.61 | 51.02 | 14.38 | 53.01 |
| 70 | 0.02 | 18.63 | 0.15 | 1.79 | 0.22 | 1.91 | 0.69 | 9.03 | 7.55 | 48.10 | 50.18 | 16.98 | 52.98 |
| 80 | 0.06 | 23.62 | 0.18 | 1.59 | 0.25 | 2.73 | 0.60 | 8.76 | 10.72 | 53.96 | 56.89 | 20.85 | 60.59 |
| 90 | 0.53 | 31.19 | 0.21 | 1.46 | 0.36 | 3.88 | 0.73 | 8.06 | 16.09 | 51.53 | 56.71 | 27.45 | 63.00 |
| 100 | 5.40 | 41.71 | 0.19 | 1.25 | 0.35 | 3.85 | 0.62 | 7.08 | 22.58 | 38.11 | 48.84 | 37.20 | 61.39 |
| 110 | 41.01 | 33.21 | 0.18 | 1.37 | 0.10 | 2.46 | 0.57 | 7.33 | 24.99 | 24.10 | 39.99 | 27.76 | 48.68 |
| 120 | 106.33 | 23.49 | 0.14 | 1.50 | 0.13 | 1.30 | 0.53 | 7.96 | 23.85 | 12.29 | 29.64 | 21.46 | 36.60 |
| 130 | 183.81 | 22.66 | 0.03 | 1.55 | 0.11 | 1.09 | 0.56 | 8.43 | 19.34 | 7.47 | 23.07 | 22.05 | 31.91 |
| 140 | 286.41 | 21.89 | 0.04 | 1.56 | 0.07 | 0.87 | 0.60 | 8.50 | 15.99 | 6.81 | 19.14 | 22.14 | 29.27 |
| 150 | 377.69 | 23.07 | 0.09 | 1.51 | 0.04 | 0.92 | 0.64 | 8.30 | 12.92 | 8.36 | 17.32 | 23.27 | 29.01 |

[Figure]

**Figure S3.** V8R_NO_561 Northern polar spring day

**Table S5.** Nitric oxide error budget for Northern polar spring night. All uncertainties are $1\sigma$.

| altitude (km) | mean target (ppmv) | NLTE (%) | interf (%) | ILS (%) | shift (%) | offset (%) | gain (%) | spectro (%) | T+LOS (%) | noise (%) | random (%) | syst (%) | total (%) |
|---|---|---|---|---|---|---|---|---|---|---|---|---|---|
| 40 | 0.00 | 6.22 | 0.53 | 0.19 | 0.02 | 0.80 | 0.97 | 1.17 | 1.21 | 41.89 | 42.32 | 2.75 | 42.41 |
| 50 | 0.00 | 6.97 | 0.22 | 0.65 | 0.14 | 0.82 | 1.61 | 6.91 | 2.40 | 36.55 | 37.20 | 7.59 | 37.97 |
| 60 | 0.01 | 15.76 | 0.27 | 1.19 | 0.14 | 1.81 | 2.02 | 8.17 | 4.54 | 54.03 | 55.56 | 13.30 | 57.13 |
| 70 | 0.13 | 2.06 | 0.14 | 0.50 | 0.23 | 1.67 | 0.70 | 6.40 | 4.22 | 41.02 | 41.29 | 6.68 | 41.82 |
| 80 | 0.13 | 4.53 | 0.28 | 0.69 | 0.34 | 2.37 | 0.45 | 5.95 | 6.51 | 51.62 | 52.17 | 6.96 | 52.63 |
| 90 | 0.74 | 28.24 | 0.30 | 1.11 | 0.40 | 3.28 | 0.91 | 7.59 | 18.24 | 43.91 | 50.59 | 23.88 | 55.94 |
| 100 | 4.78 | 28.25 | 0.28 | 1.24 | 0.33 | 3.81 | 0.68 | 8.09 | 22.58 | 38.69 | 47.01 | 26.02 | 53.73 |
| 110 | 37.28 | 21.25 | 0.25 | 1.24 | 0.15 | 2.62 | 0.59 | 7.87 | 25.12 | 27.79 | 38.75 | 20.58 | 43.88 |
| 120 | 79.08 | 17.68 | 0.13 | 1.37 | 0.05 | 1.15 | 0.49 | 8.01 | 22.13 | 12.02 | 25.94 | 18.48 | 31.85 |
| 130 | 111.49 | 18.15 | 0.02 | 1.43 | 0.07 | 1.04 | 0.49 | 7.89 | 19.02 | 7.72 | 21.21 | 19.16 | 28.58 |
| 140 | 144.80 | 18.24 | 0.09 | 1.43 | 0.05 | 1.04 | 0.50 | 7.59 | 16.58 | 9.17 | 19.70 | 19.09 | 27.43 |
| 150 | 167.97 | 18.16 | 0.13 | 1.38 | 0.03 | 1.16 | 0.51 | 7.07 | 14.24 | 11.34 | 19.30 | 18.49 | 26.73 |

[Figure]

**Figure S4.** V8R_NO_561 Northern polar spring night

**Table S6.** Nitric oxide error budget for Northern polar summer day. All uncertainties are $1\sigma$.

| altitude (km) | mean target (ppmv) | NLTE (%) | interf (%) | ILS (%) | shift (%) | offset (%) | gain (%) | spectro (%) | T+LOS (%) | noise (%) | random (%) | syst (%) | total (%) |
|---|---|---|---|---|---|---|---|---|---|---|---|---|---|
| 20 | 0.00 | 1.35 | 0.45 | 0.29 | 0.09 | 0.44 | 2.08 | 7.35 | 1.36 | 19.64 | 19.77 | 7.57 | 21.17 |
| 30 | 0.01 | 4.16 | 0.15 | 2.46 | 0.23 | 0.66 | 2.52 | 9.00 | 1.80 | 17.89 | 18.13 | 10.29 | 20.85 |
| 40 | 0.01 | 2.27 | 0.13 | 0.62 | 0.25 | 0.42 | 1.41 | 8.58 | 1.03 | 15.79 | 15.89 | 8.91 | 18.22 |
| 50 | 0.01 | 1.28 | 0.08 | 1.03 | 0.08 | 0.17 | 1.22 | 8.01 | 0.73 | 18.15 | 18.22 | 8.15 | 19.96 |
| 60 | 0.01 | 8.26 | 0.22 | 1.38 | 0.40 | 1.12 | 1.09 | 8.86 | 5.44 | 41.42 | 42.22 | 10.66 | 43.55 |
| 70 | 0.02 | 15.20 | 0.17 | 1.89 | 0.63 | 1.40 | 1.14 | 10.25 | 18.06 | 38.43 | 43.68 | 15.44 | 46.33 |
| 80 | 0.03 | 15.51 | 0.28 | 1.81 | 0.80 | 2.32 | 0.96 | 10.83 | 35.84 | 41.30 | 55.33 | 17.23 | 57.95 |
| 90 | 0.81 | 24.60 | 0.21 | 1.44 | 0.71 | 2.76 | 0.99 | 8.24 | 48.97 | 35.17 | 61.63 | 22.83 | 65.72 |
| 100 | 20.44 | 19.45 | 0.14 | 1.35 | 0.32 | 2.54 | 0.61 | 7.29 | 22.84 | 25.12 | 35.65 | 17.94 | 39.91 |
| 110 | 53.54 | 15.20 | 0.22 | 1.44 | 0.10 | 1.15 | 0.65 | 7.50 | 27.07 | 13.17 | 31.09 | 15.19 | 34.61 |
| 120 | 81.14 | 19.07 | 0.10 | 1.59 | 0.12 | 1.26 | 0.66 | 8.61 | 28.17 | 9.84 | 30.49 | 20.09 | 36.51 |
| 130 | 141.15 | 20.07 | 0.11 | 1.58 | 0.09 | 1.19 | 0.69 | 8.86 | 27.40 | 9.83 | 29.66 | 21.29 | 36.51 |
| 140 | 234.70 | 20.58 | 0.22 | 1.52 | 0.08 | 1.20 | 0.71 | 8.66 | 26.13 | 11.09 | 28.90 | 21.76 | 36.18 |
| 150 | 311.01 | 21.10 | 0.29 | 1.44 | 0.06 | 1.32 | 0.76 | 8.19 | 23.89 | 12.91 | 27.79 | 21.95 | 35.41 |

[Figure]

**Figure S5.** V8R_NO_561 Northern polar summer day

**Table S7.** Nitric oxide error budget for Northern polar summer night. All uncertainties are $1\sigma$.

| altitude (km) | mean target (ppmv) | NLTE (%) | interf (%) | ILS (%) | shift (%) | offset (%) | gain (%) | spectro (%) | T+LOS (%) | noise (%) | random (%) | syst (%) | total (%) |
|---|---|---|---|---|---|---|---|---|---|---|---|---|---|
| 40 | 0.00 | 6.98 | 0.73 | 0.25 | 0.05 | 0.70 | 1.63 | 1.83 | 1.56 | 44.18 | 44.74 | 2.85 | 44.84 |
| 50 | 0.00 | 10.23 | 0.39 | 0.80 | 0.23 | 1.20 | 3.33 | 3.78 | 3.30 | 38.75 | 40.05 | 6.43 | 40.56 |
| 60 | 0.00 | 10.91 | 0.42 | 0.89 | 0.33 | 2.09 | 1.94 | 6.71 | 8.85 | 59.72 | 60.99 | 9.96 | 61.79 |
| 90 | 0.63 | 18.05 | 0.44 | 1.00 | 0.62 | 3.93 | 0.68 | 7.29 | 27.69 | 46.90 | 55.49 | 16.85 | 57.99 |
| 100 | 11.10 | 17.42 | 0.32 | 1.20 | 0.39 | 3.30 | 0.75 | 7.58 | 22.25 | 32.56 | 41.07 | 15.54 | 43.92 |
| 110 | 50.20 | 14.39 | 0.27 | 1.24 | 0.11 | 1.81 | 0.64 | 7.63 | 26.65 | 19.94 | 34.23 | 14.37 | 37.13 |
| 120 | 86.37 | 15.10 | 0.09 | 1.21 | 0.10 | 1.22 | 0.48 | 7.35 | 24.44 | 9.46 | 26.88 | 15.79 | 31.18 |
| 130 | 101.26 | 15.97 | 0.09 | 1.26 | 0.10 | 1.36 | 0.57 | 7.07 | 20.60 | 10.01 | 23.48 | 16.80 | 28.87 |
| 140 | 131.59 | 15.88 | 0.16 | 1.27 | 0.07 | 1.29 | 0.62 | 6.71 | 18.01 | 11.21 | 21.69 | 16.75 | 27.41 |
| 150 | 160.67 | 15.63 | 0.19 | 1.23 | 0.04 | 1.33 | 0.63 | 6.21 | 15.57 | 12.58 | 20.61 | 16.21 | 26.22 |

[Figure]

**Figure S6.** V8R_NO_561 Northern polar summer night

**Table S8.** Nitric oxide error budget for Northern polar autumn day. All uncertainties are 1σ.

| altitude (km) | mean target (ppmv) | NLTE (%) | interf (%) | ILS (%) | shift (%) | offset (%) | gain (%) | spectro (%) | T+LOS (%) | noise (%) | random (%) | syst (%) | total (%) |
|---|---|---|---|---|---|---|---|---|---|---|---|---|---|
| 20 | 0.00 | 1.99 | 0.42 | 0.12 | 0.10 | 0.47 | 1.28 | 5.10 | 0.76 | 23.11 | 23.16 | 5.51 | 23.81 |
| 30 | 0.01 | 8.57 | 0.15 | 0.76 | 0.29 | 1.01 | 3.66 | 10.56 | 1.81 | 25.59 | 25.96 | 13.56 | 29.29 |
| 40 | 0.01 | 6.22 | 0.14 | 1.73 | 0.24 | 0.85 | 1.94 | 10.62 | 1.03 | 29.04 | 29.25 | 12.17 | 31.68 |
| 50 | 0.00 | 4.80 | 0.14 | 1.55 | 0.07 | 0.72 | 1.12 | 9.32 | 1.54 | 41.13 | 41.34 | 9.93 | 42.52 |
| 60 | 0.00 | 9.46 | 0.23 | 1.57 | 0.06 | 2.20 | 0.67 | 9.42 | 2.98 | 65.92 | 66.30 | 12.03 | 67.39 |
| 70 | 0.01 | 11.83 | 0.29 | 1.69 | 0.12 | 2.45 | 0.62 | 9.91 | 5.57 | 65.99 | 66.57 | 14.17 | 68.06 |
| 80 | 0.05 | 14.93 | 0.38 | 1.68 | 0.18 | 3.19 | 0.59 | 9.55 | 7.68 | 66.49 | 67.43 | 16.17 | 69.34 |
| 90 | 0.58 | 15.74 | 0.41 | 1.51 | 0.21 | 3.85 | 0.57 | 8.76 | 13.13 | 53.28 | 55.53 | 16.43 | 57.91 |
| 100 | 4.43 | 15.56 | 0.41 | 1.33 | 0.18 | 3.88 | 0.52 | 8.19 | 22.82 | 41.21 | 47.63 | 16.65 | 50.45 |
| 110 | 28.69 | 14.24 | 0.32 | 1.42 | 0.09 | 2.48 | 0.52 | 8.37 | 26.75 | 27.84 | 38.97 | 15.91 | 42.09 |
| 120 | 46.92 | 15.69 | 0.09 | 1.53 | 0.03 | 1.04 | 0.52 | 8.62 | 21.74 | 11.37 | 24.80 | 17.64 | 30.43 |
| 130 | 75.55 | 17.46 | 0.09 | 1.52 | 0.07 | 1.33 | 0.54 | 8.45 | 18.83 | 10.75 | 22.06 | 19.09 | 29.17 |
| 140 | 109.21 | 17.76 | 0.17 | 1.47 | 0.08 | 1.47 | 0.54 | 8.12 | 16.80 | 12.77 | 21.56 | 19.15 | 28.84 |
| 150 | 141.70 | 17.26 | 0.21 | 1.38 | 0.09 | 1.52 | 0.53 | 7.58 | 14.76 | 14.16 | 20.95 | 18.42 | 27.90 |

[Figure]

**Figure S7.** V8R_NO_561 Northern polar autumn day

**Table S9.** Nitric oxide error budget for Northern polar autumn night. All uncertainties are $1\sigma$.

| altitude (km) | mean target (ppmv) | NLTE (%) | interf (%) | ILS (%) | shift (%) | offset (%) | gain (%) | spectro (%) | T+LOS (%) | noise (%) | random (%) | syst (%) | total (%) |
|---|---|---|---|---|---|---|---|---|---|---|---|---|---|
| 40 | 0.00 | 5.33 | 0.35 | 0.80 | 0.13 | 2.40 | 0.86 | 6.27 | 2.88 | 69.03 | 69.42 | 5.36 | 69.63 |
| 50 | 0.00 | 12.32 | 0.27 | 1.53 | 0.10 | 1.84 | 0.90 | 9.59 | 4.07 | 64.83 | 66.17 | 9.56 | 66.85 |
| 60 | 0.01 | 14.75 | 0.26 | 1.67 | 0.05 | 2.31 | 0.87 | 10.99 | 5.22 | 60.29 | 61.99 | 12.91 | 63.32 |
| 70 | 0.04 | 11.70 | 0.29 | 1.46 | 0.12 | 2.46 | 0.84 | 10.47 | 6.64 | 58.69 | 59.66 | 13.58 | 61.19 |
| 80 | 0.14 | 16.36 | 0.40 | 1.34 | 0.17 | 2.74 | 0.78 | 10.58 | 8.93 | 58.89 | 60.69 | 15.94 | 62.75 |
| 90 | 1.27 | 14.83 | 0.44 | 1.21 | 0.22 | 3.35 | 0.73 | 9.68 | 12.54 | 47.82 | 50.27 | 15.63 | 52.64 |
| 100 | 7.00 | 15.86 | 0.40 | 1.25 | 0.18 | 3.47 | 0.64 | 9.20 | 24.75 | 38.00 | 46.00 | 17.05 | 49.06 |
| 110 | 39.70 | 14.96 | 0.30 | 1.27 | 0.09 | 2.21 | 0.55 | 8.38 | 28.86 | 26.18 | 39.57 | 15.91 | 42.65 |
| 120 | 69.50 | 15.99 | 0.05 | 1.44 | 0.04 | 0.99 | 0.54 | 8.50 | 25.30 | 10.74 | 28.01 | 17.40 | 32.97 |
| 130 | 83.35 | 17.33 | 0.14 | 1.53 | 0.08 | 1.36 | 0.54 | 8.45 | 21.06 | 11.75 | 24.82 | 18.49 | 30.95 |
| 140 | 101.47 | 17.33 | 0.21 | 1.52 | 0.08 | 1.48 | 0.54 | 8.17 | 18.36 | 13.83 | 23.78 | 18.29 | 30.00 |
| 150 | 118.75 | 16.55 | 0.24 | 1.44 | 0.08 | 1.48 | 0.51 | 7.64 | 15.89 | 14.68 | 22.46 | 17.33 | 28.37 |

[Figure]

**Figure S8.** V8R_NO_561 Northern polar autumn night

**Table S10.** Nitric oxide error budget for Northern midlatitude winter day. All uncertainties are $1\sigma$.

| altitude (km) | mean target (ppmv) | NLTE (%) | interf (%) | ILS (%) | shift (%) | offset (%) | gain (%) | spectro (%) | T+LOS (%) | noise (%) | random (%) | syst (%) | total (%) |
|---|---|---|---|---|---|---|---|---|---|---|---|---|---|
| 20 | 0.00 | 1.98 | 0.79 | 0.19 | 0.06 | 0.47 | 1.44 | 5.65 | 0.66 | 24.72 | 24.80 | 5.95 | 25.50 |
| 30 | 0.00 | 9.58 | 0.20 | 0.67 | 0.36 | 1.00 | 3.06 | 11.90 | 1.63 | 26.46 | 26.84 | 15.05 | 30.78 |
| 40 | 0.01 | 7.44 | 0.06 | 1.66 | 0.19 | 0.65 | 2.27 | 10.55 | 1.21 | 24.17 | 24.54 | 12.58 | 27.57 |
| 50 | 0.01 | 3.58 | 0.10 | 1.20 | 0.06 | 0.61 | 1.58 | 8.90 | 1.34 | 35.81 | 35.98 | 9.27 | 37.16 |
| 60 | 0.01 | 9.38 | 0.24 | 1.26 | 0.07 | 2.35 | 0.98 | 7.60 | 3.99 | 62.81 | 63.29 | 10.40 | 64.14 |
| 70 | 0.02 | 13.46 | 0.29 | 1.46 | 0.15 | 2.78 | 0.82 | 7.56 | 6.31 | 64.84 | 65.66 | 13.50 | 67.03 |
| 80 | 0.07 | 16.32 | 0.32 | 1.22 | 0.18 | 2.98 | 0.66 | 6.98 | 8.48 | 58.56 | 59.68 | 16.29 | 61.87 |
| 90 | 0.72 | 16.88 | 0.29 | 1.15 | 0.24 | 3.42 | 0.60 | 6.86 | 11.66 | 45.57 | 47.62 | 17.03 | 50.58 |
| 100 | 5.72 | 16.26 | 0.24 | 1.19 | 0.23 | 3.30 | 0.61 | 6.93 | 19.80 | 33.25 | 39.41 | 16.42 | 42.69 |
| 110 | 29.72 | 13.11 | 0.24 | 1.26 | 0.11 | 2.20 | 0.54 | 7.04 | 26.60 | 23.57 | 35.99 | 14.00 | 38.62 |
| 120 | 52.23 | 12.62 | 0.14 | 1.36 | 0.04 | 1.15 | 0.48 | 7.30 | 23.76 | 11.68 | 26.78 | 14.13 | 30.28 |
| 130 | 72.61 | 14.62 | 0.03 | 1.38 | 0.10 | 1.13 | 0.54 | 7.38 | 19.77 | 8.49 | 21.94 | 15.91 | 27.10 |
| 140 | 96.13 | 15.52 | 0.07 | 1.36 | 0.12 | 1.11 | 0.63 | 7.26 | 16.98 | 9.02 | 19.80 | 16.57 | 25.82 |
| 150 | 120.35 | 15.96 | 0.12 | 1.30 | 0.13 | 1.16 | 0.70 | 6.91 | 14.31 | 10.68 | 18.65 | 16.65 | 25.00 |

[Figure]

**Figure S9.** V8R_NO_561 Northern midlatitude winter day

**Table S11.** Nitric oxide error budget for Northern midlatitude winter night. All uncertainties are $1\sigma$.

| altitude (km) | mean target (ppmv) | NLTE (%) | interf (%) | ILS (%) | shift (%) | offset (%) | gain (%) | spectro (%) | T+LOS (%) | noise (%) | random (%) | syst (%) | total (%) |
|---|---|---|---|---|---|---|---|---|---|---|---|---|---|
| 60 | 0.00 | 5.87 | 0.39 | 0.91 | 0.15 | 3.40 | 1.72 | 6.64 | 5.75 | 87.07 | 87.57 | 6.24 | 87.80 |
| 70 | 0.02 | 4.77 | 0.33 | 0.79 | 0.17 | 2.83 | 1.04 | 7.35 | 5.04 | 69.67 | 70.04 | 7.71 | 70.47 |
| 80 | 0.16 | 12.05 | 0.34 | 0.83 | 0.48 | 2.91 | 0.52 | 5.92 | 9.44 | 64.87 | 66.53 | 7.82 | 66.99 |
| 90 | 1.30 | 11.65 | 0.25 | 0.92 | 0.41 | 3.20 | 0.52 | 6.99 | 12.84 | 46.62 | 49.50 | 9.17 | 50.34 |
| 100 | 8.93 | 11.83 | 0.23 | 0.92 | 0.32 | 3.12 | 0.55 | 7.33 | 18.89 | 33.37 | 39.27 | 11.53 | 40.92 |
| 110 | 23.75 | 9.98 | 0.20 | 0.97 | 0.11 | 1.93 | 0.75 | 6.24 | 23.60 | 21.25 | 32.19 | 10.77 | 33.94 |
| 120 | 38.76 | 10.40 | 0.08 | 1.05 | 0.05 | 1.19 | 0.50 | 5.86 | 20.72 | 11.52 | 24.03 | 11.40 | 26.59 |
| 130 | 31.41 | 10.57 | 0.11 | 1.06 | 0.06 | 1.39 | 0.49 | 5.59 | 18.26 | 13.12 | 22.85 | 11.39 | 25.53 |
| 140 | 31.51 | 10.44 | 0.20 | 1.04 | 0.06 | 1.59 | 0.51 | 5.33 | 16.57 | 15.48 | 23.10 | 11.03 | 25.60 |
| 150 | 33.27 | 9.99 | 0.24 | 0.98 | 0.06 | 1.67 | 0.50 | 4.94 | 14.82 | 16.43 | 22.60 | 10.34 | 24.86 |

[Figure]

**Figure S10.** V8R_NO_561 Northern midlatitude winter night

**Table S12.** Nitric oxide error budget for Northern midlatitude spring day. All uncertainties are $1\sigma$.

| altitude (km) | mean target (ppmv) | NLTE (%) | interf (%) | ILS (%) | shift (%) | offset (%) | gain (%) | spectro (%) | T+LOS (%) | noise (%) | random (%) | syst (%) | total (%) |
|---|---|---|---|---|---|---|---|---|---|---|---|---|---|
| 20 | 0.00 | 2.13 | 0.37 | 0.25 | 0.09 | 0.50 | 1.47 | 5.62 | 0.97 | 24.50 | 24.58 | 6.00 | 25.30 |
| 30 | 0.00 | 12.04 | 0.17 | 1.56 | 0.32 | 1.06 | 3.52 | 13.05 | 1.92 | 25.00 | 25.42 | 17.71 | 30.98 |
| 40 | 0.01 | 4.60 | 0.11 | 0.92 | 0.30 | 0.57 | 1.92 | 9.73 | 1.11 | 17.33 | 17.63 | 10.56 | 20.55 |
| 50 | 0.01 | 1.82 | 0.08 | 1.14 | 0.06 | 0.27 | 1.43 | 8.68 | 0.94 | 23.05 | 23.21 | 8.70 | 24.79 |
| 60 | 0.01 | 10.62 | 0.24 | 1.38 | 0.16 | 2.00 | 0.63 | 7.71 | 4.39 | 57.23 | 57.79 | 11.53 | 58.93 |
| 70 | 0.02 | 11.54 | 0.21 | 1.36 | 0.15 | 1.74 | 0.78 | 7.51 | 5.64 | 49.01 | 49.82 | 12.15 | 51.28 |
| 80 | 0.04 | 19.43 | 0.37 | 1.42 | 0.37 | 3.27 | 0.88 | 7.28 | 10.59 | 59.96 | 62.04 | 17.41 | 64.44 |
| 90 | 0.38 | 26.56 | 0.46 | 1.51 | 0.49 | 4.39 | 1.02 | 7.11 | 14.51 | 55.72 | 59.29 | 24.06 | 63.99 |
| 100 | 3.13 | 25.54 | 0.38 | 1.52 | 0.43 | 4.05 | 0.93 | 7.43 | 17.90 | 41.83 | 47.48 | 23.31 | 52.90 |
| 110 | 20.18 | 19.16 | 0.35 | 1.57 | 0.25 | 2.66 | 0.84 | 7.62 | 20.98 | 28.47 | 36.41 | 18.99 | 41.06 |
| 120 | 46.71 | 15.65 | 0.21 | 1.49 | 0.04 | 1.26 | 0.51 | 7.80 | 21.47 | 13.42 | 25.90 | 16.74 | 30.84 |
| 130 | 85.67 | 16.07 | 0.06 | 1.40 | 0.12 | 1.04 | 0.54 | 7.74 | 19.82 | 7.88 | 21.91 | 17.22 | 27.86 |
| 140 | 141.22 | 16.01 | 0.05 | 1.32 | 0.15 | 0.93 | 0.69 | 7.55 | 17.89 | 7.44 | 19.70 | 17.43 | 26.31 |
| 150 | 190.28 | 16.19 | 0.11 | 1.23 | 0.17 | 0.99 | 0.83 | 7.15 | 15.59 | 9.17 | 18.43 | 17.44 | 25.37 |

[Figure]

**Figure S11.** V8R_NO_561 Northern midlatitude spring day

**Table S13.** Nitric oxide error budget for Northern midlatitude spring night. All uncertainties are $1\sigma$.

| altitude (km) | mean target (ppmv) | NLTE (%) | interf (%) | ILS (%) | shift (%) | offset (%) | gain (%) | spectro (%) | T+LOS (%) | noise (%) | random (%) | syst (%) | total (%) |
|---|---|---|---|---|---|---|---|---|---|---|---|---|---|
| 60 | 0.00 | 10.30 | 0.43 | 0.93 | 0.20 | 2.55 | 2.17 | 6.57 | 5.21 | 67.94 | 68.80 | 8.37 | 69.31 |
| 70 | 0.04 | 4.51 | 0.19 | 0.69 | 0.14 | 2.23 | 0.62 | 8.07 | 3.93 | 48.70 | 49.07 | 8.41 | 49.78 |
| 80 | 0.08 | 12.62 | 0.34 | 0.84 | 0.28 | 1.96 | 0.72 | 7.44 | 9.66 | 45.52 | 47.51 | 11.32 | 48.84 |
| 90 | 0.47 | 14.27 | 0.62 | 1.04 | 0.48 | 3.84 | 0.62 | 7.98 | 13.85 | 55.89 | 58.48 | 13.41 | 60.00 |
| 100 | 3.52 | 17.35 | 0.46 | 1.13 | 0.37 | 3.75 | 0.47 | 8.62 | 17.48 | 40.32 | 45.36 | 16.28 | 48.20 |
| 110 | 23.93 | 15.74 | 0.34 | 1.27 | 0.16 | 2.33 | 0.54 | 8.33 | 21.39 | 26.47 | 34.91 | 16.26 | 38.51 |
| 120 | 56.89 | 13.89 | 0.11 | 1.26 | 0.06 | 1.19 | 0.46 | 7.52 | 20.50 | 12.94 | 24.64 | 15.28 | 28.99 |
| 130 | 75.24 | 13.76 | 0.08 | 1.25 | 0.09 | 1.22 | 0.50 | 7.03 | 18.91 | 10.35 | 21.94 | 15.02 | 26.59 |
| 140 | 93.09 | 13.54 | 0.18 | 1.23 | 0.09 | 1.32 | 0.53 | 6.63 | 17.35 | 12.43 | 21.81 | 14.52 | 26.20 |
| 150 | 103.91 | 13.02 | 0.24 | 1.17 | 0.08 | 1.44 | 0.53 | 6.11 | 15.56 | 14.09 | 21.57 | 13.64 | 25.52 |

[Figure]

**Figure S12.** V8R_NO_561 Northern midlatitude spring night

**Table S14.** Nitric oxide error budget for Northern midlatitude summer day. All uncertainties are $1\sigma$.

| altitude (km) | mean target (ppmv) | NLTE (%) | interf (%) | ILS (%) | shift (%) | offset (%) | gain (%) | spectro (%) | T+LOS (%) | noise (%) | random (%) | syst (%) | total (%) |
|---|---|---|---|---|---|---|---|---|---|---|---|---|---|
| 20 | 0.00 | 1.49 | 0.43 | 0.35 | 0.10 | 0.46 | 1.79 | 6.53 | 1.06 | 23.70 | 23.78 | 6.78 | 24.73 |
| 30 | 0.01 | 8.28 | 0.16 | 2.06 | 0.26 | 0.85 | 3.56 | 11.76 | 1.87 | 20.49 | 20.76 | 14.72 | 25.45 |
| 40 | 0.01 | 3.59 | 0.12 | 0.69 | 0.25 | 0.50 | 1.76 | 9.27 | 1.07 | 16.94 | 17.05 | 10.01 | 19.77 |
| 50 | 0.01 | 1.40 | 0.07 | 0.86 | 0.05 | 0.25 | 1.40 | 8.22 | 0.74 | 20.11 | 20.20 | 8.31 | 21.85 |
| 60 | 0.01 | 8.33 | 0.28 | 1.22 | 0.22 | 1.89 | 0.65 | 7.89 | 5.28 | 55.09 | 55.62 | 10.35 | 56.57 |
| 70 | 0.01 | 16.69 | 0.41 | 2.17 | 0.48 | 2.80 | 1.14 | 10.69 | 24.09 | 62.53 | 67.57 | 18.24 | 69.99 |
| 80 | 0.02 | 16.57 | 0.37 | 1.43 | 0.40 | 3.09 | 0.57 | 7.96 | 24.76 | 53.24 | 59.45 | 16.21 | 61.63 |
| 90 | 0.84 | 18.97 | 0.36 | 1.41 | 0.52 | 3.65 | 0.82 | 6.87 | 21.01 | 45.77 | 51.16 | 18.50 | 54.40 |
| 100 | 9.52 | 16.41 | 0.34 | 1.40 | 0.45 | 3.24 | 0.87 | 6.74 | 19.87 | 32.80 | 39.11 | 16.44 | 42.42 |
| 110 | 34.32 | 13.85 | 0.39 | 1.50 | 0.20 | 1.97 | 0.70 | 7.13 | 26.24 | 22.51 | 35.00 | 14.82 | 38.01 |
| 120 | 60.42 | 17.68 | 0.19 | 1.43 | 0.13 | 1.12 | 0.53 | 7.61 | 24.22 | 9.35 | 26.36 | 18.80 | 32.38 |
| 130 | 108.14 | 20.77 | 0.05 | 1.35 | 0.23 | 1.14 | 0.85 | 7.72 | 20.93 | 8.26 | 23.15 | 21.56 | 31.64 |
| 140 | 177.11 | 21.82 | 0.16 | 1.28 | 0.25 | 1.21 | 1.03 | 7.58 | 18.54 | 10.34 | 22.09 | 22.38 | 31.44 |
| 150 | 236.45 | 21.96 | 0.24 | 1.20 | 0.25 | 1.30 | 1.14 | 7.20 | 16.04 | 12.40 | 21.29 | 22.28 | 30.82 |

[Figure]

**Figure S13.** V8R_NO_561 Northern midlatitude summer day

**Table S15.** Nitric oxide error budget for Northern midlatitude summer night. All uncertainties are $1\sigma$.

| altitude (km) | mean target (ppmv) | NLTE (%) | interf (%) | ILS (%) | shift (%) | offset (%) | gain (%) | spectro (%) | T+LOS (%) | noise (%) | random (%) | syst (%) | total (%) |
|---|---|---|---|---|---|---|---|---|---|---|---|---|---|
| 40 | 0.00 | 1.16 | 0.63 | 0.12 | 0.09 | 0.52 | 0.47 | 0.81 | 1.33 | 38.10 | 38.15 | 1.11 | 38.16 |
| 50 | 0.00 | 3.47 | 0.38 | 0.42 | 0.27 | 1.02 | 2.66 | 2.92 | 2.23 | 34.34 | 34.57 | 4.28 | 34.83 |
| 60 | 0.00 | 5.39 | 0.49 | 0.94 | 0.32 | 2.20 | 2.04 | 6.44 | 3.75 | 61.57 | 61.96 | 6.84 | 62.34 |
| 70 | 0.01 | 7.09 | 0.64 | 0.98 | 0.65 | 2.63 | 0.45 | 6.22 | 6.70 | 60.30 | 60.94 | 8.14 | 61.48 |
| 80 | 0.04 | 10.81 | 0.72 | 1.04 | 0.85 | 3.07 | 0.44 | 6.23 | 16.83 | 54.71 | 57.71 | 10.71 | 58.69 |
| 90 | 1.11 | 10.28 | 0.65 | 1.30 | 0.74 | 3.54 | 0.72 | 7.62 | 17.83 | 46.31 | 50.20 | 11.06 | 51.40 |
| 100 | 9.27 | 12.84 | 0.57 | 1.49 | 0.51 | 3.12 | 1.11 | 7.91 | 19.85 | 31.64 | 37.85 | 14.30 | 40.46 |
| 110 | 27.36 | 15.27 | 0.41 | 1.41 | 0.14 | 1.63 | 0.70 | 7.99 | 24.58 | 18.26 | 30.85 | 16.98 | 35.21 |
| 120 | 44.30 | 17.69 | 0.07 | 1.27 | 0.14 | 1.30 | 0.57 | 7.48 | 24.53 | 10.35 | 26.90 | 18.91 | 32.88 |
| 130 | 55.02 | 17.94 | 0.35 | 1.22 | 0.18 | 1.82 | 0.79 | 6.90 | 22.22 | 15.27 | 27.39 | 18.75 | 33.19 |
| 140 | 71.41 | 17.24 | 0.43 | 1.18 | 0.17 | 1.89 | 0.84 | 6.41 | 20.22 | 16.87 | 26.81 | 17.86 | 32.21 |
| 150 | 84.75 | 16.07 | 0.45 | 1.11 | 0.15 | 1.83 | 0.84 | 5.84 | 18.06 | 17.03 | 25.31 | 16.53 | 30.23 |

[Figure]

**Figure S14.** V8R_NO_561 Northern midlatitude summer night

**Table S16.** Nitric oxide error budget for Northern midlatitude autumn day. All uncertainties are $1\sigma$.

| altitude (km) | mean target (ppmv) | NLTE (%) | interf (%) | ILS (%) | shift (%) | offset (%) | gain (%) | spectro (%) | T+LOS (%) | noise (%) | random (%) | syst (%) | total (%) |
|---|---|---|---|---|---|---|---|---|---|---|---|---|---|
| 20 | 0.00 | 1.67 | 0.35 | 0.10 | 0.12 | 0.46 | 1.01 | 4.75 | 0.79 | 24.33 | 24.39 | 4.96 | 24.89 |
| 30 | 0.01 | 9.79 | 0.16 | 1.68 | 0.32 | 1.01 | 3.56 | 11.60 | 1.96 | 24.93 | 25.23 | 15.36 | 29.54 |
| 40 | 0.01 | 5.86 | 0.10 | 1.26 | 0.33 | 0.67 | 2.21 | 10.05 | 1.31 | 19.54 | 19.77 | 11.62 | 22.93 |
| 50 | 0.01 | 1.98 | 0.07 | 1.22 | 0.07 | 0.33 | 1.52 | 8.95 | 0.93 | 25.29 | 25.43 | 9.03 | 26.99 |
| 60 | 0.00 | 7.16 | 0.17 | 1.43 | 0.11 | 2.06 | 0.67 | 8.67 | 2.90 | 60.35 | 60.72 | 9.79 | 61.51 |
| 70 | 0.01 | 11.61 | 0.22 | 1.69 | 0.18 | 2.55 | 0.61 | 9.45 | 5.29 | 64.09 | 64.99 | 12.08 | 66.10 |
| 80 | 0.04 | 12.68 | 0.33 | 1.55 | 0.29 | 3.44 | 0.67 | 8.22 | 7.74 | 67.94 | 68.91 | 13.05 | 70.14 |
| 90 | 0.50 | 14.52 | 0.33 | 1.59 | 0.29 | 3.82 | 0.76 | 8.27 | 11.56 | 55.59 | 57.49 | 14.70 | 59.34 |
| 100 | 3.92 | 15.64 | 0.27 | 1.52 | 0.24 | 3.48 | 0.67 | 8.15 | 21.15 | 37.79 | 44.18 | 15.80 | 46.92 |
| 110 | 21.47 | 15.26 | 0.27 | 1.54 | 0.15 | 2.70 | 0.60 | 8.14 | 22.62 | 30.11 | 38.24 | 16.29 | 41.57 |
| 120 | 27.38 | 13.08 | 0.12 | 1.41 | 0.01 | 1.18 | 0.47 | 7.71 | 19.62 | 14.33 | 24.68 | 14.67 | 28.71 |
| 130 | 50.35 | 13.92 | 0.02 | 1.32 | 0.09 | 1.09 | 0.53 | 7.47 | 18.50 | 9.72 | 21.38 | 15.25 | 26.26 |
| 140 | 81.36 | 14.18 | 0.08 | 1.25 | 0.12 | 1.24 | 0.60 | 7.20 | 17.21 | 10.79 | 20.87 | 15.28 | 25.87 |
| 150 | 112.02 | 13.92 | 0.13 | 1.16 | 0.14 | 1.36 | 0.65 | 6.75 | 15.58 | 12.62 | 20.64 | 14.80 | 25.39 |

[Figure]

**Figure S15.** V8R_NO_561 Northern midlatitude autumn day

**Table S17.** Nitric oxide error budget for Northern midlatitude autumn night. All uncertainties are 1σ.

| altitude (km) | mean target (ppmv) | NLTE (%) | interf (%) | ILS (%) | shift (%) | offset (%) | gain (%) | spectro (%) | T+LOS (%) | noise (%) | random (%) | syst (%) | total (%) |
|---|---|---|---|---|---|---|---|---|---|---|---|---|---|
| 50 | 0.00 | 6.63 | 0.37 | 0.62 | 0.25 | 1.36 | 0.80 | 10.23 | 1.34 | 66.20 | 66.23 | 12.23 | 67.35 |
| 60 | 0.00 | 5.54 | 0.33 | 1.17 | 0.13 | 2.28 | 1.72 | 8.63 | 7.91 | 77.26 | 77.78 | 9.82 | 78.40 |
| 70 | 0.06 | 4.80 | 0.06 | 0.75 | 0.10 | 1.19 | 0.92 | 8.41 | 5.57 | 33.51 | 34.17 | 9.10 | 35.36 |
| 80 | 0.06 | 7.68 | 0.23 | 1.02 | 0.19 | 2.28 | 0.70 | 8.17 | 12.35 | 55.71 | 57.30 | 10.28 | 58.21 |
| 90 | 0.38 | 9.79 | 0.30 | 1.31 | 0.20 | 2.93 | 0.73 | 8.41 | 17.44 | 49.10 | 52.41 | 12.04 | 53.78 |
| 100 | 2.32 | 7.39 | 0.19 | 1.48 | 0.09 | 2.56 | 0.42 | 9.36 | 33.14 | 35.76 | 48.83 | 12.02 | 50.28 |
| 110 | 9.21 | 10.46 | 0.14 | 1.30 | 0.07 | 1.86 | 0.66 | 7.65 | 23.55 | 25.49 | 34.92 | 12.57 | 37.11 |
| 120 | 13.98 | 11.76 | 0.12 | 1.33 | 0.10 | 1.98 | 0.57 | 6.93 | 16.00 | 25.81 | 30.68 | 13.16 | 33.39 |
| 130 | 19.18 | 11.19 | 0.17 | 1.24 | 0.13 | 2.06 | 0.51 | 6.40 | 14.75 | 24.03 | 28.51 | 12.44 | 31.10 |
| 140 | 25.30 | 10.62 | 0.20 | 1.18 | 0.13 | 2.06 | 0.48 | 6.00 | 13.72 | 23.28 | 27.32 | 11.77 | 29.75 |
| 150 | 30.46 | 9.82 | 0.20 | 1.09 | 0.12 | 1.98 | 0.44 | 5.50 | 12.46 | 22.00 | 25.56 | 10.86 | 27.77 |

[Figure]

**Figure S16.** V8R_NO_561 Northern midlatitude autumn night

**Table S18.** Nitric oxide error budget for Tropics day. All uncertainties are $1\sigma$.

| altitude (km) | mean target (ppmv) | NLTE (%) | interf (%) | ILS (%) | shift (%) | offset (%) | gain (%) | spectro (%) | T+LOS (%) | noise (%) | random (%) | syst (%) | total (%) |
|---|---|---|---|---|---|---|---|---|---|---|---|---|---|
| 20 | 0.00 | 2.61 | 0.10 | 1.18 | 0.04 | 0.80 | 0.25 | 3.26 | 0.56 | 23.41 | 23.49 | 3.99 | 23.83 |
| 30 | 0.00 | 10.04 | 0.17 | 2.80 | 0.20 | 0.93 | 1.53 | 12.22 | 1.83 | 29.96 | 30.21 | 15.79 | 34.09 |
| 40 | 0.01 | 4.27 | 0.09 | 0.74 | 0.34 | 0.52 | 2.00 | 9.28 | 1.30 | 13.80 | 14.03 | 10.22 | 17.36 |
| 50 | 0.01 | 1.28 | 0.07 | 1.27 | 0.06 | 0.20 | 1.50 | 8.89 | 0.77 | 19.87 | 19.96 | 9.02 | 21.90 |
| 60 | 0.00 | 8.82 | 0.17 | 1.92 | 0.29 | 2.31 | 0.40 | 8.91 | 2.55 | 62.86 | 63.10 | 11.94 | 64.22 |
| 70 | 0.01 | 11.10 | 0.19 | 2.31 | 0.40 | 3.05 | 0.63 | 9.71 | 4.61 | 65.30 | 65.70 | 14.20 | 67.22 |
| 80 | 0.02 | 13.41 | 0.23 | 2.35 | 0.49 | 3.84 | 1.01 | 8.95 | 7.53 | 61.37 | 62.11 | 15.72 | 64.07 |
| 90 | 0.38 | 18.32 | 0.25 | 2.30 | 0.54 | 4.48 | 1.24 | 8.53 | 18.38 | 54.59 | 58.14 | 19.30 | 61.26 |
| 100 | 3.87 | 18.53 | 0.25 | 2.20 | 0.49 | 4.24 | 1.20 | 8.37 | 26.33 | 44.93 | 52.72 | 19.25 | 56.12 |
| 110 | 16.35 | 14.96 | 0.21 | 2.10 | 0.30 | 2.79 | 0.91 | 8.58 | 27.67 | 30.61 | 41.71 | 16.53 | 44.87 |
| 120 | 26.08 | 14.25 | 0.09 | 1.84 | 0.03 | 1.07 | 0.54 | 8.83 | 21.18 | 11.63 | 24.40 | 16.57 | 29.50 |
| 130 | 51.20 | 16.17 | 0.03 | 1.62 | 0.14 | 1.15 | 0.60 | 8.70 | 17.37 | 8.21 | 19.41 | 18.28 | 26.66 |
| 140 | 87.67 | 16.74 | 0.07 | 1.49 | 0.17 | 1.24 | 0.69 | 8.38 | 15.06 | 9.60 | 18.05 | 18.65 | 25.96 |
| 150 | 117.41 | 16.55 | 0.10 | 1.35 | 0.18 | 1.27 | 0.74 | 7.82 | 12.91 | 11.24 | 17.32 | 18.21 | 25.14 |

[Figure]

**Figure S17.** V8R_NO_561 Tropics day

**Table S19.** Nitric oxide error budget for Tropics night. All uncertainties are $1\sigma$.

| altitude (km) | mean target (ppmv) | NLTE (%) | interf (%) | ILS (%) | shift (%) | offset (%) | gain (%) | spectro (%) | T+LOS (%) | noise (%) | random (%) | syst (%) | total (%) |
|---|---|---|---|---|---|---|---|---|---|---|---|---|---|
| 60 | 0.00 | 8.48 | 0.49 | 2.47 | 0.24 | 2.73 | 3.06 | 9.83 | 6.23 | 81.96 | 82.41 | 12.54 | 83.36 |
| 90 | 0.19 | 10.04 | 0.27 | 2.20 | 0.34 | 3.25 | 1.04 | 9.41 | 16.83 | 47.17 | 50.31 | 13.54 | 52.10 |
| 100 | 1.66 | 11.94 | 0.21 | 2.20 | 0.26 | 2.86 | 0.82 | 9.76 | 20.06 | 35.89 | 41.37 | 15.19 | 44.07 |
| 110 | 7.13 | 12.86 | 0.12 | 2.01 | 0.09 | 1.99 | 0.75 | 9.13 | 20.44 | 26.48 | 33.66 | 15.60 | 37.10 |
| 120 | 12.72 | 12.41 | 0.10 | 1.75 | 0.14 | 2.05 | 0.60 | 8.10 | 18.36 | 23.43 | 29.97 | 14.65 | 33.36 |
| 130 | 17.12 | 11.70 | 0.13 | 1.59 | 0.19 | 2.23 | 0.53 | 7.39 | 16.61 | 23.25 | 28.79 | 13.66 | 31.87 |
| 140 | 22.60 | 10.99 | 0.14 | 1.49 | 0.19 | 2.20 | 0.49 | 6.86 | 15.27 | 22.49 | 27.40 | 12.78 | 30.23 |
| 150 | 25.25 | 10.07 | 0.14 | 1.36 | 0.18 | 2.06 | 0.45 | 6.24 | 13.75 | 20.89 | 25.22 | 11.68 | 27.79 |

[Figure]

**Figure S18.** V8R_NO_561 Tropics night

**Table S20.** Nitric oxide error budget for Southern midlatitude winter day. All uncertainties are $1\sigma$.

| altitude (km) | mean target (ppmv) | NLTE (%) | interf (%) | ILS (%) | shift (%) | offset (%) | gain (%) | spectro (%) | T+LOS (%) | noise (%) | random (%) | syst (%) | total (%) |
|---|---|---|---|---|---|---|---|---|---|---|---|---|---|
| 20 | 0.00 | 2.20 | 0.37 | 0.22 | 0.04 | 0.46 | 1.27 | 5.02 | 1.05 | 24.15 | 24.23 | 5.38 | 24.82 |
| 30 | 0.00 | 12.53 | 0.25 | 1.43 | 0.37 | 1.22 | 3.49 | 12.40 | 2.34 | 30.76 | 31.39 | 17.13 | 35.76 |
| 40 | 0.01 | 6.28 | 0.09 | 1.55 | 0.24 | 0.59 | 1.90 | 10.24 | 1.28 | 25.11 | 25.48 | 11.58 | 27.98 |
| 50 | 0.01 | 2.08 | 0.09 | 1.05 | 0.04 | 0.37 | 1.21 | 8.24 | 1.13 | 30.77 | 30.87 | 8.35 | 31.98 |
| 60 | 0.01 | 7.36 | 0.17 | 1.12 | 0.25 | 1.87 | 0.72 | 7.34 | 5.47 | 56.58 | 57.06 | 9.41 | 57.83 |
| 70 | 0.02 | 11.79 | 0.21 | 1.33 | 0.32 | 2.32 | 0.81 | 7.38 | 9.02 | 58.21 | 59.26 | 12.60 | 60.59 |
| 80 | 0.07 | 15.89 | 0.24 | 1.21 | 0.36 | 2.66 | 0.96 | 7.30 | 13.97 | 53.45 | 55.83 | 15.83 | 58.03 |
| 90 | 0.87 | 18.47 | 0.24 | 1.28 | 0.34 | 3.21 | 0.75 | 7.64 | 20.51 | 44.34 | 49.68 | 18.20 | 52.91 |
| 100 | 8.57 | 17.15 | 0.21 | 1.38 | 0.26 | 3.22 | 0.61 | 7.98 | 31.36 | 33.65 | 46.72 | 17.42 | 49.87 |
| 110 | 45.65 | 13.30 | 0.22 | 1.51 | 0.08 | 1.89 | 0.58 | 8.33 | 41.40 | 21.04 | 46.81 | 14.76 | 49.08 |
| 120 | 75.08 | 14.57 | 0.10 | 1.51 | 0.07 | 1.20 | 0.54 | 8.32 | 33.46 | 10.82 | 35.41 | 16.37 | 39.01 |
| 130 | 100.98 | 16.53 | 0.06 | 1.43 | 0.10 | 1.30 | 0.54 | 7.99 | 25.71 | 9.42 | 27.57 | 18.18 | 33.03 |
| 140 | 135.85 | 17.08 | 0.13 | 1.36 | 0.09 | 1.36 | 0.54 | 7.60 | 21.46 | 11.09 | 24.35 | 18.55 | 30.61 |
| 150 | 173.61 | 17.00 | 0.17 | 1.25 | 0.08 | 1.41 | 0.53 | 7.04 | 17.85 | 12.80 | 22.23 | 18.19 | 28.72 |

[Figure]

**Figure S19.** V8R_NO_561 Southern midlatitude winter day

**Table S21.** Nitric oxide error budget for Southern midlatitude winter night. All uncertainties are $1\sigma$.

| altitude (km) | mean target (ppmv) | NLTE (%) | interf (%) | ILS (%) | shift (%) | offset (%) | gain (%) | spectro (%) | T+LOS (%) | noise (%) | random (%) | syst (%) | total (%) |
|---|---|---|---|---|---|---|---|---|---|---|---|---|---|
| 50 | 0.00 | 6.75 | 0.26 | 0.95 | 0.24 | 1.00 | 0.44 | 10.84 | 1.67 | 51.28 | 51.76 | 10.89 | 52.90 |
| 60 | 0.01 | 10.79 | 0.23 | 1.35 | 0.11 | 2.12 | 1.59 | 8.87 | 7.33 | 57.06 | 58.33 | 10.56 | 59.28 |
| 70 | 0.04 | 8.78 | 0.17 | 1.11 | 0.16 | 1.96 | 0.89 | 8.80 | 7.92 | 53.71 | 54.75 | 10.50 | 55.75 |
| 80 | 0.19 | 13.71 | 0.20 | 1.17 | 0.22 | 2.15 | 0.83 | 9.55 | 13.94 | 50.19 | 53.08 | 13.49 | 54.77 |
| 90 | 1.57 | 13.28 | 0.23 | 1.08 | 0.28 | 2.94 | 0.70 | 8.92 | 18.57 | 43.98 | 48.63 | 13.45 | 50.46 |
| 100 | 11.39 | 13.85 | 0.25 | 1.05 | 0.31 | 3.06 | 0.59 | 8.33 | 25.16 | 33.66 | 42.86 | 14.16 | 45.14 |
| 110 | 43.33 | 12.57 | 0.19 | 1.10 | 0.13 | 1.91 | 0.49 | 7.66 | 33.53 | 22.53 | 40.91 | 13.42 | 43.05 |
| 120 | 73.97 | 14.44 | 0.06 | 1.25 | 0.06 | 1.12 | 0.50 | 7.59 | 29.57 | 11.46 | 32.08 | 15.69 | 35.71 |
| 130 | 89.41 | 15.91 | 0.08 | 1.33 | 0.10 | 1.38 | 0.55 | 7.45 | 23.98 | 12.03 | 27.36 | 16.84 | 32.13 |
| 140 | 101.30 | 16.09 | 0.13 | 1.33 | 0.10 | 1.44 | 0.56 | 7.19 | 20.64 | 13.38 | 25.30 | 16.73 | 30.33 |
| 150 | 109.92 | 15.58 | 0.16 | 1.28 | 0.09 | 1.45 | 0.55 | 6.72 | 17.71 | 14.23 | 23.54 | 15.93 | 28.42 |

[Figure]

**Figure S20.** V8R_NO_561 Southern midlatitude winter night

**Table S22.** Nitric oxide error budget for Southern midlatitude spring day. All uncertainties are $1\sigma$.

| altitude (km) | mean target (ppmv) | NLTE (%) | interf (%) | ILS (%) | shift (%) | offset (%) | gain (%) | spectro (%) | T+LOS (%) | noise (%) | random (%) | syst (%) | total (%) |
|---|---|---|---|---|---|---|---|---|---|---|---|---|---|
| 20 | 0.00 | 1.46 | 0.37 | 0.28 | 0.07 | 0.46 | 1.17 | 5.08 | 0.82 | 23.12 | 23.21 | 5.15 | 23.77 |
| 30 | 0.00 | 8.46 | 0.13 | 1.53 | 0.34 | 0.76 | 2.82 | 10.36 | 1.67 | 21.11 | 21.47 | 13.32 | 25.27 |
| 40 | 0.01 | 4.85 | 0.07 | 1.14 | 0.25 | 0.45 | 1.80 | 9.62 | 1.11 | 19.83 | 20.03 | 10.68 | 22.70 |
| 50 | 0.01 | 1.79 | 0.09 | 1.10 | 0.06 | 0.29 | 1.25 | 8.28 | 0.86 | 27.34 | 27.42 | 8.41 | 28.68 |
| 60 | 0.01 | 9.91 | 0.25 | 1.41 | 0.37 | 1.96 | 0.68 | 7.94 | 4.24 | 57.42 | 57.87 | 11.57 | 59.01 |
| 70 | 0.01 | 16.77 | 0.31 | 1.52 | 0.61 | 2.55 | 1.14 | 7.27 | 7.23 | 61.33 | 62.39 | 16.32 | 64.49 |
| 80 | 0.03 | 27.34 | 0.42 | 1.68 | 0.67 | 3.83 | 1.55 | 8.51 | 11.38 | 64.85 | 67.21 | 25.66 | 71.94 |
| 90 | 0.40 | 25.13 | 0.46 | 1.60 | 0.57 | 4.51 | 1.32 | 8.08 | 15.14 | 58.55 | 61.48 | 24.48 | 66.18 |
| 100 | 3.41 | 24.05 | 0.42 | 1.64 | 0.42 | 4.09 | 1.04 | 8.37 | 19.68 | 43.90 | 49.25 | 23.63 | 54.63 |
| 110 | 20.40 | 19.21 | 0.38 | 1.69 | 0.20 | 2.75 | 0.76 | 8.63 | 24.96 | 29.91 | 39.70 | 19.91 | 44.41 |
| 120 | 46.16 | 16.88 | 0.19 | 1.62 | 0.03 | 1.15 | 0.53 | 8.65 | 24.73 | 12.52 | 28.15 | 18.44 | 33.65 |
| 130 | 83.89 | 17.73 | 0.01 | 1.53 | 0.08 | 1.03 | 0.54 | 8.44 | 21.82 | 7.44 | 23.33 | 19.40 | 30.34 |
| 140 | 143.16 | 18.14 | 0.11 | 1.45 | 0.08 | 1.06 | 0.57 | 8.11 | 19.40 | 8.25 | 21.32 | 19.71 | 29.03 |
| 150 | 200.40 | 18.18 | 0.18 | 1.35 | 0.08 | 1.16 | 0.59 | 7.58 | 16.81 | 10.53 | 20.16 | 19.45 | 28.01 |

[Figure]

**Figure S21.** V8R_NO_561 Southern midlatitude spring day

**Table S23.** Nitric oxide error budget for Southern midlatitude spring night. All uncertainties are $1\sigma$.

| altitude (km) | mean target (ppmv) | NLTE (%) | interf (%) | ILS (%) | shift (%) | offset (%) | gain (%) | spectro (%) | T+LOS (%) | noise (%) | random (%) | syst (%) | total (%) |
|---|---|---|---|---|---|---|---|---|---|---|---|---|---|
| 60 | 0.01 | 7.59 | 0.39 | 1.60 | 0.14 | 2.64 | 2.74 | 8.29 | 5.18 | 73.91 | 74.33 | 10.40 | 75.06 |
| 70 | 0.03 | 6.42 | 0.22 | 1.22 | 0.18 | 1.48 | 0.91 | 9.17 | 5.25 | 46.55 | 47.05 | 10.52 | 48.21 |
| 80 | 0.05 | 8.08 | 0.40 | 1.27 | 0.32 | 2.57 | 0.72 | 8.79 | 10.15 | 53.53 | 54.68 | 11.44 | 55.86 |
| 90 | 0.45 | 12.19 | 0.49 | 1.44 | 0.39 | 3.39 | 0.65 | 8.94 | 14.83 | 50.49 | 53.18 | 13.55 | 54.88 |
| 100 | 4.74 | 12.56 | 0.51 | 1.26 | 0.44 | 3.45 | 0.42 | 7.57 | 23.83 | 36.36 | 44.23 | 12.76 | 46.03 |
| 110 | 17.14 | 12.20 | 0.38 | 1.21 | 0.20 | 1.90 | 0.53 | 7.22 | 26.99 | 23.24 | 36.03 | 13.31 | 38.40 |
| 120 | 24.89 | 13.28 | 0.05 | 1.28 | 0.05 | 1.28 | 0.46 | 7.24 | 21.67 | 13.95 | 26.02 | 14.81 | 29.94 |
| 130 | 34.27 | 13.40 | 0.15 | 1.27 | 0.12 | 1.50 | 0.53 | 6.93 | 19.36 | 13.80 | 24.15 | 14.62 | 28.23 |
| 140 | 45.05 | 13.03 | 0.23 | 1.24 | 0.13 | 1.57 | 0.56 | 6.58 | 17.59 | 14.64 | 23.34 | 14.01 | 27.23 |
| 150 | 53.15 | 12.25 | 0.27 | 1.17 | 0.13 | 1.55 | 0.56 | 6.08 | 15.69 | 14.74 | 22.02 | 13.03 | 25.59 |

[Figure]

**Figure S22.** V8R_NO_561 Southern midlatitude spring night

**Table S24.** Nitric oxide error budget for Southern midlatitude summer day. All uncertainties are $1\sigma$.

| altitude (km) | mean target (ppmv) | NLTE (%) | interf (%) | ILS (%) | shift (%) | offset (%) | gain (%) | spectro (%) | T+LOS (%) | noise (%) | random (%) | syst (%) | total (%) |
|---|---|---|---|---|---|---|---|---|---|---|---|---|---|
| 20 | 0.00 | 1.38 | 0.49 | 0.31 | 0.29 | 0.45 | 0.63 | 4.75 | 1.15 | 23.98 | 24.06 | 4.80 | 24.53 |
| 30 | 0.01 | 8.27 | 0.30 | 2.70 | 0.39 | 0.73 | 2.12 | 10.63 | 2.07 | 21.05 | 21.43 | 13.50 | 25.33 |
| 40 | 0.01 | 3.24 | 0.03 | 0.70 | 0.33 | 0.36 | 1.70 | 8.69 | 1.26 | 13.38 | 13.53 | 9.34 | 16.44 |
| 50 | 0.01 | 1.57 | 0.07 | 0.95 | 0.05 | 0.20 | 1.21 | 7.97 | 0.95 | 18.92 | 19.01 | 8.13 | 20.67 |
| 60 | 0.01 | 11.58 | 0.26 | 1.12 | 0.72 | 1.66 | 1.60 | 7.07 | 4.36 | 52.01 | 52.74 | 11.53 | 53.99 |
| 70 | 0.01 | 17.29 | 0.34 | 1.52 | 0.93 | 2.27 | 1.93 | 7.01 | 10.31 | 53.92 | 55.51 | 17.10 | 58.09 |
| 80 | 0.03 | 27.36 | 0.26 | 1.63 | 0.82 | 2.72 | 1.66 | 9.55 | 22.13 | 49.24 | 55.29 | 26.65 | 61.38 |
| 90 | 0.79 | 28.19 | 0.21 | 1.28 | 0.81 | 3.62 | 1.53 | 7.23 | 19.14 | 46.71 | 51.77 | 27.06 | 58.42 |
| 100 | 8.42 | 29.51 | 0.12 | 1.38 | 0.45 | 3.37 | 0.76 | 7.61 | 19.21 | 33.92 | 41.08 | 27.83 | 49.62 |
| 110 | 38.82 | 22.97 | 0.27 | 1.66 | 0.06 | 2.29 | 0.60 | 7.99 | 28.03 | 21.62 | 36.75 | 22.41 | 43.04 |
| 120 | 73.09 | 22.30 | 0.26 | 1.64 | 0.13 | 1.36 | 0.55 | 8.12 | 28.02 | 11.40 | 30.83 | 23.08 | 38.51 |
| 130 | 117.05 | 24.57 | 0.04 | 1.51 | 0.09 | 1.23 | 0.50 | 8.08 | 22.91 | 7.44 | 24.76 | 25.30 | 35.40 |
| 140 | 191.05 | 24.78 | 0.10 | 1.42 | 0.03 | 1.02 | 0.48 | 7.86 | 19.19 | 7.21 | 21.08 | 25.60 | 33.16 |
| 150 | 260.92 | 25.34 | 0.20 | 1.31 | 0.03 | 1.01 | 0.47 | 7.44 | 15.63 | 9.00 | 18.77 | 25.95 | 32.03 |

[Figure]

**Figure S23.** V8R_NO_561 Southern midlatitude summer day

**Table S25.** Nitric oxide error budget for Southern midlatitude summer night. All uncertainties are $1\sigma$.

| altitude (km) | mean target (ppmv) | NLTE (%) | interf (%) | ILS (%) | shift (%) | offset (%) | gain (%) | spectro (%) | T+LOS (%) | noise (%) | random (%) | syst (%) | total (%) |
|---|---|---|---|---|---|---|---|---|---|---|---|---|---|
| 40 | 0.00 | 0.55 | 0.72 | 0.49 | 0.16 | 0.47 | 0.39 | 0.23 | 0.75 | 40.37 | 40.39 | 0.85 | 40.40 |
| 50 | 0.00 | 4.83 | 0.39 | 1.12 | 0.41 | 1.11 | 3.33 | 4.26 | 2.85 | 31.60 | 32.04 | 5.98 | 32.59 |
| 60 | 0.00 | 18.82 | 0.53 | 2.12 | 0.30 | 2.47 | 2.55 | 11.29 | 6.88 | 65.57 | 68.00 | 14.90 | 69.61 |
| 70 | 0.05 | 18.34 | 0.26 | 1.83 | 0.32 | 1.54 | 1.19 | 11.84 | 10.36 | 42.03 | 45.12 | 17.93 | 48.56 |
| 80 | 0.07 | 19.25 | 0.42 | 1.19 | 0.53 | 2.38 | 1.10 | 10.88 | 19.59 | 46.62 | 51.54 | 19.97 | 55.27 |
| 90 | 1.21 | 27.40 | 0.32 | 1.92 | 0.76 | 3.38 | 0.85 | 11.28 | 18.11 | 41.71 | 49.17 | 23.35 | 54.43 |
| 100 | 9.97 | 20.55 | 0.31 | 1.55 | 0.58 | 3.25 | 0.83 | 8.38 | 24.79 | 33.63 | 43.63 | 18.68 | 47.46 |
| 110 | 45.44 | 16.59 | 0.32 | 1.32 | 0.27 | 1.93 | 0.72 | 7.14 | 32.14 | 21.41 | 39.56 | 16.08 | 42.70 |
| 120 | 104.74 | 18.73 | 0.23 | 1.26 | 0.01 | 1.03 | 0.51 | 7.31 | 28.40 | 10.15 | 30.84 | 19.11 | 36.28 |
| 130 | 115.48 | 22.60 | 0.09 | 1.45 | 0.13 | 1.05 | 0.66 | 8.00 | 21.53 | 8.42 | 24.50 | 22.64 | 33.36 |
| 140 | 143.19 | 23.99 | 0.25 | 1.56 | 0.14 | 1.07 | 0.76 | 8.17 | 17.81 | 10.23 | 22.69 | 23.52 | 32.68 |
| 150 | 161.77 | 24.15 | 0.35 | 1.58 | 0.13 | 1.16 | 0.79 | 7.91 | 14.79 | 12.11 | 21.86 | 23.19 | 31.87 |

[Figure]

**Figure S24.** V8R_NO_561 Southern midlatitude summer night

**Table S26.** Nitric oxide error budget for Southern midlatitude autumn day. All uncertainties are $1\sigma$.

| altitude (km) | mean target (ppmv) | NLTE (%) | interf (%) | ILS (%) | shift (%) | offset (%) | gain (%) | spectro (%) | T+LOS (%) | noise (%) | random (%) | syst (%) | total (%) |
|---|---|---|---|---|---|---|---|---|---|---|---|---|---|
| 20 | 0.00 | 1.97 | 0.33 | 0.19 | 0.09 | 0.49 | 1.23 | 4.79 | 0.91 | 23.72 | 23.78 | 5.18 | 24.34 |
| 30 | 0.01 | 11.43 | 0.22 | 1.92 | 0.36 | 1.06 | 3.87 | 12.70 | 2.23 | 25.59 | 26.02 | 17.18 | 31.18 |
| 40 | 0.01 | 5.74 | 0.05 | 1.09 | 0.31 | 0.53 | 2.34 | 9.74 | 1.50 | 18.88 | 19.09 | 11.38 | 22.22 |
| 50 | 0.01 | 2.14 | 0.06 | 1.09 | 0.06 | 0.31 | 1.41 | 8.62 | 1.15 | 25.61 | 25.74 | 8.76 | 27.19 |
| 60 | 0.01 | 7.50 | 0.14 | 1.40 | 0.33 | 1.70 | 0.98 | 7.95 | 3.24 | 53.13 | 53.62 | 9.13 | 54.39 |
| 70 | 0.02 | 12.84 | 0.22 | 1.54 | 0.47 | 2.19 | 1.15 | 7.25 | 7.05 | 56.02 | 57.35 | 11.18 | 58.43 |
| 80 | 0.06 | 12.49 | 0.21 | 1.52 | 0.43 | 2.36 | 1.27 | 7.76 | 10.59 | 52.45 | 54.22 | 12.24 | 55.58 |
| 90 | 0.88 | 15.55 | 0.18 | 1.54 | 0.35 | 2.80 | 1.06 | 7.72 | 15.46 | 41.75 | 45.65 | 14.53 | 47.90 |
| 100 | 4.96 | 16.01 | 0.19 | 1.60 | 0.23 | 3.12 | 0.79 | 8.05 | 19.19 | 33.24 | 39.52 | 15.67 | 42.51 |
| 110 | 25.37 | 14.25 | 0.21 | 1.66 | 0.08 | 2.23 | 0.60 | 8.46 | 25.15 | 24.25 | 35.61 | 15.33 | 38.77 |
| 120 | 44.17 | 14.66 | 0.11 | 1.56 | 0.02 | 1.10 | 0.53 | 8.44 | 24.00 | 11.63 | 27.06 | 16.41 | 31.65 |
| 130 | 62.62 | 15.80 | 0.03 | 1.43 | 0.04 | 1.24 | 0.56 | 8.07 | 21.30 | 9.69 | 23.85 | 17.24 | 29.43 |
| 140 | 83.58 | 16.06 | 0.10 | 1.34 | 0.04 | 1.34 | 0.57 | 7.69 | 19.10 | 11.38 | 22.77 | 17.22 | 28.55 |
| 150 | 109.44 | 15.85 | 0.14 | 1.23 | 0.04 | 1.42 | 0.56 | 7.15 | 16.78 | 13.02 | 21.91 | 16.65 | 27.52 |

[Figure]

**Figure S25.** V8R_NO_561 Southern midlatitude autumn day

**Table S27.** Nitric oxide error budget for Southern midlatitude autumn night. All uncertainties are $1\sigma$.

| altitude (km) | mean target (ppmv) | NLTE (%) | interf (%) | ILS (%) | shift (%) | offset (%) | gain (%) | spectro (%) | T+LOS (%) | noise (%) | random (%) | syst (%) | total (%) |
|---|---|---|---|---|---|---|---|---|---|---|---|---|---|
| 50 | 0.00 | 5.90 | 0.18 | 1.00 | 0.15 | 1.08 | 0.33 | 10.74 | 2.09 | 49.46 | 49.81 | 11.02 | 51.02 |
| 60 | 0.01 | 15.86 | 0.25 | 1.84 | 0.13 | 2.14 | 1.63 | 11.38 | 6.25 | 65.08 | 66.86 | 13.98 | 68.31 |
| 70 | 0.07 | 30.27 | 0.16 | 2.59 | 0.14 | 1.55 | 1.14 | 12.57 | 8.24 | 41.07 | 50.33 | 17.49 | 53.28 |
| 80 | 0.10 | 21.66 | 0.34 | 1.81 | 0.32 | 2.54 | 0.83 | 11.86 | 13.08 | 56.51 | 60.49 | 18.05 | 63.13 |
| 90 | 0.90 | 17.97 | 0.32 | 1.40 | 0.35 | 3.23 | 0.65 | 9.47 | 14.91 | 49.08 | 53.08 | 15.50 | 55.29 |
| 100 | 10.00 | 20.01 | 0.27 | 1.15 | 0.30 | 3.15 | 0.57 | 8.36 | 23.00 | 35.56 | 44.21 | 17.93 | 47.71 |
| 110 | 48.61 | 15.38 | 0.21 | 1.05 | 0.14 | 2.10 | 0.45 | 6.90 | 28.91 | 24.13 | 39.00 | 13.69 | 41.33 |
| 120 | 40.55 | 13.24 | 0.08 | 1.29 | 0.03 | 1.23 | 0.48 | 7.02 | 19.77 | 13.92 | 25.18 | 13.36 | 28.50 |
| 130 | 56.35 | 14.34 | 0.04 | 1.35 | 0.05 | 1.13 | 0.50 | 7.13 | 16.72 | 11.12 | 21.60 | 14.03 | 25.75 |
| 140 | 66.77 | 15.03 | 0.10 | 1.36 | 0.05 | 1.25 | 0.51 | 7.08 | 14.62 | 12.64 | 21.35 | 14.05 | 25.56 |
| 150 | 68.64 | 15.11 | 0.15 | 1.32 | 0.06 | 1.38 | 0.50 | 6.79 | 12.66 | 14.34 | 21.49 | 13.51 | 25.38 |

[Figure]

**Figure S26.** V8R_NO_561 Southern midlatitude autumn night

**Table S28.** Nitric oxide error budget for Southern polar winter day. All uncertainties are $1\sigma$.

| altitude (km) | mean target (ppmv) | NLTE (%) | interf (%) | ILS (%) | shift (%) | offset (%) | gain (%) | spectro (%) | T+LOS (%) | noise (%) | random (%) | syst (%) | total (%) |
|---|---|---|---|---|---|---|---|---|---|---|---|---|---|
| 30 | 0.01 | 8.02 | 0.41 | 1.67 | 0.44 | 1.42 | 1.91 | 9.79 | 2.05 | 38.58 | 38.90 | 12.19 | 40.76 |
| 40 | 0.00 | 2.64 | 0.11 | 0.96 | 0.22 | 0.50 | 1.53 | 8.58 | 1.12 | 22.27 | 22.47 | 8.73 | 24.11 |
| 50 | 0.00 | 2.88 | 0.14 | 1.33 | 0.07 | 0.39 | 1.01 | 8.07 | 1.15 | 31.17 | 31.27 | 8.48 | 32.40 |
| 60 | 0.01 | 9.99 | 0.23 | 1.71 | 0.12 | 1.80 | 0.55 | 9.05 | 9.62 | 55.54 | 56.81 | 11.78 | 58.01 |
| 70 | 0.03 | 13.24 | 0.24 | 1.81 | 0.22 | 1.99 | 0.59 | 9.63 | 16.46 | 56.05 | 58.93 | 14.69 | 60.73 |
| 80 | 0.17 | 16.32 | 0.30 | 1.65 | 0.27 | 2.18 | 0.69 | 9.15 | 20.42 | 50.78 | 55.45 | 16.70 | 57.91 |
| 90 | 1.94 | 17.61 | 0.27 | 1.53 | 0.29 | 2.61 | 0.65 | 8.27 | 20.62 | 39.15 | 44.95 | 18.05 | 48.44 |
| 100 | 19.25 | 16.54 | 0.24 | 1.45 | 0.27 | 2.76 | 0.58 | 7.60 | 31.81 | 30.15 | 44.50 | 16.80 | 47.56 |
| 110 | 83.47 | 12.72 | 0.25 | 1.47 | 0.11 | 1.82 | 0.53 | 7.56 | 53.94 | 20.40 | 58.02 | 13.55 | 59.58 |
| 120 | 135.27 | 15.50 | 0.10 | 1.46 | 0.09 | 1.11 | 0.53 | 8.10 | 39.66 | 9.79 | 41.05 | 17.11 | 44.48 |
| 130 | 170.99 | 18.54 | 0.10 | 1.42 | 0.13 | 1.33 | 0.61 | 8.25 | 25.84 | 10.16 | 27.99 | 20.10 | 34.46 |
| 140 | 199.87 | 19.33 | 0.20 | 1.38 | 0.13 | 1.35 | 0.69 | 8.06 | 19.69 | 11.90 | 23.30 | 20.72 | 31.18 |
| 150 | 226.53 | 19.24 | 0.26 | 1.29 | 0.12 | 1.37 | 0.72 | 7.60 | 15.40 | 13.41 | 20.83 | 20.37 | 29.14 |

[Figure]

**Figure S27.** V8R_NO_561 Southern polar winter day

**Table S29.** Nitric oxide error budget for Southern polar winter night. All uncertainties are $1\sigma$.

| altitude (km) | mean target (ppmv) | NLTE (%) | interf (%) | ILS (%) | shift (%) | offset (%) | gain (%) | spectro (%) | T+LOS (%) | noise (%) | random (%) | syst (%) | total (%) |
|---|---|---|---|---|---|---|---|---|---|---|---|---|---|
| 40 | 0.00 | 13.98 | 0.63 | 1.90 | 0.15 | 1.59 | 3.26 | 9.46 | 4.75 | 63.21 | 64.92 | 10.29 | 65.73 |
| 50 | 0.00 | 20.14 | 0.41 | 2.14 | 0.24 | 1.20 | 2.52 | 12.17 | 5.27 | 59.59 | 62.66 | 14.81 | 64.39 |
| 60 | 0.01 | 25.60 | 0.34 | 2.51 | 0.11 | 2.07 | 1.40 | 14.09 | 7.26 | 59.16 | 63.86 | 18.47 | 66.48 |
| 70 | 0.09 | 24.85 | 0.22 | 2.24 | 0.38 | 2.03 | 0.82 | 13.63 | 12.33 | 53.65 | 59.10 | 18.73 | 62.00 |
| 80 | 0.90 | 24.16 | 0.20 | 1.81 | 0.48 | 2.61 | 0.79 | 11.84 | 14.90 | 52.51 | 57.74 | 19.49 | 60.94 |
| 90 | 6.00 | 21.58 | 0.24 | 1.38 | 0.42 | 2.70 | 0.78 | 9.29 | 19.63 | 40.99 | 47.76 | 18.61 | 51.26 |
| 100 | 26.27 | 27.56 | 0.31 | 1.25 | 0.45 | 3.00 | 0.74 | 8.27 | 24.33 | 35.73 | 45.87 | 24.57 | 52.04 |
| 110 | 88.34 | 21.36 | 0.32 | 1.21 | 0.16 | 2.11 | 0.56 | 7.05 | 36.30 | 23.86 | 44.94 | 19.48 | 48.98 |
| 120 | 144.64 | 17.57 | 0.18 | 1.28 | 0.08 | 1.03 | 0.45 | 7.14 | 31.70 | 10.14 | 34.04 | 17.64 | 38.34 |
| 130 | 191.17 | 19.46 | 0.05 | 1.46 | 0.09 | 1.01 | 0.52 | 7.93 | 23.55 | 8.22 | 25.58 | 20.31 | 32.66 |
| 140 | 237.07 | 21.55 | 0.17 | 1.57 | 0.07 | 1.10 | 0.57 | 8.35 | 19.09 | 10.26 | 22.81 | 22.09 | 31.75 |
| 150 | 262.90 | 23.13 | 0.26 | 1.61 | 0.05 | 1.24 | 0.59 | 8.38 | 15.57 | 12.29 | 21.85 | 22.92 | 31.67 |

[Figure]

**Figure S28.** V8R_NO_561 Southern polar winter night

**Table S30.** Nitric oxide error budget for Southern polar spring day. All uncertainties are $1\sigma$.

| altitude (km) | mean target (ppmv) | NLTE (%) | interf (%) | ILS (%) | shift (%) | offset (%) | gain (%) | spectro (%) | T+LOS (%) | noise (%) | random (%) | syst (%) | total (%) |
|---|---|---|---|---|---|---|---|---|---|---|---|---|---|
| 20 | 0.00 | 1.54 | 0.28 | 1.15 | 0.14 | 0.47 | 0.88 | 5.16 | 0.79 | 20.76 | 21.04 | 4.51 | 21.52 |
| 30 | 0.00 | 3.40 | 0.12 | 2.25 | 0.29 | 0.55 | 2.00 | 9.21 | 1.26 | 14.40 | 14.80 | 9.78 | 17.74 |
| 40 | 0.01 | 2.00 | 0.05 | 0.68 | 0.22 | 0.25 | 1.57 | 8.46 | 0.80 | 14.02 | 14.18 | 8.65 | 16.60 |
| 50 | 0.01 | 1.66 | 0.09 | 1.28 | 0.09 | 0.20 | 1.28 | 8.50 | 0.77 | 24.55 | 24.65 | 8.61 | 26.11 |
| 60 | 0.01 | 11.05 | 0.23 | 2.15 | 0.43 | 1.81 | 0.98 | 9.29 | 3.91 | 57.35 | 58.04 | 12.40 | 59.35 |
| 70 | 0.02 | 15.61 | 0.17 | 2.71 | 0.56 | 1.80 | 1.46 | 9.86 | 7.16 | 50.60 | 51.99 | 16.23 | 54.46 |
| 80 | 0.06 | 20.26 | 0.29 | 2.87 | 0.72 | 2.65 | 2.13 | 8.97 | 10.31 | 53.53 | 55.66 | 19.62 | 59.01 |
| 90 | 0.59 | 22.62 | 0.34 | 2.74 | 0.81 | 3.90 | 2.25 | 8.93 | 13.58 | 52.80 | 55.52 | 22.58 | 59.93 |
| 100 | 4.63 | 19.86 | 0.35 | 2.46 | 0.72 | 3.94 | 1.98 | 8.27 | 21.59 | 42.56 | 48.41 | 20.56 | 52.60 |
| 110 | 28.33 | 17.65 | 0.36 | 2.33 | 0.48 | 2.66 | 1.60 | 8.11 | 29.52 | 30.13 | 42.61 | 18.88 | 46.60 |
| 120 | 50.99 | 19.19 | 0.18 | 1.88 | 0.05 | 0.96 | 0.69 | 8.50 | 26.58 | 10.96 | 29.04 | 20.71 | 35.67 |
| 130 | 78.75 | 22.00 | 0.05 | 1.56 | 0.18 | 1.08 | 0.66 | 8.64 | 21.31 | 7.71 | 23.11 | 23.28 | 32.81 |
| 140 | 122.06 | 22.77 | 0.16 | 1.40 | 0.24 | 1.21 | 0.91 | 8.45 | 18.09 | 10.33 | 21.42 | 23.87 | 32.07 |
| 150 | 164.81 | 22.53 | 0.23 | 1.26 | 0.26 | 1.29 | 1.08 | 8.01 | 15.25 | 12.57 | 20.45 | 23.42 | 31.09 |

[Figure]

**Figure S29.** V8R_NO_561 Southern polar spring day

**Table S31.** Nitric oxide error budget for Southern polar spring night. All uncertainties are $1\sigma$.

| altitude (km) | mean target (ppmv) | NLTE (%) | interf (%) | ILS (%) | shift (%) | offset (%) | gain (%) | spectro (%) | T+LOS (%) | noise (%) | random (%) | syst (%) | total (%) |
|---|---|---|---|---|---|---|---|---|---|---|---|---|---|
| 40 | 0.00 | 1.02 | 0.61 | 1.33 | 0.03 | 0.96 | 2.03 | 1.11 | 1.72 | 52.16 | 52.26 | 1.49 | 52.28 |
| 50 | 0.00 | 9.07 | 0.50 | 3.93 | 0.44 | 1.38 | 7.65 | 5.53 | 2.51 | 38.96 | 40.32 | 9.35 | 41.39 |
| 60 | 0.01 | 12.84 | 0.36 | 2.42 | 0.29 | 1.95 | 4.09 | 9.06 | 4.58 | 56.72 | 57.96 | 12.38 | 59.26 |
| 70 | 0.20 | 8.36 | 0.08 | 1.11 | 0.22 | 1.58 | 0.95 | 8.44 | 6.50 | 33.35 | 34.86 | 9.22 | 36.06 |
| 80 | 0.23 | 12.89 | 0.25 | 1.77 | 0.44 | 1.69 | 0.84 | 7.69 | 14.35 | 41.22 | 44.55 | 12.36 | 46.23 |
| 90 | 0.95 | 21.27 | 0.60 | 2.23 | 0.85 | 3.65 | 2.57 | 7.83 | 18.56 | 50.27 | 54.56 | 20.84 | 58.41 |
| 100 | 5.84 | 17.55 | 0.63 | 2.01 | 0.74 | 3.78 | 1.87 | 7.38 | 23.16 | 40.87 | 47.91 | 17.23 | 50.92 |
| 110 | 25.35 | 17.95 | 0.55 | 1.73 | 0.42 | 2.57 | 1.35 | 7.51 | 24.50 | 28.98 | 38.68 | 18.27 | 42.78 |
| 120 | 45.81 | 18.28 | 0.17 | 1.46 | 0.02 | 0.97 | 0.55 | 7.72 | 22.39 | 10.88 | 25.47 | 19.19 | 31.89 |
| 130 | 66.91 | 19.58 | 0.12 | 1.43 | 0.16 | 1.08 | 0.65 | 7.83 | 19.33 | 8.63 | 22.01 | 20.29 | 29.94 |
| 140 | 93.13 | 19.61 | 0.24 | 1.42 | 0.17 | 1.21 | 0.74 | 7.61 | 17.10 | 11.02 | 21.35 | 20.11 | 29.33 |
| 150 | 112.01 | 18.83 | 0.29 | 1.36 | 0.16 | 1.26 | 0.77 | 7.13 | 14.94 | 12.52 | 20.59 | 19.12 | 28.10 |

[Figure]

**Figure S30.** V8R_NO_561 Southern polar spring night

**Table S32.** Nitric oxide error budget for Southern polar summer day. All uncertainties are $1\sigma$.

| altitude (km) | mean target (ppmv) | NLTE (%) | interf (%) | ILS (%) | shift (%) | offset (%) | gain (%) | spectro (%) | T+LOS (%) | noise (%) | random (%) | syst (%) | total (%) |
|---|---|---|---|---|---|---|---|---|---|---|---|---|---|
| 20 | 0.00 | 1.87 | 0.55 | 0.38 | 0.24 | 0.42 | 1.61 | 7.00 | 1.27 | 20.96 | 21.13 | 7.10 | 22.29 |
| 30 | 0.01 | 3.75 | 0.28 | 2.78 | 0.38 | 0.65 | 2.03 | 8.84 | 1.70 | 16.51 | 16.78 | 9.92 | 19.49 |
| 40 | 0.01 | 1.84 | 0.06 | 0.63 | 0.29 | 0.29 | 1.52 | 8.34 | 1.04 | 10.10 | 10.28 | 8.56 | 13.37 |
| 50 | 0.01 | 1.70 | 0.07 | 1.04 | 0.11 | 0.16 | 1.19 | 7.86 | 0.89 | 18.16 | 18.26 | 8.01 | 19.94 |
| 60 | 0.01 | 8.98 | 0.19 | 1.24 | 0.40 | 0.90 | 1.57 | 8.17 | 5.07 | 38.75 | 39.87 | 9.51 | 40.98 |
| 70 | 0.01 | 20.24 | 0.20 | 1.84 | 0.82 | 1.66 | 1.49 | 9.08 | 13.62 | 47.45 | 51.25 | 17.65 | 54.21 |
| 80 | 0.04 | 32.63 | 0.17 | 1.62 | 0.77 | 1.69 | 1.36 | 10.05 | 39.54 | 40.26 | 60.75 | 25.82 | 66.01 |
| 90 | 1.73 | 32.55 | 0.08 | 1.22 | 0.74 | 2.16 | 1.27 | 8.29 | 57.10 | 31.36 | 67.53 | 28.63 | 73.35 |
| 100 | 17.39 | 32.76 | 0.08 | 1.26 | 0.48 | 2.63 | 0.95 | 6.89 | 24.08 | 27.22 | 41.30 | 27.30 | 49.51 |
| 110 | 35.22 | 21.69 | 0.24 | 1.51 | 0.03 | 1.57 | 0.81 | 7.16 | 23.05 | 16.15 | 31.25 | 18.52 | 36.32 |
| 120 | 64.28 | 24.03 | 0.18 | 1.57 | 0.19 | 1.31 | 0.64 | 7.95 | 27.68 | 10.25 | 31.76 | 22.53 | 38.94 |
| 130 | 119.09 | 24.74 | 0.03 | 1.52 | 0.17 | 1.27 | 0.68 | 8.42 | 26.37 | 9.01 | 29.80 | 23.99 | 38.26 |
| 140 | 218.90 | 24.02 | 0.13 | 1.47 | 0.14 | 1.14 | 0.80 | 8.47 | 24.16 | 9.32 | 27.28 | 24.06 | 36.38 |
| 150 | 318.79 | 24.44 | 0.20 | 1.42 | 0.10 | 1.11 | 0.89 | 8.28 | 21.22 | 10.60 | 25.01 | 24.65 | 35.11 |

[Figure]

**Figure S31.** V8R_NO_561 Southern polar summer day

**Table S33.** Nitric oxide error budget for Southern polar summer night. All uncertainties are $1\sigma$.

| altitude (km) | mean target (ppmv) | NLTE (%) | interf (%) | ILS (%) | shift (%) | offset (%) | gain (%) | spectro (%) | T+LOS (%) | noise (%) | random (%) | syst (%) | total (%) |
|---|---|---|---|---|---|---|---|---|---|---|---|---|---|
| 40 | 0.00 | 1.68 | 0.81 | 0.40 | 0.07 | 0.89 | 0.11 | 0.74 | 0.74 | 41.50 | 41.52 | 1.88 | 41.57 |
| 50 | 0.00 | 8.49 | 0.55 | 0.92 | 0.30 | 1.68 | 4.33 | 4.23 | 7.18 | 44.67 | 46.12 | 5.71 | 46.47 |
| 60 | 0.00 | 17.43 | 0.36 | 1.80 | 0.21 | 1.93 | 2.76 | 8.84 | 7.63 | 56.32 | 59.19 | 11.09 | 60.22 |
| 70 | 0.04 | 12.61 | 0.28 | 1.60 | 0.37 | 1.61 | 1.15 | 8.27 | 6.21 | 44.54 | 45.44 | 13.85 | 47.50 |
| 80 | 0.05 | 14.93 | 0.45 | 0.75 | 0.55 | 2.84 | 0.56 | 7.67 | 17.44 | 48.82 | 51.92 | 16.81 | 54.58 |
| 90 | 0.71 | 27.53 | 0.53 | 1.30 | 0.64 | 3.88 | 1.13 | 7.38 | 16.62 | 49.04 | 52.70 | 27.12 | 59.27 |
| 100 | 7.07 | 22.21 | 0.48 | 1.20 | 0.57 | 3.56 | 1.10 | 6.08 | 17.72 | 35.74 | 41.63 | 20.11 | 46.24 |
| 110 | 24.61 | 16.37 | 0.43 | 1.12 | 0.26 | 2.44 | 0.89 | 5.70 | 22.39 | 25.16 | 34.94 | 14.90 | 37.99 |
| 120 | 51.27 | 16.06 | 0.14 | 1.18 | 0.08 | 1.00 | 0.47 | 6.64 | 23.43 | 8.96 | 25.89 | 16.24 | 30.57 |
| 130 | 73.42 | 18.60 | 0.13 | 1.34 | 0.15 | 1.24 | 0.60 | 7.30 | 21.02 | 8.98 | 24.20 | 18.43 | 30.42 |
| 140 | 108.96 | 19.55 | 0.23 | 1.42 | 0.12 | 1.29 | 0.68 | 7.39 | 18.83 | 11.32 | 23.71 | 19.01 | 30.39 |
| 150 | 138.32 | 19.98 | 0.29 | 1.42 | 0.08 | 1.35 | 0.71 | 7.15 | 16.53 | 13.01 | 23.27 | 18.85 | 29.95 |

[Figure]

**Figure S32.** V8R_NO_561 Southern polar summer night

**Table S34.** Nitric oxide error budget for Southern polar autumn day. All uncertainties are $1\sigma$.

| altitude (km) | mean target (ppmv) | NLTE (%) | interf (%) | ILS (%) | shift (%) | offset (%) | gain (%) | spectro (%) | T+LOS (%) | noise (%) | random (%) | syst (%) | total (%) |
|---|---|---|---|---|---|---|---|---|---|---|---|---|---|
| 20 | 0.00 | 1.46 | 0.41 | 0.22 | 0.17 | 0.60 | 3.00 | 7.44 | 1.22 | 22.35 | 22.40 | 8.12 | 23.83 |
| 30 | 0.01 | 10.15 | 0.28 | 1.53 | 0.48 | 1.49 | 4.53 | 11.07 | 3.19 | 31.94 | 32.63 | 14.72 | 35.80 |
| 40 | 0.01 | 6.83 | 0.12 | 2.03 | 0.28 | 0.81 | 2.08 | 11.34 | 1.77 | 33.23 | 33.48 | 13.08 | 35.94 |
| 50 | 0.00 | 7.29 | 0.14 | 1.65 | 0.02 | 0.76 | 1.02 | 9.33 | 2.90 | 42.59 | 43.21 | 9.97 | 44.35 |
| 60 | 0.01 | 15.47 | 0.21 | 1.97 | 0.24 | 1.95 | 0.93 | 10.38 | 6.52 | 59.09 | 60.77 | 14.04 | 62.37 |
| 70 | 0.03 | 15.98 | 0.25 | 1.66 | 0.33 | 2.07 | 0.83 | 9.13 | 10.35 | 56.77 | 58.85 | 14.59 | 60.63 |
| 80 | 0.13 | 24.19 | 0.24 | 1.58 | 0.26 | 2.16 | 0.82 | 9.14 | 13.95 | 49.18 | 53.61 | 20.39 | 57.36 |
| 90 | 1.25 | 24.97 | 0.23 | 1.30 | 0.24 | 2.97 | 0.61 | 8.10 | 16.82 | 42.36 | 47.98 | 21.80 | 52.70 |
| 100 | 10.66 | 24.24 | 0.20 | 1.19 | 0.15 | 3.22 | 0.52 | 7.52 | 20.39 | 33.40 | 41.23 | 22.09 | 46.77 |
| 110 | 72.02 | 17.29 | 0.18 | 1.28 | 0.01 | 2.06 | 0.56 | 7.30 | 34.33 | 21.04 | 41.37 | 16.38 | 44.49 |
| 120 | 127.31 | 15.74 | 0.09 | 1.48 | 0.06 | 1.13 | 0.52 | 7.81 | 29.18 | 9.62 | 31.32 | 16.60 | 35.44 |
| 130 | 153.83 | 17.89 | 0.07 | 1.54 | 0.05 | 1.10 | 0.55 | 8.19 | 21.08 | 8.11 | 23.27 | 18.97 | 30.02 |
| 140 | 174.87 | 19.09 | 0.13 | 1.51 | 0.05 | 1.10 | 0.57 | 8.11 | 16.86 | 10.44 | 20.94 | 19.73 | 28.77 |
| 150 | 197.14 | 20.03 | 0.16 | 1.42 | 0.05 | 1.27 | 0.56 | 7.70 | 13.74 | 12.57 | 20.40 | 19.88 | 28.48 |

[Figure]

**Figure S33.** V8R_NO_561 Southern polar autumn day

**Table S35.** Nitric oxide error budget for Southern polar autumn night. All uncertainties are $1\sigma$.

| altitude (km) | mean target (ppmv) | NLTE (%) | interf (%) | ILS (%) | shift (%) | offset (%) | gain (%) | spectro (%) | T+LOS (%) | noise (%) | random (%) | syst (%) | total (%) |
|---|---|---|---|---|---|---|---|---|---|---|---|---|---|
| 50 | 0.00 | 12.50 | 0.17 | 1.09 | 0.10 | 0.90 | 0.52 | 10.93 | 5.02 | 51.04 | 52.72 | 11.39 | 53.93 |
| 60 | 0.02 | 14.20 | 0.18 | 1.29 | 0.12 | 1.39 | 0.89 | 9.91 | 5.74 | 46.88 | 48.81 | 12.35 | 50.35 |
| 70 | 0.08 | 15.11 | 0.19 | 1.22 | 0.26 | 1.86 | 0.81 | 9.44 | 7.32 | 54.03 | 55.74 | 13.75 | 57.41 |
| 80 | 0.40 | 24.27 | 0.23 | 1.15 | 0.33 | 2.03 | 0.86 | 9.78 | 9.36 | 51.90 | 54.94 | 21.29 | 58.93 |
| 90 | 3.69 | 22.46 | 0.19 | 0.96 | 0.38 | 2.81 | 0.77 | 9.11 | 11.91 | 44.58 | 48.17 | 20.15 | 52.22 |
| 100 | 23.66 | 19.85 | 0.17 | 0.79 | 0.26 | 2.82 | 0.69 | 7.81 | 21.06 | 34.16 | 41.30 | 19.22 | 45.55 |
| 110 | 117.69 | 23.13 | 0.17 | 1.09 | 0.07 | 1.90 | 0.53 | 7.24 | 34.86 | 21.78 | 42.74 | 21.32 | 47.77 |
| 120 | 142.48 | 19.78 | 0.10 | 1.56 | 0.05 | 1.04 | 0.53 | 8.26 | 28.29 | 11.34 | 31.60 | 19.82 | 37.31 |
| 130 | 184.74 | 20.21 | 0.05 | 1.70 | 0.05 | 0.88 | 0.54 | 8.71 | 19.72 | 7.14 | 21.87 | 21.21 | 30.47 |
| 140 | 218.33 | 21.26 | 0.11 | 1.70 | 0.03 | 0.99 | 0.54 | 8.74 | 15.31 | 9.43 | 19.15 | 22.12 | 29.26 |
| 150 | 232.97 | 22.05 | 0.14 | 1.62 | 0.02 | 1.18 | 0.51 | 8.40 | 12.17 | 11.82 | 18.72 | 22.32 | 29.13 |

[Figure]

**Figure S34.** V8R_NO_561 Southern polar autumn night

---

## Author Response (AR1)

**Response to the reviewers' comments on amt-2022-260.**

We thank the reviewers for their valuable comments, questions and suggestions. Please find below our detailed point-by-point replies (in blue color) to the comments, which we hope have addressed all satisfactorily, as well as the actions taken on the manuscript.

In addition to the suggested changes by the reviewers, we have added a supplement document with detailed error budget information for all data products (V8H_NO_61 (NOM) for the FR period, and V8R_NO_261 (NOM), V8R_NO_561 (MA), V8R_NOwT_662 (UA) and V8R_TwNO_662 (UA) for the RR period) discussed in the manuscript. Further, we have updated Figure 5 which showed the error budgets only for V8H_NO_61 and V8R_NO_261 for four different atmospheric conditions, by two new figures (Figure 5 and 6) showing the error budgets for all products for a reduced set of atmospheric conditions. The complete set of figures for all atmospheric conditions can be found in the supplement.

**Point-by-point responses to the comments of Referee #1**

*We thank Chris Boone for his thoughtful comments and suggestions, which certainly helped to improve the clarity of the manuscript.*

Comment: This paper describes a new combined data product of NO and thermospheric temperature retrieved from MIPAS measurements. The analysis procedure and error assessment are described. Comparisons are made to previous processing versions.

Overall, nice work. I have no real changes to suggest.

*Reply: Thank you very much!*

Comment: I will point out that this article makes heavy use of acronyms that are not defined. To name some: GRANADA, SAMONA, SMR, NOEM, SNOE, NRLMSIS, ECMWF, ERA, JPL, HITRAN, EUV. While most people reading the article will likely be familiar with these acronyms, it makes it somewhat jargon heavy. Perhaps the greatest concern might be the fact that seasonal acronyms (MAM, JJA, SON) are not defined.

*Reply: All undefined acronyms are now defined in the revised version.*

Comment: In a retrieval paper, it would have made me happy to see a figure showing observed and calculated spectra, to see how well things fit. However, this is just personal preference, the spectroscopist in me.

*Reply: We agree that it could be useful to show measured and modeled spectra in a retrieval paper, in particular, if the paper deals with retrievals from spectral signatures that are difficult to detect. However, since the NO 5.3 um emission is a well-known spectral feature, and further taking into account the already quite exhaustive number of figures in the manuscript, we would prefer not adding additional figures in this particular case.*

Comment: Minor items:

Lines 233 and 450: peroxyacyl nitrate. I believe peroxyacyl nitrate is a class of molecules. Why do you not call it peroxyacetyl nitrate?

*Reply: This has been changed accordingly.*

Comment: Caption to Figure 2: …2006–2012 period. I can't tell if the averages excluded 2005 for some unspecified reason or this was a typo.

*Reply: The reason for excluding 2005 from the composite is that, due to operation interruptions, there is only a poor and uneven temporal coverage in this particular year. We thus decided to remove 2005 from the composite in order to guarantee a homogeneous seasonal coverage.*

Comment: Figure 5: NO error budget for FR (a, c, e) and RR (b, d, f). No 'a, b, c, d, e, and f' labels in the figure.

*Reply: This figure has been replaced by the new Figure 5 and 6.*

Comment: Line 571: Northern hemispheric. Northern Hemispheric

*Reply: This has been changed accordingly.*

Comment: Line 701: In the mesosphere, biases of the version 5 NO data in comparison with correlative measurements, found at 65–100 km, seem to have been considerably reduced or even removed in the new version. The new NO data is likely also in better agreement with NO observations from other satellite instruments in the upper mesosphere, where the MIPAS NO from version 5 was low-biased,

What is the difference between "correlative measurements in the mesosphere" and "observations from other satellite measurements in the upper mesosphere?"

*Reply: There is no difference. Both expressions are used as synonyms in order to avoid repetition. We have rephrased to "In the mesosphere, biases of the version 5 NO data in comparison with correlative measurements, found at 65–100 km, seem to have been considerably reduced or even removed in the new version. The new NO data is likely also in better agreement with correlative measurements in the upper mesosphere, where the MIPAS NO from version 5 was low-biased."*

**Point-by-point responses to the comments of Referee #2**

*We thank Referee #2 for the thoughtful comments and suggestions, which certainly helped to improve the clarity of the manuscript.*

Comment: This is a very well written and comprehensive paper that is of relevance to AMT. I suggest it be published after a few minor issues are addressed:

*Reply: Thank you very much!*

Comment: Lines 18-21: If you're going for completeness, I'd recommend adding the OSIRIS NO measurements (doi:10.1029/2009JD013205, doi:10.1029/2011GL048054). Or, if you're just giving examples, please put "e.g." at the beginning of the list.

*Reply: We have added the OSIRIS reference (Sheese et al., 2011) in the revised manuscript.*

Comment: Line 78: I think "constraints" should be "constrains"

*Reply: You are right. This typo has been corrected.*

Comment: Line 334: In what sense are you using the word significant? According to Table 3, the improvement to the convergence rate is ~0.4-0.7%. Are you saying that the improvement significantly affects mean NO results or simply that the increase of converged retrievals is non-trivial?

*Reply: In the latter sense. We now state that the improvement is noticeable (instead of significant) in order to make this clearer.*

Comment: Section 3: Is there a reason why you use just the diagonal elements for data filtering instead of the retrieval response (i.e., sum of Ak) as is more typical with other instruments? It could be interesting to have the retrieval response plotted in Fig 1 as well.

*Reply: The use of the sum of AK would be adequate to discriminate data points with high a priori information content in case of an optimal estimation approach. Since we use a Tikhonov regularization (smoothing constraint), our retrievals contain in principle no a priori information (except for the a priori profile shape), such that the AK sum is close to one over most of the profile range. In contrast, the AK diagonal indicates the content of **local** information. A small diagonal element means that most of the information comes from other (typically lower) altitudes.*

Comment: Line 363: I'm not sure I understand where the statistical biases come from. Wouldn't leaving those retrievals data points with Akd < 0.03 in the averaging lead to a bias towards the a priori?

*Reply: Statistical biases arise because the averaging kernels (and hence their diagonal elements) depend on the vmr of the retrieved profile. This is because the Jacobian in a logarithmic retrieval scales with the vmr. In consequence, akd-filtering favors high vmr values (low vmr values with smaller akd will be discarded), such that the result is prone to be high-biased. A sentence has been added to clarify this.*

Comment: Figure 2: What is the reason for the worsening vertical resolution in the Northern mid latitudes (MA and UA, especially)?

*Reply: We speculate that the worse vertical resolution around 30N-50N in December is caused by the relatively low vmrs found in that region. Due to the "self-adapting" effect of regularization in a logarithmic retrieval (stronger for low vmrs), resolution is degraded, there.*

Comment: Figure 6: plot titles all have the term "esd," which I don't believe has been defined.

*Reply: esd (estimated standard deviation) is now defined in the revised version.*

Comment: Line 575: I found this sentence a bit confusing. Are you saying that differences between v8 and v5 for MA are consistent with those for UA? Please consider rephrasing.

*Reply: This has been rephrased to "Differences of both MA and UA datasets with respect to their respective predecessor versions are very similar".*

Comment: Figure 15: please add a legend.

*Reply: All symbols and colors used in the panels of this figure are explained in the caption. Thus, we think that an additional legend would be redundant.*

Comment: Line 634: "allow to assess" doesn't sound right. I'd suggest something like "allows for an assessment of"

*Reply: This has been changed accordingly.*

Comment: Lines 634-642: The description of the migrating diurnal tide could use a reference. Perhaps Brasseur and Solomon (and refs therein)?

*Reply: This reference has been be added.*